*Resource*

**EMBO** *reports*

# The proximity-based protein interactome and regulatory logics of the transcription factor p65 NF-κB/RELA

Lisa Leib[1,9], Jana Juli[1,9], Liane Jurida[1], Christin Mayr-Buro[1], Jasmin Priester[1], Hendrik Weiser[1], Stefanie Wirth[1], Simon Hanel[1], Daniel Heylmann[1], Axel Weber [ID][1], M Lienhard Schmitz [ID][2], Argyris Papantonis [ID][3], Marek Bartkuhn[4,5,6], Jochen Wilhelm[5,6,7], Uwe Linne [ID][8], Johanna Meier-Soelch [ID][1]✉ & Michael Kracht [ID][1,6,7]✉

## Abstract

**The protein interactome of p65/RELA, the most active subunit of the transcription factor (TF) NF-κB, has not been previously determined in living cells. Using p65-miniTurbo fusion proteins and biotin tagging, we identify >350 RELA interactors from untreated and IL-1α-stimulated cells, including many TFs (47% of all interactors) and >50 epigenetic regulators belonging to different classes of chromatin remodeling complexes. A comparison with the interactomes of two point mutants of p65 reveals that the interactions primarily require intact dimerization rather than DNA-binding properties. A targeted RNAi screen for 38 interactors and subsequent functional transcriptome and bioinformatics studies identify gene regulatory (sub)networks, each controlled by RELA in combination with one of the TFs ZBTB5, GLIS2, TFE3/TFEB, or S100A8/A9. The large, dynamic and versatile high-resolution interactome of RELA and its gene regulatory logics provides a rich resource and a new framework for explaining how RELA cooperativity determines gene expression patterns.**

**Keywords** RELA; MiniTurboID; Proximity-based Interactome; Genetic Networks; IL-1
**Subject Categories** Chromatin, Transcription & Genomics; Immunology; Methods & Resources

## Introduction

Transcription factors (TFs) comprise a large family of proteins that read and interpret the genome to decode the DNA sequence. TFs are defined by their sequence-specific DNA-binding domains (DBD) and by the ability to induce or repress transcription (Fulton et al, 2009; Vaquerizas et al, 2009). A recent combinatorial survey cataloged a total of 1639 human TFs (http://humantfs.ccbr.utoronto.ca) (Lambert et al, 2018).

TFs are key components of gene regulatory networks in which the spatio-temporal expression patterns of TFs and their auto-regulatory loops determine cell identity, developmental processes, and disease states (Almeida et al, 2021; Fuxman Bass et al, 2015).

So far, for only a limited number of TFs the (direct) binding to chromatin has been mapped comprehensively using ChIPseq or related techniques. However, bioinformatics analyses of sequence motifs deduced from genome-wide DNA footprints in open chromatin regions suggest that transcription factors generally bind cooperatively to enhancers or promoters to execute their gene regulatory functions (Funk et al, 2020; Neph et al, 2012; Vierstra et al, 2020).

In contrast to the wealth of information on (predicted) DNA binding of TFs, we currently lack a global understanding of TF protein–protein interactions (PPIs) and their functional contributions to TF cooperativity within transcriptional networks. A recent, large-scale study examined the basal interactomes of 109 common TFs to find, depending on the method applied, 1538 to 6703 PPI, respectively. This new evidence suggests that TF cooperativity may be largely determined through the repertoire of (dynamic) PPI (Goos et al, 2022).

The REL DBD is found in only 10 (0.6%) of all human TFs, including the five members of the nuclear factor of kappa light polypeptide gene enhancer in B-cells (NF-κB) family of TFs (Lambert et al, 2018). The five NF-κB subunits are evolutionary conserved, inducible regulators of development and disease conditions and are particularly important in regulating innate and adaptive immune responses (Williams and Gilmore, 2020; Zhang et al, 2017).

The p65/RELA NF-κB transcription factor contains a highly structured and conserved N-terminal REL homology domain

[1]Rudolf Buchheim Institute of Pharmacology, Justus Liebig University, Giessen, Germany. [2]Institute of Biochemistry, Justus Liebig University, Giessen, Germany. [3]Institute of Pathology, University Medical Center Göttingen, Göttingen, Germany. [4]Biomedical Informatics and Systems Medicine, Justus Liebig University Giessen, Giessen, Germany. [5]Institute for Lung Health, Justus Liebig University Giessen, Giessen, Germany. [6]Member of the Excellence Cluster Cardio-Pulmonary Institute (CPI), Giessen, Germany. [7]German Center for Lung Research (DZL) and Universities of Giessen and Marburg Lung Center (UGMLC), Giessen, Germany. [8]Mass Spectrometry Facility of the Department of Chemistry, Philipps University, Marburg, Germany. [9]These authors contributed equally: Lisa Leib, Jana Juli. ✉E-mail: johanna.meier-soelch@pharma.med.uni-giessen.de; michael.kracht@pharma.med.uni-giessen.de

(RHD) that is involved in DNA binding and dimerization, as shown by the crystal structure of the p65/p50 NF-κB heterodimer bound to DNA (Chen et al, 1998; Williams and Gilmore, 2020). The C-terminal half of p65/RELA contains two potent transactivation domains (TA$_1$, TA$_2$) and is inducibly phosphorylated at multiple residues (Christian et al, 2016; Viatour et al, 2005). In contrast to the RHD, the C-terminus is highly unstructured as revealed originally by NMR and supported by bioinformatics predictions (Jumper et al, 2021; Schmitz et al, 1994; Schmitz et al, 1995; Varadi et al, 2022). This phenomenon may also underly the previously reported cytokine-dependent conformational switches of p65/RELA (Milanovic et al, 2014).

Both, post-translational modifications and structural flexibility of the p65/RELA C-terminus are features that have very likely evolved to expand the repertoire of PPI under changing conditions and within subcellular compartments. However, the p65/RELA interactome has not yet been determined comprehensively with methods that also cover labile, transient or substoichiometric interactions of p65/RELA with cellular proteins as they occur in living cells.

Accordingly, we have a limited understanding of how exactly p65/RELA cooperates with partner TFs (*in cis* at overlapping DNA elements or in trans across chromosomes), chromatin modifiers, the transcription machinery, and other nuclear cofactors to regulate transcription of specific groups of genes (Bacher et al, 2021).

Biotin-proximity labeling is a relatively new approach by which cells, expressing a bait protein fused to an engineered bacterial biotin ligase (such as BirA* used for BioID), are incubated with biotin for several hours to biotinylate all proteins in close vicinity (i.e., a radius of 1–10 nm) to the fusion protein. After cell lysis, biotinylated proteins are captured on streptavidin affinity matrices and subsequently identified by liquid chromatography-tandem mass spectrometry (LC-MS/MS) (Qin et al, 2021; Roux et al, 2012). Key advantages of this approach include its ability to capture weak or transient interactions from both soluble and insoluble proteins, or subcellular organelles, and the possibility to use high-stringency protein purification methods to reduce background contaminants (Zhou and Zou, 2021). MiniTurbo (mTb, used for miniTurboID) is a recently improved small 28 kDa variant of BirA that more efficiently biotinylates intracellular proteins within minutes in the presence of exogenously added biotin (Branon et al, 2018).

The pro-inflammatory cytokine interleukin-1 (IL-1) rapidly activates p65/RELA in a broad range of cell types (Meier-Soelch et al, 2021). Numerous studies, including our own work, have shown the importance of p65/RELA for the expression of IL-1-target genes, rendering this system an ideal model to study the dynamic p65/RELA interactome (Barter et al, 2021; Jurida et al, 2015; Weiterer et al, 2020).

Here, we report the first proximity-based p65/RELA interactome using inducible wild-type, or mutant p65-miniTurbo fusion proteins devoid of either DNA binding or dimerization properties, to identify 366 high-confidence p65/RELA interactors (HCI) of which 87% are novel. The p65/RELA interactome is highly enriched for nuclear proteins, including 172 TFs (47% of all p65/RELA interactors) and 74 epigenetic regulators (20% of all interactors) and appears to be partially remodeled by phosphorylation of interacting proteins. A targeted siRNA screen for 38 interactors and subsequent functional gene expression studies identify new gene regulatory (sub)networks, each activated or repressed by p65/RELA

in combination with one of the TFs ZBTB5, GLIS2, TFE3/TFEB, or S100A8/S100A9.

Taken together, these data reveal a remarkably large, dynamic and versatile (transcription factor and epigenetic regulator) interactome of p65, which determines gene expression patterns mainly via DNA-independent protein–protein interactions (PPIs).

# Results

## Identification of new p65/RELA interactors using miniTurbo-based proximity labeling

Point mutations in the RHD of p65/RELA, such as glutamate 39 to isoleucine (E/I), inhibit DNA binding, while mutations of two other amino acids, phenylalanine 213 and leucine 215, to aspartic acid (FL/DD), prevent dimerization as previously shown by co-immunoprecipitation and EMSA experiments (Riedlinger et al, 2019) (Fig. 1A, left image). In contrast, the C-terminal half of p65/RELA is highly unstructured as revealed by alphafold (Jumper et al, 2021; Varadi et al, 2022) (Fig. 1A, right image). We reasoned that these features of the RHD and the p65/RELA C-terminus very likely evolved to expand the repertoire of possible protein interactions of p65/RELA under changing conditions and within subcellular compartments.

To comprehensively determine the p65/RELA interactome as it occurs in intact cells, we constructed expression vectors containing wild-type p65/RELA or the E/I and FL/DD mutants fused in frame to a C-terminal HA-tag and a modified miniTurbo biotin ligase (Branon et al, 2018) (Fig. 1B). The fusion proteins were expressed under the control of a tetracycline-sensitive promoter using a single vector system (Fig. 1B). Parental HeLa cells and stable HeLa cell lines genetically edited by CRISPR-Cas9 to have strongly reduced p65/RELA levels were reconstituted with the constructs and showed doxycycline-dependent expression, nuclear translocation of NF-κB and basal as well as IL-1α-inducible transcriptional activity sensitive to the aforementioned mutations, demonstrating that the fusion proteins were functional in the NF-κB system (Appendix Fig. S1A–C).

Following intracellular expression, proteins in the vicinity of p65/RELA were supposed to be modified with biotin in a distance-dependent manner by the miniTurbo part, as shown schematically in Fig. 1C. The efficient biotinylation of both cytosolic and nuclear proteins was validated in p65-deficient cells that were reconstituted with either HA-miniTurbo alone or with p65-HA-miniTurbo (Fig. EV1). Wild-type p65/RELA or the two mutant versions were expressed at comparable levels (Fig. 1D, left panel). Biotinylated proteins were purified from whole-cell extracts of untreated cells or cells exposed to IL-1α for 1 h to cover proteins binding to cytosolic and nuclear p65/RELA, and visualized by streptavidin–HRP conjugates (Fig. 1D, right panel). Biotinylation was strictly dependent on the simultaneous addition of doxycycline and biotin to the cell cultures, as no signals were detected with doxycycline or biotin alone (Fig. 1D, right panel). The latter conditions, together with samples from cells expressing HA-miniTurbo alone (empty vector, EV), served as important negative controls to define specific p65/RELA interactors in the bioinformatics analyses later on (Fig. 1D, gray colors).

Across all conditions, a total of 3928 protein IDs were identified from purified biotinylated proteins by LC-MS/MS from the two

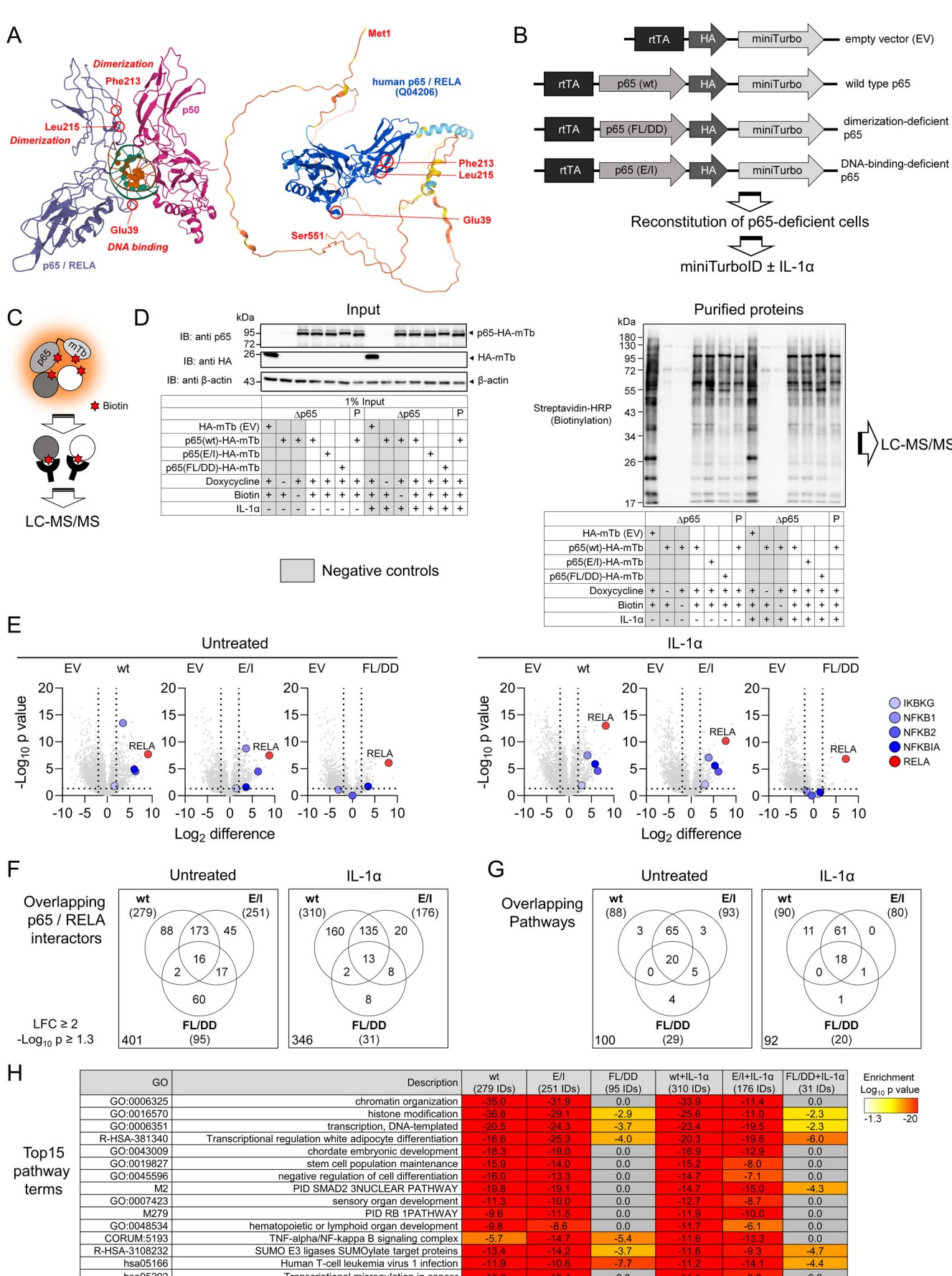

◀  **Figure 1.  Proteome-wide identification of the dimerization- or DNA-binding dependent p65/RELA interactomes by proximity labeling.**

(A) The left graph shows the X-ray crystal structure of a p50/p65 heterodimer bound to DNA as published in (Chen et al, 1998) (PDB 1kvx), while the right graph shows the entire p65 protein structure including the disordered C-terminal half as calculated by alphafold (https://alphafold.ebi.ac.uk/entry/Q04206). Residues required for dimerization (Phe (F) 213, Leu (L) 215) or DNA binding (Glu (E) 39) are indicated in both structures. (B) Scheme of the HA-tagged p65-miniTurbo fusion proteins that were used to reconstitute p65-deficient HeLa cells under the control of a tetracycline-sensitive promoter. F213 and L215 in p65 wild type (wt) were mutated to Asp (FL/DD) for dimerization-deficient p65 or E39 to Ile (E/I) for DNA-binding-deficient p65. (C) Principle of proximity-based biotin tagging. (D) Pools of HeLa cells with CRISPR/Cas9-based suppression of endogenous p65/RELA (Δp65) were transiently transfected (using branched Polyethyleneimine, PEI) with the constructs shown in (B) and their expression was induced with doxycycline (1 μg/ml) for 17 h. At the end of this incubation, intracellular biotinylation was induced by adding 50 μM biotin for 70 min as indicated. Additionally, half of the samples were treated with IL-1α (10 ng/ml) for the last 60 min. Cell cultures expressing HA-miniTurbo only (empty vector, EV) or receiving only doxycycline or biotin served as negative controls (indicated by gray font). Parental HeLa cells (p) were included as further controls. Left panel: Cells were lysed and proteins were analyzed by Western blotting for the expression of p65-HA-miniTurbo and HA-miniTurbo using anti-p65 and anti-HA antibodies. Equal loading was confirmed by probing the blots with anti β-actin antibodies. Right panel: Biotinylated proteins from the same samples were purified on streptavidin-agarose beads and biotinylation patterns were visualized by Western blotting using streptavidin-horseradish peroxidase (HRP) conjugates (representative images from two independent experiments). (E) Biotinylated proteins from the experiment shown in (C) and from a second biological replicate were identified by mass spectrometry. Volcano plots show the ratio distributions of $Log_2$-transformed mean protein intensity values on the $X$ axes obtained with wild-type p65 or the p65 mutants compared to the empty vector controls in the presence or absence of IL-1α treatment. $Y$ axes show corresponding p values from Student's $t$ test results. Strong enrichment of the bait p65/RELA proteins together with the core canonical NF-κB components is shown in red and blue colors, respectively (two biologically independent experiments and three technical replicates per sample). (F) Specific proteins binding to p65/RELA wild-type were defined by significant enrichment ($Log_2$ fold change (LFC) ≥ 2, $-log_{10} P ≥ 1.3$, Student's $t$ test) compared to HA-miniTurbo only and to cells exposed to doxycycline or biotin only (see Fig. EV2). This set of proteins was intersected with proteins enriched in cells expressing p65 mutant proteins (LFC ≥ 2, −log10 P ≥ 1.3). Venn diagrams show the numbers of p65/RELA interactors and their overlaps before and after IL-1α-treatment, with values in the lower left corners indicating total numbers of interactors. (G) The six protein sets shown in (E) were subjected to parallel overrepresentation pathway analysis using Metascape software (Zhou et al, 2019). The Venn diagrams show the overlap of the top 100 enriched pathway terms. For IL-1α samples, only 92 terms were enriched. Values in the lower left corners indicate the total number of unique pathways. (H) The table shows the most strongly enriched pathway categories associated with the p65 /RELA wild-type or mutant interactomes. Numbers in brackets indicate the total numbers of p65/RELA interactors per condition according to (E, F). Enrichment P values for overrepresentation analyses were computed by Metascape software (Zhou et al, 2019). rtTA reverse tetracycline-controlled transactivator. The mass spectrometry data and bioinformatics analysis results are provided in Dataset EV1. Source data are available online for this figure.

---

biological replicates (Dataset EV1). Volcano plot analyses show that many of these proteins are labeled by the small and presumably more mobile HA-miniTurbo protein (Fig. 1E, samples labeled EV). In contrast, p65/RELA was highly enriched along with its canonical interaction partners p50 (NFKB1), p52 (NFKB2), IκBα (NFKBIA), and NEMO (IKBKG) in samples expressing the fusion proteins (Fig. 1E). Based on a significant at least fourfold enrichment compared with all negative controls, we found 279 specific p65/RELA interactors in untreated cells and 310 in IL-1α-treated cells (Figs. 1E and EV2A,B). With the E/I mutant, 251 interactors were identified in comparison, compared with only 176 after IL-1 treatment (Fig. 1E). Striking was the significantly reduced number of p65/RELA interactors in the FL/DD mutant, which amounted to 95 in untreated cells and only 31 after IL-1 treatment (Fig. 1F). Since, as shown in Fig. 1D, a comparable enrichment and thus (auto)biotinylation of the RELA (FL/DD) fusion protein was measured, we consider this effect to be specific. Of 401 specific interactors in untreated cells, only 16 (4%) were associated with all p65/RELA bait proteins, and these numbers (13 or 4%) were similar after IL-1α treatment (Fig. 1F).

To investigate the extent to which these differences were also reflected at the functional level, comparative overrepresentation analyses were performed for all six protein groups shown in Fig. 1G to identify the 100 most enriched pathways. These data show that the p65/RELA wild-type and the E/I mutant, but not the FL/DD mutant, behave largely similarly in terms of the biological function of the interacting proteins (Fig. 1G), whereas the FL/DD mutant is essentially associated with a loss of pathway terms (Fig. 1G). This is particularly evident when considering the 15 most enriched pathways, which include processes such as chromatin organization, histone modifications, transcription, NF-κB signal transduction, and various developmental and differentiation steps (Fig. 1H).

The aggregated 366 interactors from untreated and IL-1α-stimulated conditions mapped to 330 proteins in the STRING

database that had 2479 one-way protein-protein interactions (PPI) (Fig. 2A,B).

In this set, we found 46 proteins for which an interaction with p65/RELA had already been documented in STRING at different experimental levels of evidence (Fig. 2B).

The 46 p65/RELA interactors shared 318 PPI (Fig. 2B). They further interacted with 881 redundant additional interactors, of which 199 were unique and overlapped with the STRING input list of 330 proteins, suggesting that our miniTurboID approach also captured the indirect recruitment of factors into the p65/RELA interactome (Fig. 2C).

The cytokine-regulated part of the p65/RELA interactome shown in Fig. 2A might result from phosphorylation of p65/RELA or its interaction partners. Immunoblot analyses demonstrated weakly regulated phosphorylation of p65-HA-mTb at S468, S529, and S536 but not S276 (Fig. EV3A,B). Re-analysis of the mass spectra from the proteomic interactome data revealed low-level phosphorylation of p65-HA-mTb at five additional sites in the N-terminus (Fig. EV3C). The same approach identified 14 additional phosphorylated amino acid residues that were consistently found in 13 unknown and known RELA interaction partners, including c-JUN, FOSL1/2, KMT2D and NCOR2 (Fig. EV4A). Phosphorylation of S63 in the N-terminal transactivation domain of c-JUN was normally induced by IL-1α in cells reconstituted with either p65-HA-mTb or HA-mTb (Fig. EV4B). The Phosphopeptide containing this site was specifically enriched in the p65-HA-mTb interactome (Fig. EV4A), whereas comparable enrichment of the c-JUN protein by miniTurboID was observed before and after IL-1α treatment (Fig. EV4C). c-JUN was predicted by STRING to interact with approximately 50 proteins that are also part of the p65-HA-mTb interactome, as shown by Volcano plots (Fig. EV4C). These indirect interactors segregated into three subnetworks in an IL-1α-dependent manner (Fig. EV4D). Similar, albeit smaller subnetworks were obtained for FOSL2, KMT2D and

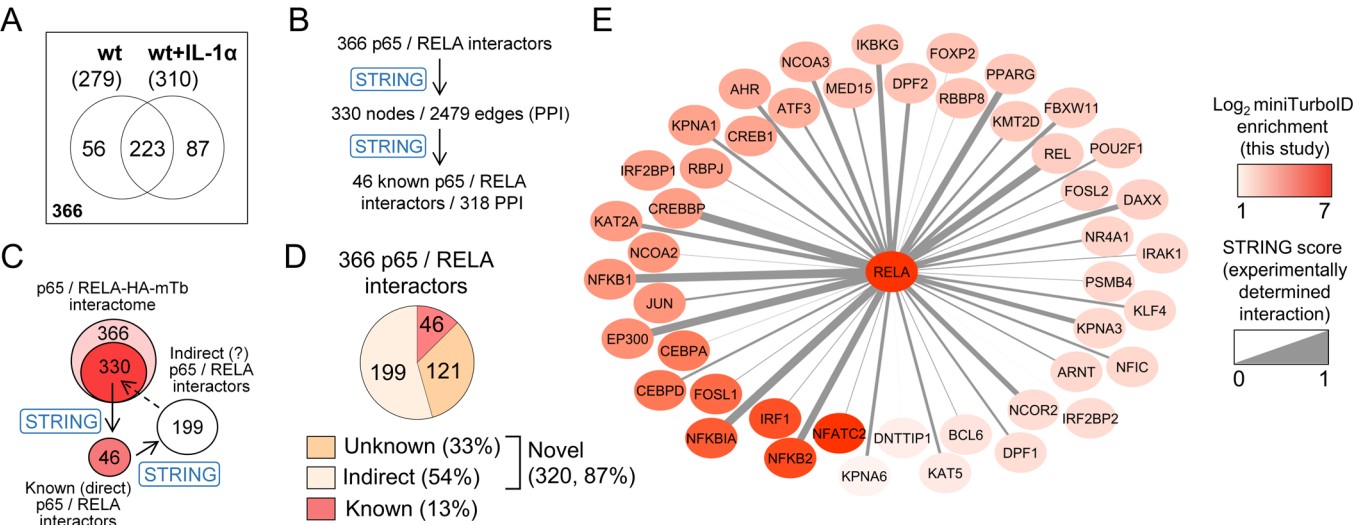

**Figure 2. Protein composition of the p65/RELA interactome.**

(A) Venn diagram of p65/RELA interactors in IL-1α or untreated cells revealing the principal set of 366 unique p65/RELA interactors that were further investigated in this study. (B) Scheme showing the results of mapping the 366 interactors from (A) to the STRING database of protein-protein interactions (PPI) (Szklarczyk et al, 2019). (C) Scheme visualizing a set of 199 proteins enriched in the experimentally determined p65/RELA proximity interactome that have documented PPIs with the 46 known p65/RELA interactors in STRING. (D) Classification of the 366 p65/RELA interactors. (E) Protein interaction network of the 46 known p65/RELA interactors found by miniTurboID. Edge widths visualize the evidence for experimental interactions deposited in the STRING database. Nodes are colored in red and are arranged according to the enrichment found by proximity labeling in our study. Source data are available online for this figure.

NCOR2 (Fig. EV4E). These data provide exemplary evidence to suggest that parts of the (inducible) p65-HA-mTb interactome are remodeled by post-translational modifications mediated by IL-1α-activated protein kinase pathways.

Overall, the results demonstrate that the proximity-based p65/RELA interactome not only corroborates the existence of numerous p65/RELA interactors, but also identifies 87% of novel interactors, more than half of which already have annotated interactions amongst themselves (Fig. 2D).

For the most part, the 46 known p65/RELA interactors were strongly enriched by miniTurboID and included many well-characterized transcription factors (e.g., NFATC2, IRF1, FOSL1, CEBPa/d, JUN), histone-modifying enzymes (e.g., EP300, CREBBP, KAT2A), chromatin remodelers (e.g., DPF1/2, NCOR2), nuclear cofactors (e.g., MED15, BCL6) and signaling factors (e.g., IRAK1) (Fig. 2E).

Next, particularly in light of the many molecular functions in gene regulation attributed to p65/RELA (Martin et al, 2020), we focused on a detailed analysis of the composition of the p65/RELA interactome, primarily concerning proteins with a role in chromatin-associated processes or transcription.

In terms of molecular functions, the 366 p65/RELA interactors shown in Fig. 2A are almost exclusively associated with RNA polymerase II-regulated transcription processes (Fig. 3A, left panel) and are localized largely in the nucleus, in membrane-less organelles, and in chromatin (Fig. 3A, right panel).

When compared to the list of 1639 TFs documented in the human genome by Lambert et al (Lambert et al, 2018), 172 (47%) of all p65/RELA interactors are classified as DBD-containing TF proteins (Fig. 3B). Based to 801 epigenetic regulators contained in the newest version of the Epifactors database (Marakulina et al, 2023), a further

74 (20%) of all p65/RELA interactors are chromatin writers, readers, erasers or remodelers (Fig. 3B).

Of the 172 TFs, 137 were enriched under basal and 150 under IL-1α-stimulated conditions, with most factors (115) overlapping (Fig. 3C).

In total, 117 of the 172 TFs were distributed among 7 TF classes, with C2H2 ZF and homeodomain TFs being the most abundant and accounting for 40% of all p65/RELA interactors (Fig. 3D). bZIP TFs were the third most abundant and contained a number of already known p65/RELA interactors such as JUN, ATF2, and FOSL1/2, among others (Figs. 2E and 3D). The remaining 55 TFs represented 31 different TF classes (Fig. 3D). Overall, we found that in IL-1α stimulated cells, 13 ZBTB and 12 ZNF transcription factors, both from the C2H2 ZF class, were the most frequently identified p65/RELA interactors among all enriched TF families (Fig. 3E).

In untreated cells, 91 (66%) of 137 TFs were still present with the E/I mutant, whereas in IL-1α-stimulated cells this was the case for only 65 TFs out of 150 (43%) (Fig. 3F). Under both conditions, only 4 TFs (3%) were enriched with the FL/DD mutant (Fig. 3F). These data indicate that only about 35% to 55% of the TF interactions of p65/RELA require DNA binding, and this dependency increases during cytokine activation, whereas virtually all TF interactions require dimerization.

The interaction of p65/RELA with nuclear cofactors was completely abolished by the FL/DD (but not the EI) mutant in almost every case, as shown in Fig. 3G using the top-10 strongest p65/RELA interactors as an example. Based on annotations in the Epifactors database, 50 of the epigenetic regulators are subunits of a total of 19 established nuclear multiprotein complexes, with components of the BAF complex being most abundant in the

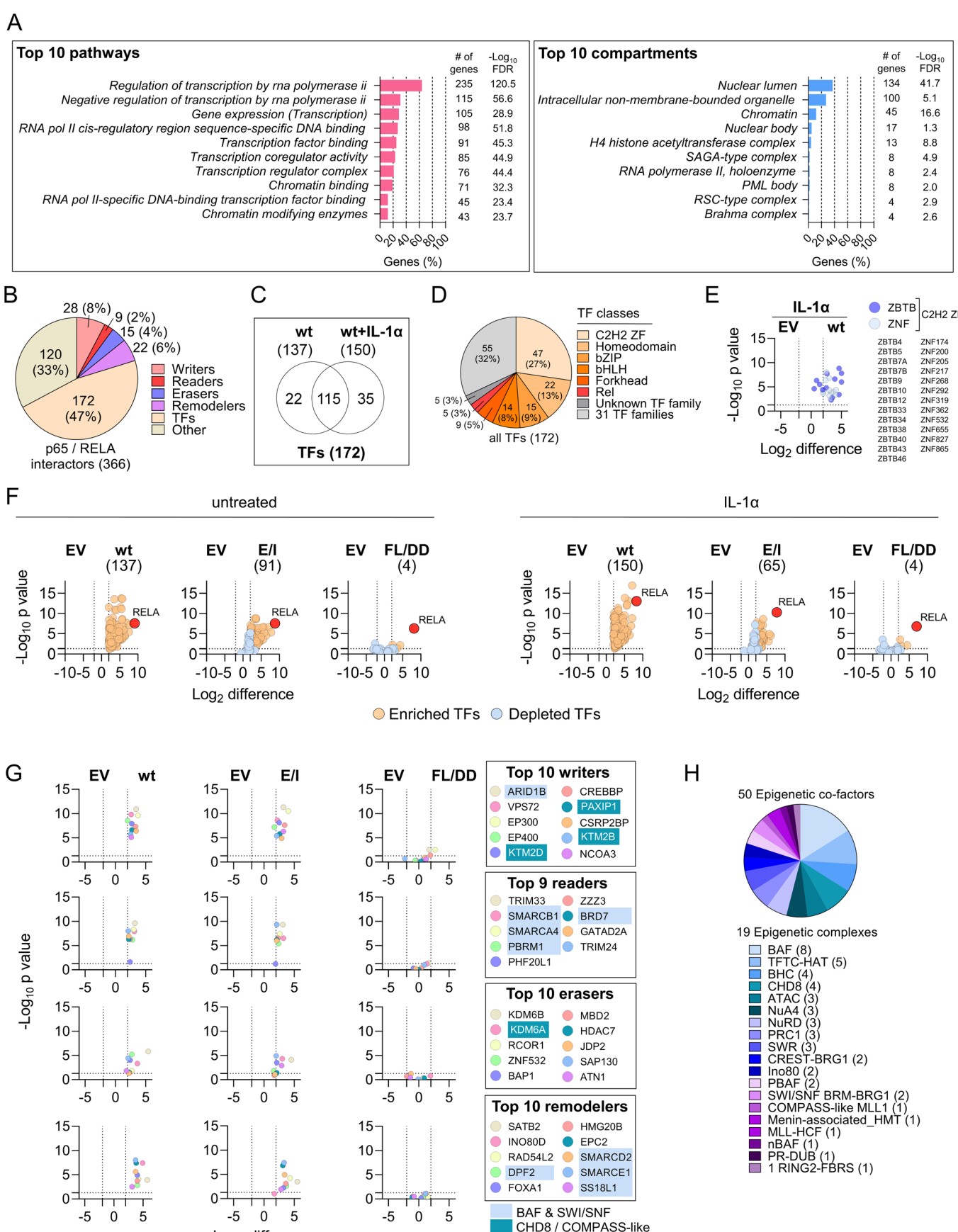

**Figure 3.  p65/RELA interactions with transcription factors and epigenetic regulators.**

(A) The top-10 pathway terms according to GO (BP, CC, MF), KEGG, Reactome, STRING clusters and WikiPathways database entries and the top-10 subcellular localizations associated with the 366 p65/RELA interactors. Annotations, number of components and false discovery rates (FDR) were retrieved using the STRING plugin of Cytoscape (Shannon et al, 2003). (B) Overlap of the RELA interactome with 1639 human TFs (Lambert et al, 2018) and 801 epigenetic regulators (Marakulina et al, 2023). (C) Venn diagram showing the overlap of enriched TFs in basal or IL-1α-stimulated conditions. (D) Distribution of TF families found to be associated with p65/RELA in basal and IL-1α-stimulated conditions according to the annotation provided by (Lambert et al, 2018). (E) IL-1α-dependent enrichment of all TF belonging to ZBTB and ZNF families as identified by miniTurboID. (F) Volcano plots visualizing all TFs significantly enriched with wt p65/RELA (LFC ≥ 2, −log$_{10}$ P ≥ 1.3) compared with empty vector control (EV) and the changes obtained with p65 mutants in basal or IL-1α-stimulated conditions. (G) Graphs visualizing the top-10 enriched epigenetic regulators. Volcano plots show the ratio distributions of Log$_2$ transformed mean protein intensity values obtained with wild-type p65/RELA (wt) or with p65/RELA mutants (FL/DD, E/I) compared to empty vector controls (EV). Only 9 reader proteins were found. (H) Association of enriched epigenetic regulators with known epigenetic complexes according to the annotation provided by (Marakulina et al, 2023). Numbers in brackets show identified components per complex. (E–G) P values were calculated by Student's t test. Source data are available online for this figure.

p65/RELA interactome (Fig. 3H). Particularly prominent were four to eight subunits each of the BAF, TFC-HAT, BHC, and CHD8 complexes. A total of nine factors were assigned to the canonical BRG/BRM-associated BAF (BAF or cBAF), polybromo-associated BAF (PBAF), non-canonical BAF (ncBAF) and mammalian SWI/SNF (short for SWItch/sucrose nonfermentable) complexes (Fig. 3H, highlighted in light blue). Four factors belonged to the CHD8 or COMAPASS (short for proteins associated with Set1C) complexes (Fig. 3H, highlighted in turquoise).

Taken together, these data demonstrate that proximity labeling reveals a much larger p65/RELA interactome and its dynamics than previously appreciated. Through this approach, we find a complex DNA-binding, dimerization-, and IL-1α-dependent remodeling of the p65/RELA interactome, with mutation of only two dimerization-related amino acids in RHD exerting the strongest influence, consistent with the interpretation that most p65/RELA interactions with other cellular proteins do not require direct or stable interactions with DNA but rely primarily on intact dimerization functions. Half of the p65/RELA interactome is dominated by a large number of TFs distributed across many different classes, of which about up to 50% also require an intact RELA DBD in addition to dimerization. The second largest group, besides TFs, is represented by components from protein complexes that affect chromatin modifications and remodeling and whose interaction with p65/RELA appears to be largely independent of DNA binding. Overall, based on bioinformatic analyses, 87% of the p65/RELA interactors are novel, defining a previously unknown dimension of the extensive interaction of p65/RELA with other transcriptional regulators and cofactors.

## Functional validation of 38 p65/RELA high-confidence interactors by a targeted siRNA screen

Based on greater than eightfold enrichment in both replicates in untreated or IL-1α-stimulated cells and lack of published evidence for clearly defined functions in the NF-κB system, we finally extracted a list of 38 "high confidence" p65/RELA interactors (HCI) (Fig. 4A).

As before, these proteins were highly associated with transcriptional functions (Fig. 4B). Because they had almost no known interactions with each other and only two had a documented interaction with p65/RELA (CEBPD and FOSL1) in the STRING database, this set defines a new, previously unknown part of the p65/RELA interactome that we selected for follow-up validation (Fig. 4B).

We then performed a targeted siRNA screen to investigate all 38 HCI concerning their relevance for NF-κB-mediated gene

expression. Specifically, we used commercial siRNAs to suppress all 38 factors individually in untreated and in IL-1α-stimulated cells and then assessed the effects on the expression levels of endogenous NF-κB target genes. The screen was performed in a miniaturized format, by which RT-qPCRs were performed in total cell lysates with an intermediate linear pre-amplification PCR step, using gene-specific primers/Taqman probes for all of the targeted genes (38 plus p65/RELA), three IL-1α-inducible genes (CXCL8, NFKBIA, CXCL2) and two housekeeping genes (GUSB, GAPDH) (Appendix Fig. S2A).

Results from three independent screens were normalized and expressed as mean difference between the siRNA target and a control siRNA-directed against luciferase (Luc). The mRNA levels of 38 HCI were successfully suppressed by the siRNAs, while there was little effect on GUSB and GAPDH (Appendix Fig. S2B; Dataset EV2).

As expected, the strongest suppression of IL-1α-inducible expression of CXCL8, NFKBIA, and CXCL2 was observed with p65/RELA knockdown (Fig. 4C). However, we found that for all tested genes and conditions the knockdown of a single HCI affected at least one NF-κB target gene in basal or IL-1α-stimulated conditions, or both (Fig. 4C).

Hierarchical clustering of the expression patterns revealed four clusters (3–6) of HCI whose suppression resulted in relatively uniform and strong suppression of IL-1α target genes, while cluster 2 comprised a set of genes whose suppression had a strong effect on the basal expression of CXCL8 and CXCL2 (Fig. 4C).

While p65/RELA seemed to be essential, these data showed a functional contribution of all 38 top HCI to NF-κB-dependent gene expression and suggest that each interacting protein in a (gene-) specific manner shapes the basal and inducible state of p65/RELA-dependent genes.

To further investigate the contribution of HCI to the regulatory functions of p65/RELA, we selected six transcription factors for more detailed and genome-wide follow-up studies, namely zinc finger and BTB domain containing 5 (ZBTB5, encoded by KIAA0354), Zinc finger protein GLI-similar 2 (GLIS2, also called Neuronal Krueppel-like protein, NKL), S100A8 (also called CAGA, CFAG, MRP8), and S100A9 (also called CAGB, CFAG, MRP14), and transcription factors E3 (TFE3, BHLHE33) and EB (TFEB, BHLHE35) (Fig. 4C). All six proteins clearly interacted with p65/RELA in the miniTurboID screen (Fig. 4D). The interactions of TFE3, TFEB, GLIS2, and ZBTB5 with p65/RELA were confirmed for the endogenous proteins at the single cell level by proximity ligation assays (PLA) and were significantly reduced in p65-deficient cells (Fig. EV5). By PLA, we also found that TFE3, but not

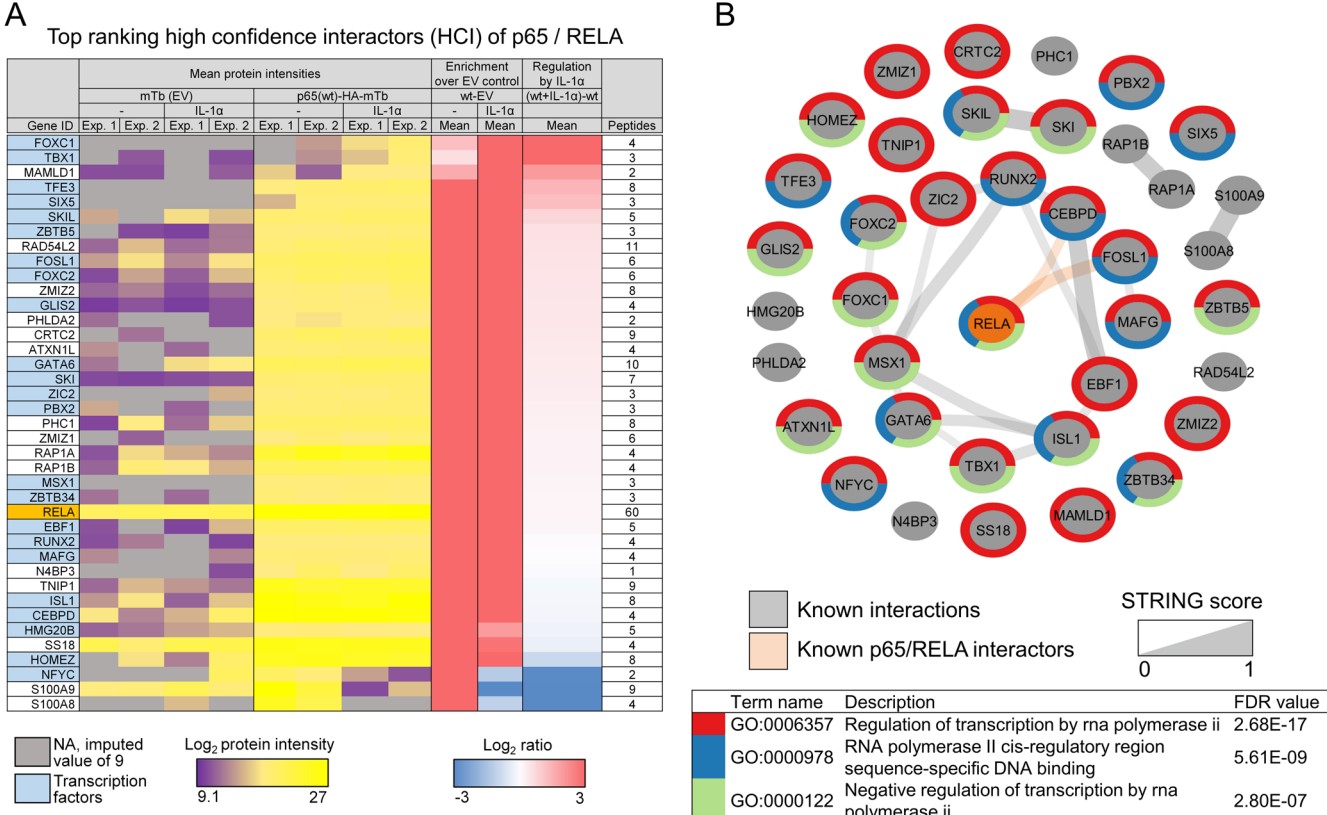

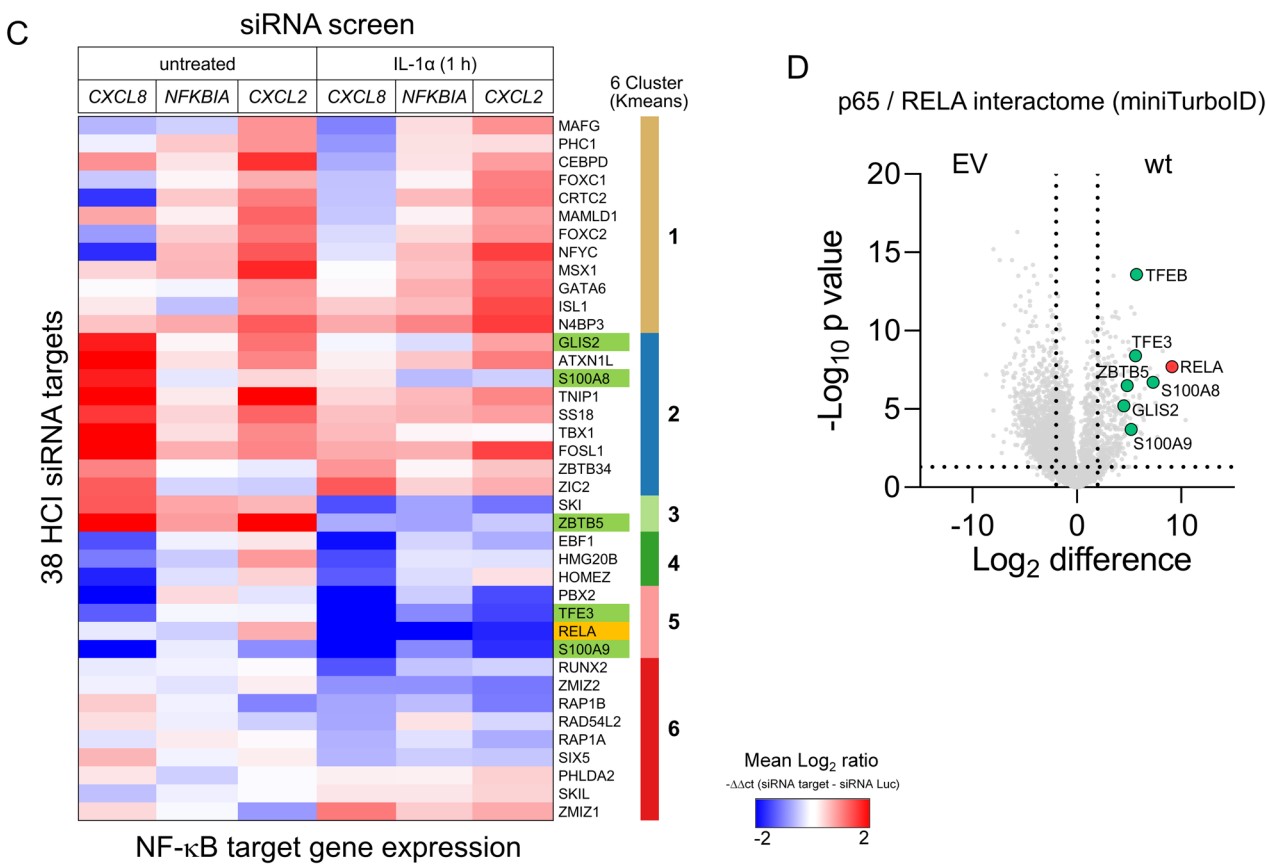

**Figure 4. A targeted siRNA screen of 38 novel high-confidence interactors (HCI) of p65/RELA shows their function in the regulation of prototypical NF-κB target genes.**

(A) Final list of top ranking high-confidence interactors of p65/RELA selected for further studies. The heatmap shows the Log$_2$ transformed mean protein intensity values from technical triplicates of the two biological independent miniTurboID experiments, the enrichment ratio values compared to the empty vector (HA-miniTurbo) control (EV) and the regulation by IL-1α. With the exception of N4BP3, all proteins were identified by at least two peptides. (B) Graph showing that the top 38 p65/RELA interactors are largely devoid of known protein interactions based on STRING entries. According to STRING, only two factors (CEBPD and FOSL1) interact with p65/RELA. Node borders visualize the main functional annotations. (C) HeLa cells were transiently transfected for 48 h with 20 nM of siRNAs mixtures for 38 HCI and p65/RELA, a siRNA targeting luciferase, transfection reagent alone or were left untreated. Half of the cells per plate were treated for 1 h with IL-1α (10 ng/ml) at the end of the incubation. cDNAs were transcribed in lysates and amplicons for three NF-κB target genes, two housekeeping genes and all 38 HCI p65/RELA interactors were pre-amplified by linear PCR and then quantified by qPCR. Based on Ct values, mRNA levels were quantified and normalized against *GUSB*. The effects of knockdowns were calculated separately for basal and IL-1α-inducible conditions against the luciferase siRNA. The heatmap shows hierarchically Kmeans clustered mean ratio values derived from three biologically independent siRNA screens. As a positive control, RELA knockdowns were performed in parallel. Green colors highlight p65/RELA interactors selected for further analysis. (D) The miniTurboID enrichment of six p65/RELA interactors (green colors) chosen from (C) is shown. *P* values were calculated by Student's *t* test. Source data for the siRNA screen are provided in Dataset EV2. Source data are available online for this figure.

TFEB bound to p50 NF-κB in a p65/RELA-dependent manner, providing examples for factors with differential preferences for p65/RELA homo- compared with heterodimers (Appendix Fig. S3A). With the antibodies available to us, we also confirmed the interaction of small amounts of TFE3 with the endogenous p65/p50 heterodimer by conventional co-immunoprecipitation (Appendix Fig. S3B).

This selection of factors provided an opportunity to validate the p65 interactors using the example of two poorly characterized transcription factors with completely unknown relationships to p65/RELA (ZBTB5, GLIS2) and two pairs of related factors (S100A8/A9, TFE3/TFEB) that play a role in inflammation but do not have a well-established mechanistic link to the NF-κB system.

## Crosstalk of lysosomal transcription factors TFE3/TFEB and GLIS2 with the (inducible) NF-κB system

The miniTurboID data showed that three out of four MiT-TFE family members, i.e., TFE3, TFEB and microphthalmia-associated transcription factor (MITF) and all three GLIS family members (GLIS1-3) bound to p65/RELA wt (Fig. 5A) This interaction was largely abolished in the FL/DD dimerization-deficient p65/RELA mutant for all six factors and reduced in the E/I DNA-binding deficient mutant mainly after cytokine stimulation (Fig. 5A).

TFE3 and TFEB proteins appeared as multiple bands in cell extracts in the absence of nutritional stress and their dephosphorylated, faster-migrating forms rapidly accumulated in the nuclear fraction upon starvation as described before (Fig. 5B; Appendix Fig. S4) (Martina et al, 2016; Martina et al, 2014; Settembre et al, 2011). Under non-starved conditions, considerable amounts of both phosphorylated TFE3 and TFEB were present in the nucleus (Fig. 5B; Appendix Fig. S4). IL-1α treatment did not change the phosphorylation patterns and the subcellular distributions of TFE3/TFEB, but caused transient nuclear translocation of p65 and p50 between 0.5 h and 1 h as expected (Fig. 5B; Appendix Fig. S4).

Silencing of p65/RELA, TFE3/TFEB, and GLIS2 by siRNA revealed their profound suppression at the protein levels (Fig. 5C). We also observed that p65/RELA knockdown reduced TFE3 levels, TFE3 knockdown reduced TFEB and GLIS2 levels, and GLIS2 knockdown reduced p65/RELA and TFE3 levels, suggesting a mutual regulation of the three factors at the protein level (Fig. 5D).

RT-qPCR analysis of six prototypical TFE3/TFEB autophagy and lysosomal target genes with a conserved cis-element in the regulatory region, the so-called Coordinated Lysosomal Expression and Regulation (CLEAR) element (Martina et al, 2016; Martina et al, 2014; Sardiello et al, 2009), revealed no effect of silencing p65/RELA on mRNA expression of any of these genes (Appendix Fig. S5). These CLEAR genes were also not regulated by IL-1α (Appendix Fig. S5).

In contrast, knockdown of TFE3 or TFEB strongly suppressed the IL-1α-mediated expression of five prototypical inflammatory IL-1α target genes (*IL8/CXCL8*, *CSF2*, *CCL2*, *TNFAIP3*, *NFKBIA*), while GLIS2 knockdown suppressed the IL-1α-mediated expression of *CSF2*, *CCL2*, and *NFKBIA* (Fig. 5E).

These data provided additional functional validation of three of the p65/RELA interactors and suggested a unidirectional crosstalk of lysosomal transcription factors TFE3/TFEB with the IL-1α–NF-κB system.

## ZBTB5, GLIS2, S100A8/A9, and TFE3/TFEB are co-regulators of the p65/RELA gene response

We then assessed the genome-wide roles of the six selected factors. In a first series of transcriptome analyses using 48,000-probe microarrays, we compared silencing of p65/RELA with silencing of ZBTB5 and S100A8/A9, whereas in a second series we compared silencing of p65/RELA with silencing of GLIS2, TFE3, and TFEB. Differentially expressed genes (DEGs) in the seven knockdowns were defined based on a Log$_2$ fold change (LFC) ≥ 1 with a −Log$_{10}$ *P* value ≥ 1.3 compared to luciferase siRNA transfections.

First, we addressed the question of the extent to which the six factors are involved in (co)regulation of the IL-1α-regulated NF-κB response (Fig. 6A; Dataset EV3).

Of the 756 (series 1) and 617 (series 2) IL-1α-induced genes, 230 (30%) and 168 (27%) genes, respectively, were expressed in a p65/RELA-dependent manner (Fig. 6B; Dataset EV3).

Each individual knockdown affected a comparable number of IL-1α target genes, which partially overlapped with the p65/RELA-regulated sets of genes as indicated by red colors in the Venn diagrams shown in Fig. 6B.

Similar to RELA, suppression of ZBTB5, GLIS2, S100A8/A9, and TFE3/TFEB overall resulted in a significant reduction in the mean expression levels of their respective sets of IL-1α target genes, consistent with them acting primarily as coactivators in the IL-1 system (Fig. 6C; Dataset EV3).

Correlation analyses of the effects of p65/RELA knockdowns with the respective knockdown of a p65/RELA interactor showed a

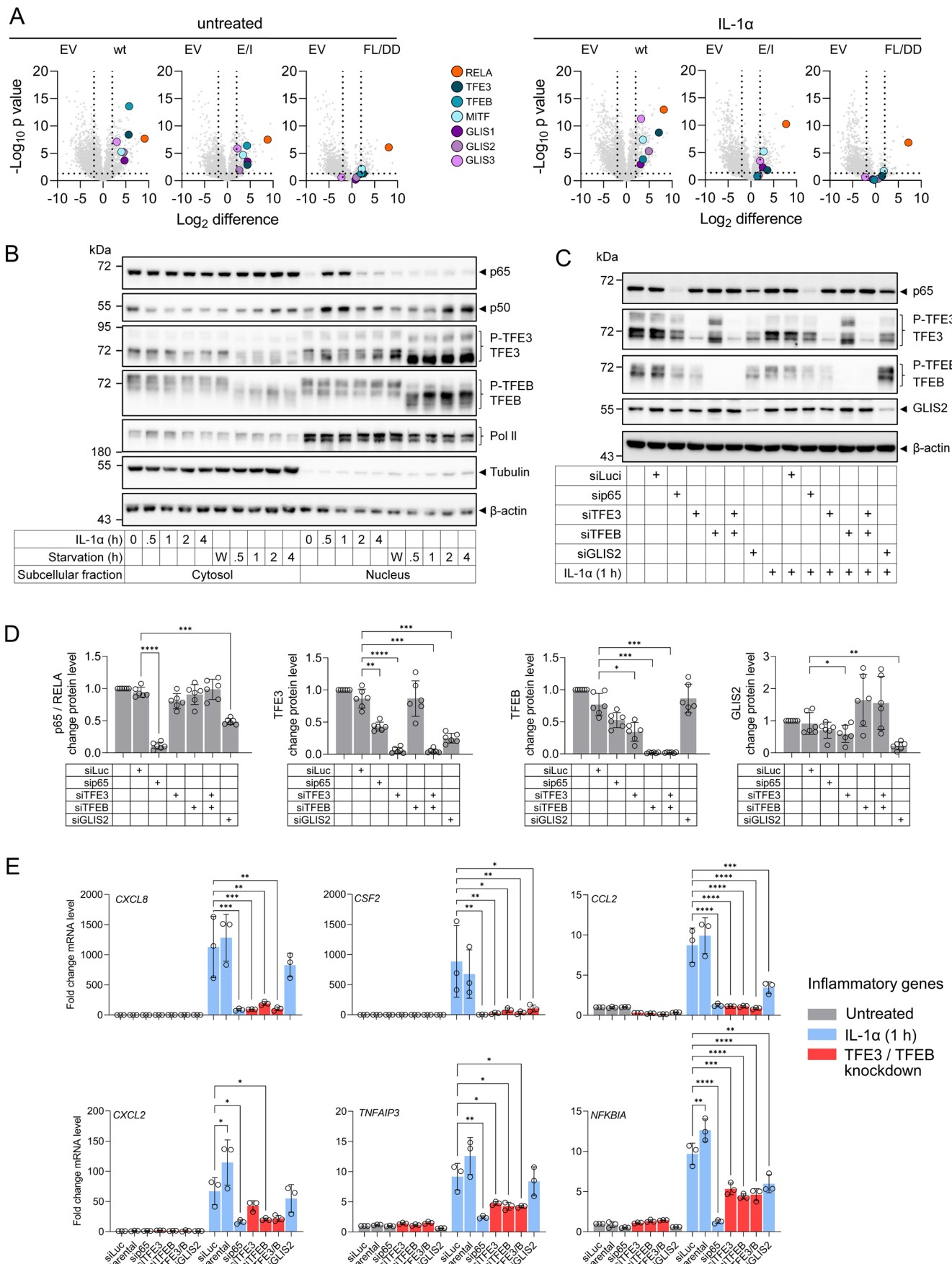

**Figure 5. Interdependent regulation of NF-κB target genes by TFE3, TFEB, and GLIS2.**

(A) Volcano plots revealing the basal and IL-1α-dependent enrichment of all MiT/TFE and GLIS family members in the miniTurboID experiments. *P* values were calculated by Student's *t* test. For details, see Fig. 1D. (B) The subcellular distribution of phosphorylated (P) and dephosphorylated forms of TFE3 and TFEB, p50, and p65 NF-κB was evaluated by Western blotting in cell extracts from HeLa cells stimulated with IL-1α or subjected to starvation (HBSS) for the indicated times. "W" indicates samples washed four times with HBSS and then supplemented with their previous cell culture medium to control for effects caused by the washing procedure prior to addition of starvation medium. Antibodies against RNA polymerase (pol II), tubulin or β-actin served as control for fractionation and equal protein loading. Shown is one out of three biologically independent experiments. See Appendix Fig. S4 for the quantification of replicates. (C) Parental HeLa cells or cells transfected with siRNAs (20 nM) against TFs or luciferase (as a negative control) were cultivated for 48 h. Then, half of the cells were stimulated for 1 h with IL-1α (10 ng/ml) or were left untreated. Total cell extracts were examined for the expression of the indicated proteins by Western blotting. Antibodies against β-actin served as loading controls. Shown is one out of three biologically independent experiments. (D) Quantification of the basal expression levels of the indicated TFs in extracts of cells transfected as in (C). Bar graphs show data points and mean values ± s.d. relative to parental cells from six biologically independent experiments. Asterisks indicate *P* values (*$P \le 0.05$, **$P \le 0.01$, ***$P \le 0.001$, ****$P \le 0.0001$) obtained by one-way ANOVA. (E) Total RNA isolated from cells treated as in (C) was analyzed for mRNA expression of the indicated NF-κB target genes by RT-qPCR. Bar graphs show data points and mean values ± s.d. relative to cells transfected with luciferase siRNA from three biologically independent experiments. siTFE3/B indicates double knockdown of TFE3 and TFEB. Asterisks indicate *P* values (*$P \le 0.05$, **$P \le 0.01$, ***$P \le 0.001$, ****$P \le 0.0001$) obtained by one-way ANOVA. Source data are available online for this figure.

pronounced coregulation of jointly regulated IL-1α target genes, which are represented by the red sets of genes in Fig. 6B,D, into the same direction. This means, if a gene was suppressed or induced with the p65/RELA siRNA relative to the luciferase siRNA, this was also the case with the knockdown of the respective interactor (Fig. 6D).

In basal conditions, the knockdowns of ZBTB5, GLIS2, S100A8/A9, and TFE3/TFEB affected the expression of more than 1000 genes, the majority of which did not overlap with the likewise more than 1000 p65/RELA target genes. These data illustrate broad role and distinct roles of RELA and its HCI in homeostatic cell functions (Appendix Fig. S6A,B; Dataset EV3).

Unlike for the IL-1α-regulated gene subsets, the distribution of the mean expression levels of the siRNA target gene sets did not reveal any global coactivator or corepressor effects (Appendix Fig. S6C; Dataset EV3).

However, the relatively small sets of genes specifically overlapping with p65/RELA targets in basal gene expression were coregulated into the same direction (Appendix Fig. S6D).

These data emphasize the broader relevance of the new p65/RELA interactors at the functional level, by showing that all representatively selected factors affect specific sets of RELA target genes in homeostatic conditions and profoundly participate in the regulation of inducible subsets of the IL-1α gene response.

## p65/RELA, ZBTB5, GLIS2, S100A8/A9, and TFE3/TFEB engage in complex multilayer gene regulatory networks

To identify additional functional connections between all of the groups of genes deregulated by siRNA knockdowns in the IL-1α response as shown in Fig. 6B, we used their STRING entries of functional protein–protein interactions to construct multidimensional interaction networks.

The basic idea of this analysis is that the gene products regulated by RELA or a RELA interactor may themselves have many other direct or functional protein interactions, providing the cell with a much larger and ultimately interconnected gene regulatory network as shown schematically in Fig. 7A.

To test this hypothesis, we used the gene sets defined via Venn diagrams in Fig. 6B and constructed four separate PPI networks for the ZBTB5/RELA (325 genes), S100A8/S100A9/RELA (405 genes), GLIS2/RELA (243 genes), and TFE3/TFEB/RELA (318 genes) groups. Noteworthy, only 69–77% of the genes affected by siRNA

knockdown had a PPI entry in STRING, corroborating the notion that the combined miniTurboID/transcriptome analysis effectively revealed many new components of novel genetic networks (Fig. 7B).

The known interactions of factors retrieved from STRING are shown by gray lines, whereby edge width corresponds to the underlying evidence (Fig. 7C). As highlighted by the light pink colors of the edges, RELA interacts with a relatively small number of proteins in all four networks. S100A8/9 and TFE3/TFEB strongly interacted with each other (as expected from the literature (Raben and Puertollano, 2016; Vogl et al, 2018)) and with very few other proteins, whereas ZBTB5 and GLIS2 had no known interactions at all.

The coloring of network nodes according to their dependence on RELA, ZBTB5, GLIS2, S100A8/9, or TFE3/TFEB links the known levels of connectivity deposited in STRINGC to the novel patterns of regulation of the corresponding genes observed in our study and reveals a multitude of experimentally determined new relations between the different groups of siRNA target genes (Fig. 7C).

Taken together, this refined analysis, using the IL-1 response as an example, demonstrates that p65/RELA controls large genetic networks composed of gene regulatory subnetworks which are assembled from specific interactions of p65/RELA with ZBTB5, GLIS2, S100A8/9, and TFE3/TFEB and their target genes.

## Chromatin recruitment of RELA and its interactors

We next extended the functional analysis of selected p65/RELA interactors to the chromatin level. For this purpose, we used our published p65/RELA ChIPseq data from the HeLa subclone KB and screened the experimentally determined binding regions of p65/RELA for underlying significantly enriched DNA motifs of TFE3, TFEB or GLIS2 within a range of ±500 base pairs around the p65/RELA peaks (Fig. 8A; Dataset EV4) (Jurida et al, 2015; Weiterer et al, 2020).

First, we found that 42% of all 35,024 p65/RELA peaks were associated with motifs specific for either p65/RELA or for any REL (NF-κB) transcription factor (Fig. 8B). Second, 2%, 7%, or 12% of all 35,024 p65/RELA ChIPseq peaks contained a RELA motif and a TFEB, TFE3, or GLIS2 motif, respectively, suggesting that RELA chromatin recruitment occurs at composite DNA-binding elements which would facilitate direct binding of RELA and either TFE3,

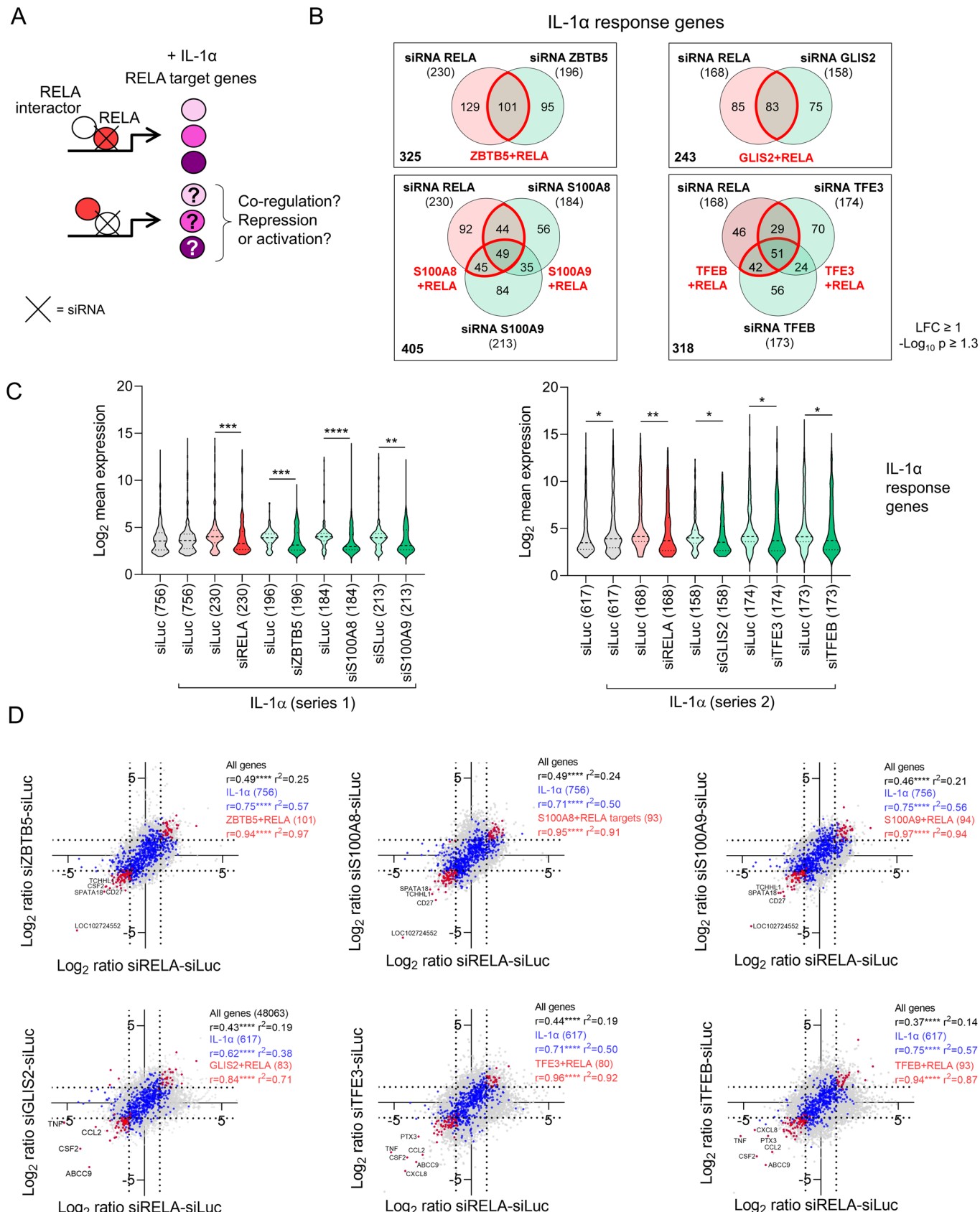

**Figure 6.  ZBTB5, GLIS2, S100A8/S100A9 and TFE3/TFEB co-regulate IL-1α-inducible subsets of RELA target genes.**

(A) Schematic illustrating the strategy to analyze the influences of novel p65/RELA interactors on IL-1α-regulated p65/RELA target genes by combining siRNA-mediated knockdown with transcriptome analysis. (B) HeLa cells were transiently transfected for 48 h with 20 nM siRNA mixtures against RELA, ZBTB5, S100A8, S100A9 (series 1) or RELA, GLIS2, TFE3, TFEB (series 2) and an siRNA against luciferase (siLuc) as control. Half of the cells were treated with IL-1α (10 ng/ml) for 1 h at the end of incubation, and Agilent microarray analyses were performed from total RNA. Normalized data were used to identify DEGs based on an LFC ≥ 1 with a $-\log_{10} P$ value ≥ 1.3 (moderated $t$ test). Venn diagrams show the overlap of all DEGs that were affected at least twofold by siRNA knockdown in IL-1α-treated samples, with the ratio of siLuc to individual knockdown determined in each case. Red colors mark genes jointly regulated by knockdown of RELA and one of its interactors (two biologically independent experiments). (C) Violin plots show the distribution, medians, and interquartile ranges of normalized expression levels for all IL-1α-regulated genes and the corresponding changes in the gene subsets defined in Fig. 6B that were affected by siRNA knockdown. The number of these genes is indicated in parentheses. Asterisks indicate significant changes as determined by a two-tailed Mann–Whitney test (*$P \leq 0.05$, **$P \leq 0.01$, ***$P \leq 0.001$, ****$P \leq 0.0001$). (D) Superimposed pairwise correlation analyses of the mean ratio changes of all genes (gray), IL-1α-regulated genes (blue), and gene sets significantly up- or downregulated by siRNA knockdown (red). Ratio values from RELA knockdown conditions were compared with the knockdown of a RELA interactor in each case. Genes that are jointly regulated by knockdown of RELA and one of its interactors correspond to the Venn diagrams of (B) and are marked in red. Coefficients of correlation (Pearson's $r$), corresponding $P$ values and coefficients of determination ($r^2$) rare indicated for all comparisons. The complete set of data is provided in Dataset EV3. Source data are available online for this figure.

TFEB, or GLIS2 to DNA (Fig. 8C). As indicated by p values, TFE3, TFEB and GLIS2 motifs under p65/RELA peaks were highly significantly overrepresented, compared to their distribution across the whole genome sequence (Fig. 8C). In all, 19–29% of all RELA peaks contained a RELA motif, but no motif for TFE3, TFEB or GLIS2 (Fig. 8C). Vice versa, 5–25% of all RELA ChIPseq peaks contained a motif for TFE3, TFEB or GLIS2, respectively, but no RELA motif, suggesting indirect recruitment to DNA by PPI (Fig. 8C).

Around 50% of all genes which were deregulated by RELA siRNAs in basal or IL-1α-stimulated conditions (as shown in Fig. 7; Appendix Fig. S6) were associated with at least one motif for either RELA, TFE3/TFEB, GLIS2 (Fig. 8D). In most instances, gene sets contained 1–3 motifs alone or in combination, in line with the notion of RELA genetic subnetworks as described above (Fig. 8D,E). Only 9 genes were annotated with all four motifs, such as *TNFAIP3*, whereas 16 genes, such as *CXCL2*, contained TFE3, GLIS2, and p65/RELA motifs (Fig. 8E). The highly IL-1α-inducible *TNFAIP3* gene locus contains two major p65/RELA peaks within the promoter region that are associated with multiple TFE3, TFEB and GLIS2 bindings sites (Fig. 8F). As a proof of concept, we chose this gene to demonstrate the IL-1α-inducible recruitment of p65 and TFE3 to the promoter of *TNFAIP3* (Fig. 8G). This effect was increased by long-term starvation for 24 h, the condition known to promote translocation of TFE3 to the nucleus (Fig. 8G). The stable depletion of TFE3 partially suppressed p65/RELA recruitment to *TNFAIP3* or *CXCL2* promoters, providing further evidence for the cooperativity of p65/RELA and TFE3 at the chromatin level (Appendix Fig. S7).

Similar results were obtained from the analyses of available motifs for various ZBTB family members that, in addition to ZBTB5, were identified in our p65/RELA interactomes as shown in Fig. 3E. While the DNA-binding motif for ZBTB5 is unknown, ZBTB33, ZBTBT7A and ZBTB7B motifs were clearly enriched under p65/RELA ChIPseq peaks (Appendix Fig. S8). Interestingly, ZBTB7A is the only ZBTB factor, for which a role in the NF-κB system has been described. It was found to bind to p65/RELA and control the accessibility of promoters for p65/RELA (Ramos Pittol et al, 2018).

These combined experimental and bioinformatics analyses suggest that a considerable part of p65/RELA chromatin recruitment could occur indirectly, through interactions with one of its many TF binding partners as defined in this study by proximity labeling.

## Discussion

Transcription factors, which account for ~8% of all human genes, are defined by their ability to interact with DNA and stimulate or repress gene transcription, but their experimentally determined binding sites are not necessarily predictors of the genes that they actively regulate (Cusanovich et al, 2014; Lambert et al, 2018). It has been suggested that this relative lack of specificity is compensated for by cooperativity and synergy between TFs and by their interactions with other nuclear proteins. However, understanding of these interactions and relationships is still very limited (Lambert et al, 2018). Here, we used proximity labeling to investigate the interactome of the REL family member p65/RELA at high resolution. The data reveal hundreds of p65/RELA interactors in a single cell type, demonstrating the enormous extent of intermolecular connectivity of a single mammalian TF. Taken together with the exemplary functional study of selected TF partners, the results have far-reaching implications for interpreting p65/RELA-driven processes in homeostatic and diseased states.

Limited evidence exists for p65/RELA interactors (or any other NF-κB protein) from large-scale studies (Dataset EV5). Overall, 71–92 interactors were reported for NF-κB1 (p50) or p65/RELA in HEK293 cells by means of tandem affinity purification/mass spectrometry (TAP-MS) (Bouwmeester et al, 2004; Li et al, 2015). An average of 61.5 PPIs per TF was found by a survey of 109 TFs using BirA fusions in HEK293 cells, with only 16 interactors for NFKB1 (Goos et al, 2022). The Gilmore group lists 115 p65/RELA-interacting proteins derived from various models and systems (https://www.bu.edu/nf-kb/physiological-mediators/interacting-proteins/), while the STRING database currently documents less than 50 p65/RELA interactors (Fig. 2, Szklarczyk et al, 2019). Thus, our results significantly exceed the number of reported p65/RELA interactors and provide in depth functional validation, which is lacking in the large-scale screens cited above.

The p65/RELA interactome consisted of specifically and dynamically recruited factors, as its composition changed in response to both p65/RELA mutations and cytokine treatment. Some of the IL-1α-dependent effects appear to be mediated by phosphorylation of p65/RELA or its interactors, with evidence for enhanced recruitment of phosphorylated c-JUN to p65/RELA along with a specific protein interaction network. Given that phosphorylation events occur in a substoichiometric manner, it will be of interest to combine miniTurboID with subsequent phosphopeptide enrichment techniques in future studies to gain a

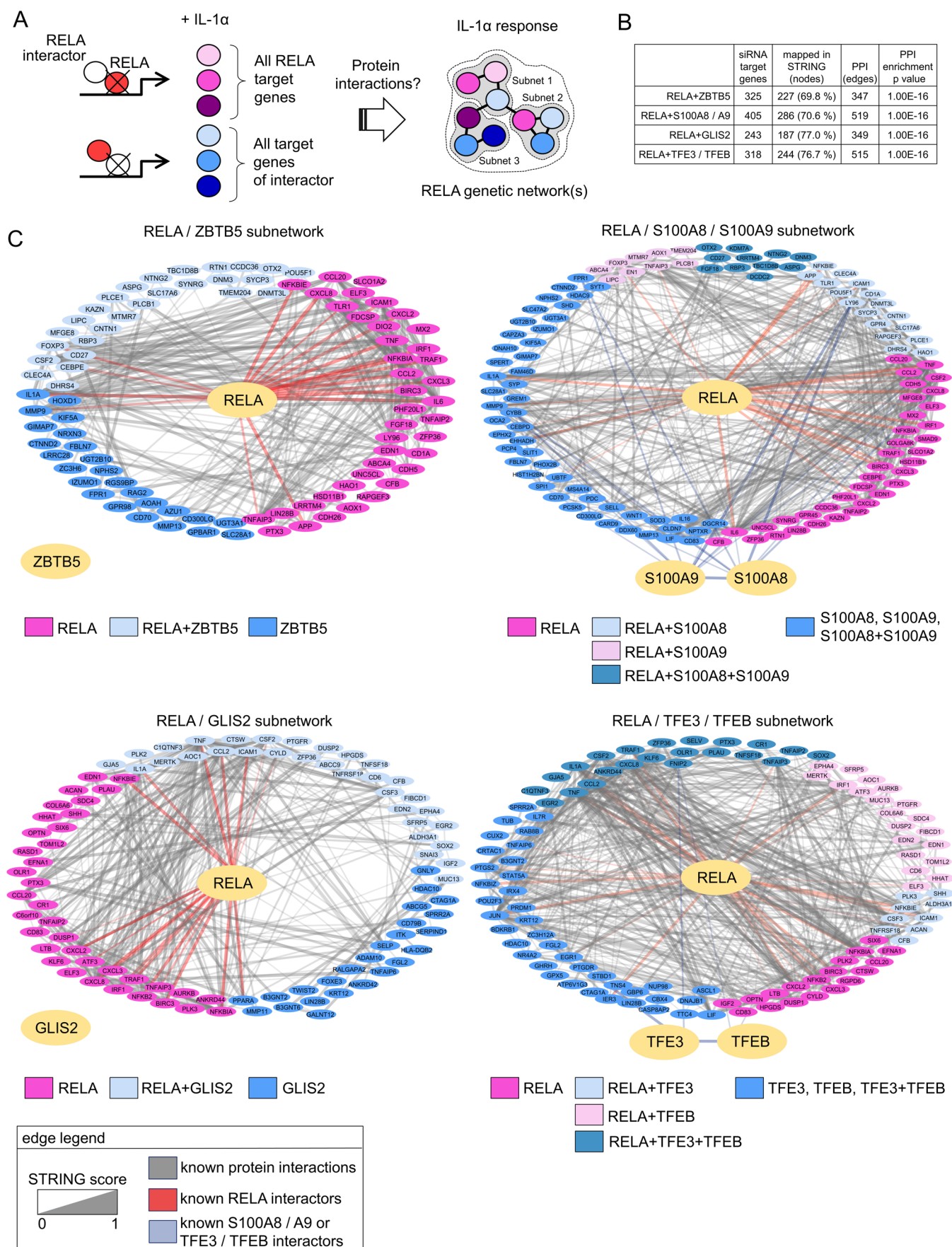

**Figure 7.  IL-1α-regulated RELA genetic networks derived from genome-wide loss of function analysis of its interaction partners.**

(A) Schematic illustrating the strategy to project the protein interactions of all target genes defined by knockdowns of p65/RELA or its interactors in IL-1α-stimulated cells into combined functional networks. (B) Table summarizing the numbers of mapped IDs ( = nodes) corresponding to the gene groups shown in Fig. 6B, their protein interactions ( = edges) and the protein interaction network enrichment *P* values as derived from algorithms embedded in STRING (Franceschini et al, 2013). (C) Cytoscape-derived PPI networks. Nodes are colored and arranged according to the deregulation of the corresponding genes by knockdown of p65/RELA or its interactors. Edges visualize known protein interactions, including the small number of interactions reported for p65/RELA, S100A8/9, and TFE3/TFEB. No interactions were found for ZBTB5 and GLIS2. Source data are available online for this figure.

comprehensive understanding of the role of phosphorylation in the remodeling of p65/RELA protein interaction networks (Muraoka and Adachi, 2024; Urban, 2022).

In this context, it was remarkable that half of the p65/RELA interactors represented other TFs. This result supported earlier studies showing p65/RELA's interaction with basic leucine zipper domain (bZIP) TFs (e.g., JUN, ATF1/2/3/7, FOSL1/2, CEBPA/D) or IRF1 (Jurida et al, 2015; Merika et al, 1998; Stein et al, 1993; Wolter et al, 2008). The sheer number of possible TF interactions suggested an extraordinary level of p65/RELA cooperativity with this class of proteins.

About one-fifth of the p65/RELA interactome consisted of chromatin regulators. Our data reproduced interactions of p65/RELA with the histone acetyltransferases (HATs) p300/CBP (also called EP or CREBBP), TIP60 (KAT5) and the histone deacetylases HDAC1/2 (Ashburner et al, 2001; Brockmann et al, 1999; Merika et al, 1998; Perkins et al, 1997), which were later also confirmed in the chromatin context, e.g., at H3K27-marked enhancers and promoters (Garber et al, 2012; Kim et al, 2012; Mukherjee et al, 2013; Raisner et al, 2018).

Beyond this, miniTurboID greatly advances our understanding of the complexity of the p65/RELA nuclear cofactor interactome as with this method we detected more than 50 subunits of chromatin complexes associated with p65/RELA. The data suggest that p65/RELA preferentially interacts with complexes that promote active gene expression and counteract repressive programs mediated by Polycomb proteins, such as COMPASS, SWI/SNF, or BAF (Cenik and Shilatifard, 2021; Kadoch and Crabtree, 2015; Mashtalir et al, 2020; Hodges et al, 2016; Varga et al, 2021; Schick et al, 2021). Further p65/RELA interactors (KDM6A, KMT2D, NCOA6, PAGR1, PAXIP1) are subunits of COMPASS-like or CHD8 complexes that regulate H3K4 methylation or chromatin remodeling at promoters and enhancers during transcriptional activation (Manning and Yusufzai, 2017; Schuettengruber et al, 2017). However, how different subunits of these complexes are assembled and recruited to chromatin for stimulus-specific functions remains an open question. Our data shed light on a possible role of the p65/RELA interactome to coordinate these events.

NF-κB subunits are known to dimerize, potentially contributing to transcriptional selectivity (Saccani et al, 2003; Siggers et al, 2012; Smale, 2012). Testing two p65/RELA mutants that disrupt chromatin recruitment and activation of TNFα response genes to a similar extent (Riedlinger et al, 2019), we found that dimerization played a larger role in interactions with epigenetic regulators and TFs than DNA binding. This behavior can be reconciled with the observation that, in living cells, promoter-bound NF-κB exists in dynamic, oscillating equilibrium with nucleoplasmic dimers, with short residence times at high-affinity DNA-binding sites (Bosisio et al, 2006). We suggest that miniTurboID, being crosslink-free and rapid, appears to provide a snapshot of the consequences of this

dynamic equilibrium. The distinct interactomes of the E/I or the FL/DD mutants imply that, on average in the cell population studied, the majority of the p65/RELA interactome does not require stable interactions with DNA and the complexes are (pre) assembled outside the chromatin, presumably in the nucleoplasm. Proximity-based labeling thus informs on additional layers of TF cooperativity beyond the coordinated formation of p65/RELA complexes on accessible chromatin templates.

From a genome-wide perspective, the miniTurboID-based interactome suggested a model where p65/RELA, in cooperation with its TF partners and associated epigenetic regulator complexes, instructs the cell to execute specific transcriptional programs. To test this hypothesis, we extended our analysis to three functional levels: (i) a targeted siRNA screen of a panel of high-confidence interactors, (ii) a detailed identification of overlapping sets of target genes, and (iii) the analysis of TF motifs under p65/RELA peaks.

RNAi-mediated suppression of the 38 most enriched novel p65/RELA interactors, demonstrated the gene-specific, functional contributions of 24 TFs, spanning multiple families, in regulating three canonical NF-κB target genes. The TFs exhibited disparate quantitative contributions to basal and IL-1α-inducible gene expression, encompassing both gene activation or repression. These phenotypes are likely indicative of TF cooperativity in the fine-tuning of NF-κB responses in conjunction with gene-specific coactivator/corepressor assemblies, in accordance with observations from an earlier targeted shRNA screen of nuclear cofactors conducted in murine fibroblast cells (Meier-Soelch et al, 2018).

For genome-wide loss-of-function analyses, we focused on candidates from the C2H2 (GLIS2, ZBTB5) and bHLH (TFE3, TFEB) TF families, and S100A8/A9 as non-TF interactors with p65/RELA. GLIS2 and ZBTB5, poorly characterized TFs, are implicated in processes like epithelial–mesenchymal transition and nephronophthisis but not in the NF-κB system (Attanasio et al, 2007; Cheng et al, 2021b; de Dieuleveult and Miotto, 2018; Wilson et al, 2021). S100A8/A9 are typical, secreted drivers of the innate immune responses, but with no role in p65/RELA-mediated transcription (La Spina et al, 2020; Pruenster et al, 2016; Wang et al, 2018). TFEB and TFE3 are known for lysosomal gene regulation under conditions of starvation (Tan et al, 2022; Yang and Wang, 2021), but have also been suggested to contribute to LPS-mediated inflammatory gene secretion in macrophages by unknown mechanisms (La Spina et al, 2020; Pastore et al, 2016). Here, we found that the constitutively phosphorylated forms of TFE3 and TFEB contributed to IL-1α-NF-κB-regulated inflammatory gene expression in non-starved conditions. Additional data at the protein level suggest mutual regulation between p65/RELA, TFE3/TFEB, and GLIS2. The reduced p65/RELA levels in GLIS2 knockdown cells possibly suggest a role of GLIS2 in preventing E3 ligase-mediated ubiquitination and proteasomal degradation of p65/RELA that has been observed in several systems (Geng et al, 2009).

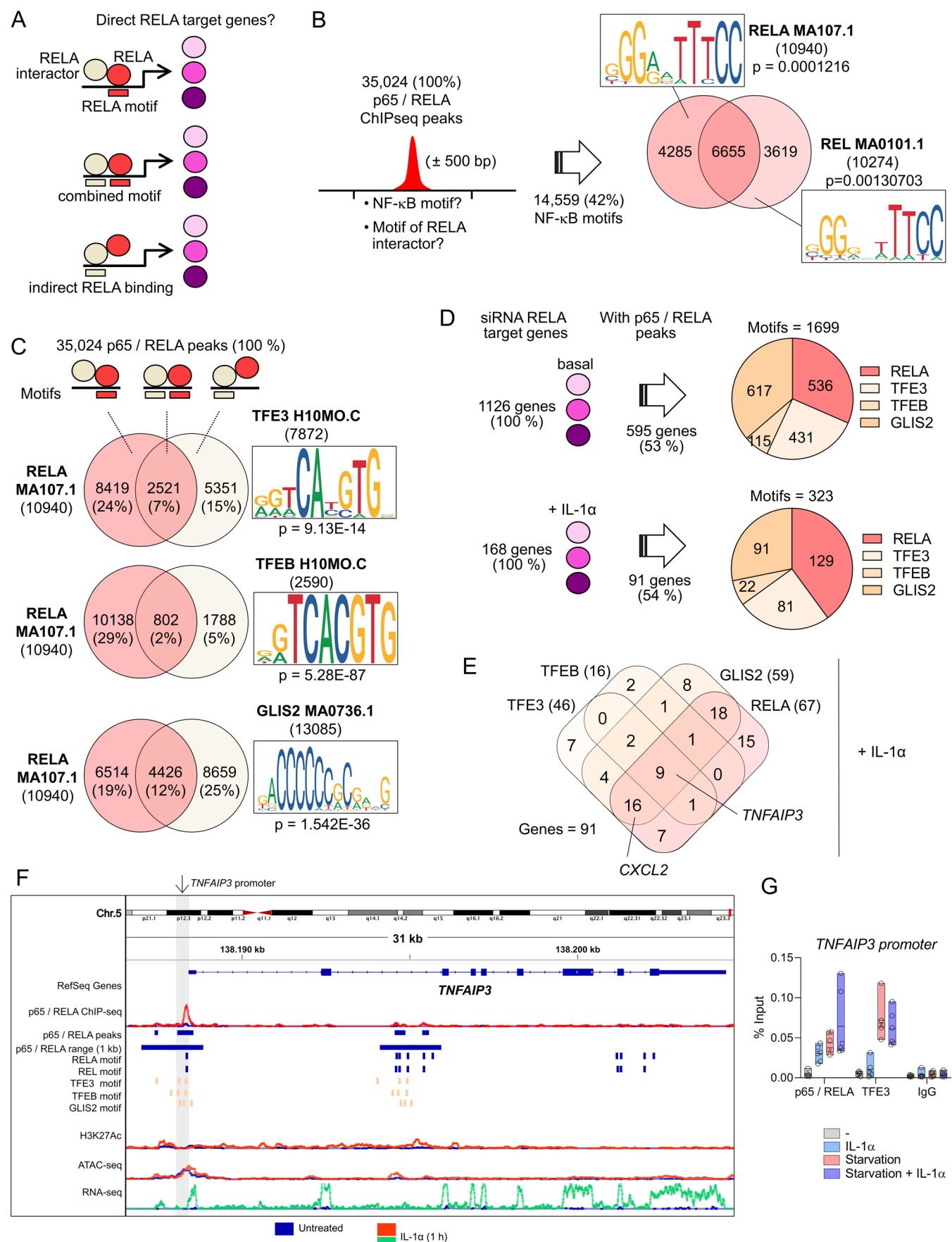

◄ **Figure 8. Motif analysis predicts chromatin recruitment of RELA interactors to p65/RELA ChIPseq peaks.**

(A) Schematic illustrating the strategy to use p65/RELA ChIPseq data for delineating chromatin recruitment of RELA together with its interactors on the basis of DNA motifs and three possible scenarios of interactions. (B) Windows of 1000 base pairs surrounding experimentally determined p65/RELA ChIPseq peaks (Data ref: Jurida et al, 2015; Jurida et al, 2015) were searched for motifs of RELA and REL TFs using matrices from the JASPAR database. P values indicated significant enrichment compared to the whole genome calculated by MEME-ChIP (Ma et al, 2014). The Venn diagram shows the overlap and inserts show motif compositions. (C) Venn diagrams indicating the overlap of motifs found for RELA or the RELA interactors TFE3, TFEB or GLIS2 in chromosomal regions assigned to p65/RELA ChIPseq peaks. P values indicate significant enrichment compared to the whole genome calculated by MEME-ChIP (Ma et al, 2014). Inserts show motif compositions. (D) All target genes that were significantly up- or downregulated under basal or IL-1α-stimulated conditions as shown in Fig. 6 or Appendix Fig. S6 were collected and were examined for their association with a p65/RELA ChIPseq peak. The pie charts show the numbers of RELA, TFE3, TFEB and GLIS2 motifs detected in siRNA RELA target genes with an annotated p65/RELA peak in their promoters or enhancers. (E) Overlap of all genes with p65/RELA peaks in promoters or enhancers and at least one motif for the indicated transcription factors in IL-1α-stimulated conditions. (F) Genome browser view of the *TNFAIP3* locus with p65/RELA ChIPseq peaks, activated enhancers and promoters (H3K27ac), accessible chromatin (ATACseq) and mRNA production (RNAseq) before and after 1 h of IL-1α stimulation. Data sets were from GSE64224, GSE52470 and GSE134436 and are aligned to HG19 (Data ref: Handschick et al, 2014; Data ref: Weiterer et al, 2020; Data ref:Jurida et al, 2015; Handschick et al, 2014; Jurida et al, 2015; Weiterer et al, 2020). p65/RELA binding regions of 1000 bp under p65/RELA peaks and identified TF motifs are indicated by horizontal lines. (G) KB cells were left untreated or were starved for 24 h in HBSS. Half of the cells was treated with IL-1α (10 ng/ml) for 1 h before the end of the experiment. ChIP-qPCR was performed with the indicated antibodies or IgG controls and a primer pair covering the *TNFAIP3* promoter region (marked with an arrow in Fig. 8F) and relative enrichment of *TNFAIP3* promoter fragments was calculated (percent input). Box plots show all data points with means and minimum/maximum values from three independent biological replicates performed with two technical replicates. The complete set of data is provided in Dataset EV4. Source data are available online for this figure.

By intersecting transcriptome-wide analyses of cells with reduced TF levels, we identified gene sets co-regulated by p65/RELA and six of its interactors, differing in basal conditions compared to IL-1α-activated cells. The combinatorial actions of multiple TFs and associated epigenetic regulators, as indicated by the p65/RELA interactome, are reminiscent of gene regulatory networks (GRN) (Spitz and Furlong, 2012). GRNs can consist of several subnetworks, each of which executes individual segments of a complex biological process (Davidson, 2010; Peter and Davidson, 2011). We tested this concept for the p65/RELA-driven IL-1α-response by bioinformatics analyses to find that, at the systems level, a large part of the genes affected by RNAi form subnetworks with dense functional interactions. Genome-wide ChIPseq data revealed multiple motifs for TFE3/TFEB and GLIS2 coinciding with 35,000 p65/RELA peaks. A recent study reported a similar scale of p65 binding events across different mammalian species (31,602–90,570 p65/RELA peaks) (Alizada et al, 2021). NF-κB binds to both accessible and nucleosome-occluded chromatin in a TNFα-dependent manner, with defined chromatin states regulated by p65/RELA that distinguish conserved and constitutive functions from specific pro-inflammatory functions of NF-κB (Alizada et al, 2021). Our data suggest that these chromatin states may, at least in part, be established by basal or IL-1α-regulated multiprotein complexes associated with p65/RELA.

Identifying p65/RELA genetic subnetworks driven by p65/RELA and its TF interactors provides a novel explanation for the NF-κB response's stimulus-specificity and its impact on the epigenome which, in macrophages, was recently attributed to fluctuations in cellular NF-κB activity (Adelaja et al, 2021; Cheng et al, 2021a). Combining proximity-based p65/RELA interactomics with RNAi experiments, as demonstrated in our study, thus complements and extends approaches that sought to combine ChIPseq with RNAseq to unveil the regulation of context-dependent NF-κB target genes by the two p65/RELA TAD domains (Ngo et al, 2020).

The p65/RELA subunit is the only NF-κB subunit whose deletion leads to embryonic lethality in mice (Beg and Baltimore, 1996). In most cells, it comprises the predominant NF-κB transcriptional activity and regulates a plethora of processes during development, the immune response and cancer (Mitchell and Carmody, 2018; Zhang et al, 2017). Consequently, the p65/RELA pathway is tightly regulated to adjust nuclear p65/RELA

concentration and kinetics (Meier-Soelch et al, 2021). Despite significant mechanistic insights from genetically altered cells or organisms with altered p65/RELA expression and improved knowledge of p65/RELA-driven genetic changes, our understanding of this pathway has not yet advanced enough to yield specific and effective anti-p65/RELA drugs. In line with a recent evaluation of NF-κB component copy numbers in various immune cells (Kok et al, 2021), our study demonstrates that quantitative proteomics can significantly contribute to understanding the NF-κB pathway's connectivity. Assuming that stable, physically connected protein complexes are more often labeled with biotin compared to transient or indirect interactions, proximity labeling also informs on interaction strength and frequency, which will allow further classification of p65/RELA networks.

In summary, the high-resolution novel p65/RELA interactome and its gene regulatory logics reported in this study provide a rich resource and a new framework for explaining p65/RELA function in living cells.

## Limitations

Our study was restricted to a single cell type in order to standardize and integrate the different levels of molecular analyses. HeLa cells express all core components of the NF-κB pathway and have a fully functional IL-1 system (Weiterer et al, 2020). They represent one of the most widely used cellular models to date (Adey et al, 2013). Thus, we expect that p65/RELA interactomes of other cell types will show some overlap but will also differ. A further limitation is the necessity to ectopically express a p65/RELA fusion protein that, although performed in a p65/RELA-deficient background using a conditional system, could influence the stoichiometry of some of the interactions we discovered with a potential preference for constitutive binding events or for p65/RELA homodimers.

Proximity labeling generates a cumulative snapshot of protein–protein interactions, but does not allow inference of physical interactions. Because the analyses were performed on whole-cell extracts, no information is available on the subcellular regulation of the interactions, although bioinformatics analyses revealed a clear predominance of nuclear p65/RELA interactors. MiniTurboID does not resolve the time wise sequence of events and rather reports cumulatively on all interactions that happened within the 70 min of labeling time.

Despite including three types of negative controls (omission of doxycycline or biotin, miniTurbo enzyme only) and the usage of p65/RELA point mutants to confirm the specificity of the p65/RELA interactome, we cannot fully exclude false positive hits. 189 factors (51.6%) of our 366 p65/RELA interactors were absent from a large list of 3977 probable unspecific interactors obtained with a GFP-BirA* fusion protein in a different cell type, whereas the overlapping ones were detected mostly at low frequency and low spectral counts (Appendix Fig. S9) (Gawriyski et al, 2024). This also applies to TFE3, whose interaction with p65/RELA was validated in our study by several means (Appendix Fig. S9A). However, the statistical definition of background proteins using GFP baits is difficult to compare with our targeted approach that includes multiple levels of functional validation.

The generation of a specified list of 366 RELA/p65 interactors resulted from the adoption of relatively rigorously defined filtering criteria based on a combination of negative controls, levels of enrichment and $t$ test criteria. The same applied to the definition of p65/RELA target genes. As with any bioinformatics analysis, it is clear that weakening or tightening these filtering criteria would increase or decrease the number of p65/RELA interactors or genes affected by siRNA knockdown. Similarly, the annotation of factors to gene ontologies or protein networks depends on the underlying databases and the parameters set. Here, we chose to use the default settings of current versions of Metascape and STRING, respectively. However, publication of the entire raw dataset and all source data will allow colleagues in the field to (i) track all analysis results and (ii) generate alternative p65/RELA interactor lists and GRN under their own chosen criteria.

# Methods

### Reagents and tools table

| Reagent/resource | Reference or source | Identifier or catalog number |
|---|---|---|
| **Experimental models** | | |
| HeLa (Human epithelial cervix carcinoma cells) | Schmitz lab (Weiterer et al, 2020) | |
| HeLa-pX459-RELA-sg1 (Δp65) | Kracht lab | CRISPR-Cas9 engineered |
| HeLa lentiCRISPR V2 (EV) | Kracht lab | CRISPR-Cas9 engineered |
| HeLa lentiCRISPR V2-RELA-sg1 (p65 KO) | Kracht lab | CRISPR-Cas9 engineered |
| HeLa lentiCRISPR V2-TFE3-sg5 (TFE3 KO) | Kracht lab | CRISPR-Cas9 engineered |
| KB cells | Kracht lab | |
| HEK293FT cells | Thermo Fisher Scientific | Cat.#R70007 |
| E.coli XL-1 Blue (Chemically Competent) | Kracht lab | |
| NEB® Stable Competent E. coli bacteria | New England Biolabs | Cat.#C3040H |
| **Recombinant DNA** | | |
| lentiCRISPR V2 | Addgene (#52961) | Source for cloning lentiviral sgRNA vectors |

| Reagent/resource | Reference or source | Identifier or catalog number |
|---|---|---|
| lentiCRISPR V2 sgRNA 1 RELA | Kracht lab, cloned in this study | Generating lentiviral supernatants for CRISPR KO cells |
| lentiCRISPR V2 sgRNA 5 TFE3 | Kracht lab, cloned in this study | Generating lentiviral supernatants for CRISPR KO cells |
| pMD2.G | Addgene | Cat.#12259 |
| psPAX2 | Addgene | Cat.#12260 |
| linker-HA-miniTurbo in pUC57-Bsal-Free | Designed by M.L. Schmitz, ordered from General Biosystems | Source for cloning of HA-miniTurbo |
| pEF-Puro-hu p65 E/I-HA | Schmitz Lab (Riedlinger et al, 2019) | Source for cloning of p65 DNA-binding-deficient mutant (E/I) |
| pEF-Puro-hu p65 FL/DD-HA | Schmitz Lab (Riedlinger et al, 2019) | Source for cloning of p65 dimerization-deficient mutant (FL/DD) |
| pEF-Puro-hu p65 WT-HA | Schmitz Lab (Riedlinger et al, 2019) | Source for cloning of p65 wildtype (WT) |
| pSpCas9(BB)-2A-Puro (pX459) | Addgene (#48139) | Source for cloning sgRNA vectors |
| pTet-on-Puro-HA-miniTurbo | Kracht lab, cloned in this study | HA-miniTurbo control vector for BioID |
| pTet-on-Puro-Myc-BirA | Gift from Schmitz lab | Vector pTet-on-Puro backbone source for cloning of p65-HA-miniTurbo vectors |
| pTet-on-Puro-p65(E/I)-HA-miniTurbo | Kracht lab, cloned for this study | p65(E/I)-HA-miniTurbo for BioID |
| pTet-on-Puro-p65(FL/DD)-HA-miniTurbo | Kracht lab, cloned for this study | p65(FL/DD)-HA-miniTurbo for BioID |
| pTet-on-Puro-p65(wt)-HA-miniTurbo | Kracht lab, cloned for this study | p65(wt)-HA-miniTurbo for BioID |
| pX459 sgRNA 1 RELA | Kracht lab | Generating CRISPR KO cells |
| **Antibodies** | | |
| Anti-c-JUN (mouse, monoclonal clone 4H9) for WB | Invitrogen | Cat.#MA5-15889 |
| Anti-GLIS2 (rabbit polyclonal) for WB, PLA | Thermo Fisher Scientific | Cat.#PA5-40314 |
| Anti-HA (mouse, monoclonal clone 12CA5) for WB | Roche | Cat.#11583816001 |
| Anti-HA (rabbit, polyclonal) for WB | Abcam | Cat.#ab9110 |
| Anti-IκBα (rabbit, polyclonal) for WB | Cell Signaling | Cat.#9242 |
| Anti-NF-κB p50 (E-10) (mouse monoclonal) for WB, PLA | Santa Cruz Biotechnology | Cat.#sc-8414 |
| Anti-NF-κB p50 (NLS) (rabbit polyclonal) for ChIP | Santa Cruz Biotechnology | Cat.#sc-114 (discontinued) |
| Anti-NF-κB p65 (C-20) (rabbit polyclonal) for WB, ChIP, PLA | Santa Cruz Biotechnology | Cat.#sc-372 (discontinued) |

| Reagent/resource | Reference or source | Identifier or catalog number |
|---|---|---|
| Anti-NF-κB p65 (F-6) (mouse monoclonal) for WB, IF | Santa Cruz Biotechnology | Cat.#sc-8008 |
| Anti-NF-κB p65 (goat polyclonal) for PLA | Bethyl Lab. | Cat.#A303-945A |
| Anti-NF-κB p65 XP® (rabbit monoclonal clone D14E12) for ChIP | Cell Signaling | Cat.#8242 |
| Anti-Phospho c-JUN Ser63 (rabbit polyclonal) for WB | Cell Signaling | Cat.#9261 |
| Anti-Phospho NF-κB p65 Ser276 (rabbit polyclonal) for WB | Cell Signaling | Cat.#3037 (discontinued) |
| Anti-Phospho NF-κB p65 Ser468 (rabbit polyclonal) for WB | Cell Signaling | Cat.#3039 |
| Anti-Phospho NF-κB p65 Ser529 (rabbit monoclonal clone E3K3J) for WB | Cell Signaling | Cat.#78764 |
| Anti-Phospho NF-κB p65 Ser536 (rabbit monoclonal clone 93H1) for WB | Cell Signaling | Cat.#3033 |
| Anti-Phospho Polymerase II Ser5 (rabbit polyclonal) for WB | Abcam | Cat.#ab5131 |
| Anti-Polymerase II (Pol II) (mouse monoclonal) for WB | Millipore | Cat.#17-620 |
| Anti-TFE3 (rabbit polyclonal) for ChIP | Proteintech | Cat.#14480-1-AP |
| Anti-TFE3 (rabbit polyclonal) for WB, PLA, ChIP | Sigma-Aldrich | Cat.#HPA023881 (discontinued) |
| Anti-TFEB (rabbit polyclonal) for WB, PLA | Cell Signaling | Cat.#4240 |
| Anti-Tubulin (TU-02) (mouse monoclonal) for WB | Santa Cruz Biotechnology | Cat.#sc-8035 |
| Anti-ZBTB5 (rabbit polyclonal) for WB, PLA | Sigma | Cat.#HPA021521 |
| Anti-β-Actin (C4) (mouse monoclonal) for WB | Santa Cruz Biotechnology | Cat.#sc-47778 |
| Dylight 488-coupled anti-mouse IgG | ImmunoReagent | Cat.#DkxMu-003D488NHSX |
| HRP-conjugated/anti-mouse IgG (goat) | DakoCytomation | Cat.#P0447 |
| HRP-conjugated/anti-rabbit IgG (goat) | DakoCytomation | Cat.#P0448 |
| HRP-Streptavidin | PerkinElmer | Cat.#NEL750001EA |
| Normal mouseIgG | Santa Cruz Biotechnology | Cat.#sc-2025 |
| Normal rabbit IgG | Cell Signaling | Cat.#2729 |
| TrueBlot® ULTRA Anti-Mouse Ig HRP | Rockland | Cat.# 18-8817-30 |
| **Oligonucleotides (5'-3' sequence, restriction site overhangs in bold)** | **Forward** | **Reverse** |
| Cloning of p65 variants | **TCCAGCCTACCGGTAAC**ATGGACGAACTGTTCCCCTCATCTT | **TATCGATGTACA**GGAGCTGATCTGACTCAGCAGGGCTG |
| qPCR primer for ChIP *CXCL2* | GTCAGACCCGGACGTCACT | ACCCCTTTTATGCATGGTTG |
| qPCR primer for ChIP neg.ctrl. (gene free region upstream *CXCL8*) | ATCATGGGTCCTCAGAGGTCAGAC | GGTGGGAGGGAGGTGTTATCTAATG |

| Reagent/resource | Reference or source | Identifier or catalog number |
|---|---|---|
| qPCR primer for ChIP *TNFAIP3* | TGCACAGCCCAAACTTTTCA | GTGAGTCACCTGGGCATTTC |
| Sanger sequencing for HA-miniTurbo insert | CATCCACGCTGTTTTGACC | TAAGATCTGGCCTCCGCG and CGATTCGATCCAGCAGGTA |
| Sanger sequencing sgRNA in lentiCRISPR V2 (hU6-fwd) | GAGGGCCTATTTCCCATGATT | |
| sgRNA 1 for *RELA* | **CACCG**GCTTCCGCTACAAGTGCGA | **AAAC**TCGCACTTGTAGCGGAAGC**C** |
| sgRNA 5 for *TFE3* | **CACCG**GAGAGGCAGGTGCAGGACTG | **AAAC**CAGTCCTGCACCTGCCTCTC**C** |
| **siRNAs** | | |
| *ATXN1L* | Qiagen | Cat.#GS342371 |
| *CEBPD* | Qiagen | Cat.#GS1052 |
| *CRTC2* | Qiagen | Cat.#GS200186 |
| *EBF1* | Qiagen | Cat.#GS1879 |
| *Firefly Luciferase* | Eurofins Genomics synthesis order | CGUACGCGGAAUACUUCGA |
| *FOSL1* | Qiagen | Cat.#GS8061 |
| *FOXC1* | Qiagen | Cat.#GS2296 |
| *FOXC2* | Qiagen | Cat.#GS2303 |
| *GATA6* | Qiagen | Cat.#GS2627 |
| *GLIS2* | Qiagen | Cat.#GS84662 |
| *HMG20B* | Qiagen | Cat.#GS10362 |
| *HOMEZ* | Qiagen | Cat.#GS57594 |
| *ISL1* | Qiagen | Cat.#GS3670 |
| *MAFG* | Qiagen | Cat.#GS4097 |
| *MAMLD1* | Qiagen | Cat.#GS10046 |
| *MSX1* | Qiagen | Cat.#GS4487 |
| *N4BP3* | Qiagen | Cat.#GS23138 |
| *NFYC* | Qiagen | Cat.#GS4802 |
| *PBX2* | Qiagen | Cat.#GS5089 |
| *PHC1* | Qiagen | Cat.#GS1911 |
| *PHLDA2* | Qiagen | Cat.#GS7262 |
| *RAD54L2* | Qiagen | Cat.#GS23132 |
| *RAP1A* | Qiagen | Cat.#GS5906 |
| *RAP1B* | Qiagen | Cat.#GS5908 |
| *RELA/p65* | Qiagen | Cat.#GS5970 |
| *RUNX2* | Qiagen | Cat.#GS860 |
| *S100A8* | Qiagen | Cat.#GS6279 |
| *S100A9* | Qiagen | Cat.#GS6280 |
| *SIX5* | Qiagen | Cat.#GS147912 |
| *SKI* | Qiagen | Cat.#GS6497 |
| *SKIL* | Qiagen | Cat.#GS6498 |
| *SS18* | Qiagen | Cat.#GS6760 |
| *TBX1* | Qiagen | Cat.#GS6899 |
| *TFE3* | Qiagen | Cat.#GS7030 |
| *TFEB* | Qiagen | Cat.#GS7942 |
| *TNIP1* | Qiagen | Cat.#GS10318 |

| Reagent/resource | Reference or source | Identifier or catalog number |
|---|---|---|
| ZBTB34 | Qiagen | Cat.#GS403341 |
| ZBTB5 | Qiagen | Cat.#GS9925 |
| ZIC2 | Qiagen | Cat.#GS7546 |
| ZMIZ1 | Qiagen | Cat.#GS57178 |
| ZMIZ2 | Qiagen | Cat.#GS83637 |
| **TaqMan Gene Expression Assays (FAM-MGB labeled)** | | |
| ACTB | Applied Biosystems | Cat.#HS99999903_m1 |
| ATP6V0D1 | Applied Biosystems | Cat.#HS00371517_m1 |
| ATXN1L | Applied Biosystems | Cat.#HS01370353_g1 |
| CCL2 | Applied Biosystems | Cat.#HS00234140_m1 |
| CEBPD | Applied Biosystems | Cat.#HS00270931_s1 |
| CRTC2 | Applied Biosystems | Cat.#HS01064500_m1 |
| CSF2 | Applied Biosystems | Cat.#HS00929873_m1 |
| CXCl2 | Applied Biosystems | Cat.#HS00236966_m1 |
| EBF1 | Applied Biosystems | Cat.#HS01092694_m1 |
| FOSL1 | Applied Biosystems | Cat.#HS04187686_g1 |
| FOXC1 | Applied Biosystems | Cat.#HS00559473_s1 |
| FOXC2 | Applied Biosystems | Cat.#HS00270951_s1 |
| GAPDH | Applied Biosystems | Cat.#HS02758991_g1 |
| GATA6 | Applied Biosystems | Cat.#HS00232018_m1 |
| GLIS2 | Applied Biosystems | Cat.#HS00261493_m1 |
| GUSB | Applied Biosystems | Cat.#HS99999908_m1 |
| HEXA | Applied Biosystems | Cat.#HS00942659_m1 |
| HMG20B | Applied Biosystems | Cat.#HS00173091_m1 |
| HOMEZ | Applied Biosystems | Cat.#HS00603839_m1 |
| IL8 | Applied Biosystems | Cat.#HS00174103_m1 |
| ISL1 | Applied Biosystems | Cat.#HS00158126_m1 |
| LAMP1 | Applied Biosystems | Cat.#HS00931461_m1 |
| MAFG | Applied Biosystems | Cat.#HS00361648_g1 |
| MAMLD1 | Applied Biosystems | Cat.#HS00193976_m1 |
| MAP1LC3B | Applied Biosystems | Cat.#HS00917682_m1 |
| MCOLN1 | Applied Biosystems | Cat.#HS01100653_m1 |

| Reagent/resource | Reference or source | Identifier or catalog number |
|---|---|---|
| MSX1 | Applied Biosystems | Cat.#HS00427183_m1 |
| N4BP3 | Applied Biosystems | Cat.#HS01585915_g1 |
| NFKBIA | Applied Biosystems | Cat.#HS00153283_m1 |
| NFYC | Applied Biosystems | Cat.#HS00360259_g1 |
| PBX2 | Applied Biosystems | Cat.#HS00855025_s1 |
| PHC1 | Applied Biosystems | Cat.#HS01863307_s1 |
| PHLDA2 | Applied Biosystems | Cat.#HS00169368_m1 |
| RAD54L2 | Applied Biosystems | Cat.#HS00379387_m1 |
| RAP1A | Applied Biosystems | Cat.#HS01092205_g1 |
| RAP1B | Applied Biosystems | Cat.#HS04275955_g1 |
| RELA | Applied Biosystems | Cat.#HS01042019_g1 |
| RUNX2 | Applied Biosystems | Cat.#HS01047973_m1 |
| S100A8 | Applied Biosystems | Cat.#HS00374264_g1 |
| S100A9 | Applied Biosystems | Cat.#HS00610058_m1 |
| SIX5 | Applied Biosystems | Cat.#HS01650774_m1 |
| SKI | Applied Biosystems | Cat.#HS01057032_m1 |
| SKIL | Applied Biosystems | Cat.#HS01045418_m1 |
| SQSTM1 | Applied Biosystems | Cat.#HS00177654_m1 |
| SS18 | Applied Biosystems | Cat.#HS01075912_m1 |
| TBX1 | Applied Biosystems | Cat.#HS00962558_g1 |
| TFE3 | Applied Biosystems | Cat.#HS00232406_m1 |
| TFEB | Applied Biosystems | Cat.#HS00292981_m1 |
| TNFAIP3 | Applied Biosystems | Cat.#HS00234713_m1 |
| TNIP1 | Applied Biosystems | Cat.#HS00374581_m1 |
| ZBTB34 | Applied Biosystems | Cat.#HS00291772_s1 |
| ZBTB5 | Applied Biosystems | Cat.#HS04996213_m1 |
| ZIC2 | Applied Biosystems | Cat.#HS00600845_m1 |
| ZMIZ1 | Applied Biosystems | Cat.#HS01119362_m1 |
| ZMIZ2 | Applied Biosystems | Cat.#HS00230211_m1 |
| **Chemicals, enzymes, and other reagents** | | |

| Reagent/resource | Reference or source | Identifier or catalog number |
|---|---|---|
| Ampicillin sodium salt | BioChemica | Cat.#A0839,0025 |
| APS (Ammonium peroxodisulfate) | Merck | Cat.#1.01201.0500 |
| ATP (Adenosine triphosphate) | Thermo Fisher Scientific | Cat.#R0441 |
| BD Difco™ Dehydrated Culture Media: LB Broth, Miller (Luria-Bertani) | BD | Cat.#244610 |
| Binding Buffer NTB | Macherey-Nagel | Cat.#740595.150 |
| Biotin | Sigma-Aldrich | Cat.#B4501-100MG |
| Branched PEI (Polyethylenimine) | Sigma-Aldrich | Cat.#408727 |
| BSA | Serva | Cat.#11930.04 |
| BshTI (AgeI) FastDigest | Thermo Fisher Scientific | Cat.#FD1464 |
| Bsp1407I (BsrGI) | Thermo Fisher Scientific | Cat.#FD0934 |
| Chromabond C18WP spin columns | Macherey-Nagel | Cat.#730522 |
| DMEM (Dulbecco´s Modified Eagle´s Medium) #P40-47500 | PAN Biotech | Cat.#P04-03550 |
| dNTP mix | Thermo Fisher Scientific | Cat.#R0192 |
| Doxycycline | Sigma-Aldrich | Cat.#D9891 |
| DPBS (Dulbecco's Phosphate Buffered Saline) | PAN Biotech | Cat.#P04-36500 |
| DTT (Dithiothreitol) | Serva | Cat.#20710.04 |
| E-64 | Sigma-Aldrich | Cat.#324890 |
| ECL Western Blotting Detection Reagent | GE Healthcare/ Cytiva | Cat.#RPN2106 |
| Esp3I (BsmBI) FastDigest | Thermo Fisher Scientific | Cat.#FD0454 |
| FBS Good Forte | PAN Biotech | Cat.#P40-47500 |
| Formaldehyde (37%) | Applichem | Cat.#A0877.0250 |
| Geneticin | Gibco | Cat.#10131019 |
| HBSS (Hanks' Balanced Salt solution) | PAN Biotech | Cat.#P04-32505 |
| Hi-PerFect transfection reagent | Qiagen | Cat.#301705 |
| Hoechst 33342 | Thermo Fisher Scientific | Cat.#H3570 |
| ibiTreat μ-Slide VI 0.4 | Ibidi | Cat.#80606 |
| IL-1α (human; recombinant) | Kracht lab | |
| Immobilon Western Chemiluminescent HRP Substrate | Merck Millipore | Cat.#WBKLS0500 |
| Leupeptin hemisulfate | Carl Roth | Cat.#CN33.2 |
| L-Glutamine | PAN Biotech | Cat.#P04-80100 |
| Lipofectamine™ LTX and PLUS™ Reagent | Thermo Fisher Scientific | Cat.#15338100 |
| Microcystin | Enzo Life Sciences | Cat.#ALX-350-012-M001 |
| MluI FastDigest | Thermo Fisher Scientific | Cat.#FD0564 |
| Non-essential amino acids (NEAA) | Gibco | Cat.# 11140-035 |

| Reagent/resource | Reference or source | Identifier or catalog number |
|---|---|---|
| Opti-MEM™ | Gibco | Cat.#31985070 |
| PageRuler™ Prestained Protein Ladder | Thermo Fisher Scientific | Cat.#26616 |
| Paraformaldehyde (4%) | Santa Cruz | Cat.#sc-281692 |
| Penicillin/Streptomycin | PAN Biotech | Cat.#P06-07100 |
| Pepstatin A | Applichem | Cat.#A2205 |
| Phusion™ high-fidelity DNA polymerase | Thermo Fisher Scientific | Cat.#F-530XL |
| PMSF | Sigma-Aldrich | Cat.#P-7626 |
| Polybrene (Hexadimethrine bromide) | Sigma-Aldrich | Cat.#S107689 |
| Protease inhibitor cocktail tablets | Roche | Cat.#11873580001 |
| Protein A-Sepharose® CL-4B Beads | GE Healthcare | Cat.#17-0780-01 |
| Protein G Sepharose® 4 Fast Flow Beads | GE Healthcare | Cat.#17-0618-01 |
| Proteinase K | Macherey-Nagel | Cat.#740506 |
| Puromycin | Merck Millipore | Cat.#540411-100MG |
| Random Hexamer Primer | Thermo Fisher Scientific | Cat.#S0142 |
| RevertAid Reverse Transcriptase | Thermo Scientific | Cat.#EP0441 |
| RNAse A | Thermo Scientific | Cat.#EN0531 |
| ROTI®Load (4X) | Carl Roth | Cat.#K929.3 |
| ROTI®PVDF membrane | Carl Roth | Cat.#T830.1 |
| ROTI®Quant | Carl Roth | Cat.#K929.3 |
| ROTIPHORESE®Gel 30 (Acrylamide solution) | Carl Roth | Cat.#3029.1 |
| Saponin | Sigma-Aldrich | Cat.#S4521-10G |
| Sodium pyruvate | Gibco | Cat.#11360-039 |
| Standard nutrient agar 1 | Merck | Cat.#1.07881.0500 |
| Streptavidin Agarose Resin Beads | Thermo Fisher Scientific | Cat.#20353 |
| T4 DNA Ligase | Invitrogen | Cat.#15224041 |
| T4-Polynukleotid-Kinase (PNK) | Thermo Fisher Scientific | Cat.#EK0031 |
| TEMED (N,N,N',N'-Tetramethyl ethylenediamine) | Sigma Life Science | Cat.#T9281-25ML |
| Tetracycline-free FBS | PAN Biotech | Cat.#P30-3602 |
| TrueBlot® Anti-Mouse Ig IP Agarose Beads | Rockland | Cat.#00-8811-25 |
| Trypsin, MS approved | Serva | Cat.#37286.04 |
| Trypsin/EDTA | PAN Biotech | Cat.#P10-023100 |
| Urea | Thermo Fisher Scientific | Cat.#29700 |
| Whatman® gel blotting paper, Grade GB003 | Cytiva | Cat.#10426892 |
| **Kits** | | |
| Duolink® in Situ PLA® Detection Reagents Orange | Sigma-Aldrich | Cat.#DUO92007 |
| Duolink® in Situ PLA® Probe Anti-Goat MINUS | Sigma-Aldrich | Cat.#DUO92006 |
| Duolink® in Situ PLA® Probe Anti-Rabbit PLUS | Sigma-Aldrich | Cat.#DUO92002 |

| Reagent/resource | Reference or source | Identifier or catalog number |
|---|---|---|
| Fast SYBR™ Green PCR Master Mix | Applied Biosystems | Cat.#4385612 |
| NucleoBond® Xtra Midi Kit | Macherey-Nagel | Cat.#740410.50 |
| NucleoSpin® Gel and PCR Clean-Up Kit | Macherey-Nagel | Cat.#740609.50 |
| NucleoSpin® Plasmid Kit | Macherey-Nagel | Cat.#740588.250 |
| NucleoSpin® RNA Kit | Macherey-Nagel | Cat.#740955.250 |
| PCR Mycoplasma Test Kit | Applichem | Cat.#A3744 |
| SurePrint G3 Human Gene Expression v3 8x60K Microarray Kit | Agilent Technologies | Cat.# G4851C |
| TaqMan® Fast Universal PCR Master Mix | Applied Biosystems | Cat.#4352042 |
| TaqMan® PreAmp Cells-to-CT™ Kit | Invitrogen | Cat.#4387299 |
| Software and webtools | | |
| Adobe Photoshop CS2 v. 9.0 | Adobe | |
| BlobFinder v. 3.0 beta | Allalou and Wahlby, 2009 | |
| Cellpose v. 2.2.3 | Pachitariu and Stringer, 2022 | |
| CellProfiler v.4.2.5 | Stirling et al, 2021 | |
| Cytoscape v.3.8.2/3.9.1 | Shannon et al, 2003 | |
| DNAStAR v. 11.1.0 (59) | DNASTAR | |
| Draw Venn | http://bioinformatics.psb.ugent.be/webtools/Venn/ | |
| GraphPadPrism v.9.5.1 | GraphPad Software | |
| IGV Viewer v.2.8.0 | Robinson et al, 2011 | |
| ImageLab v.6.1.0 build 7 | Bio-Rad | |
| LasX v.3.7.4.23463 | Leica | |
| limma package from BioConductor | Ritchie et al, 2015 | |
| Mapix v.9.0.0 | Innopsys. (2023). Mapix (Version 9.0) [Software]. Innopsys. https://www.innopsys.com/mapix/ | |
| MaxQuant v.1.6.17.0 | Tyanova et al, 2016a | |
| Metascape | Zhou et al, 2019, https://metascape.org/ | |
| Microsoft Office 2016 | Microsoft | |
| Pathview R-package 1.18.2 and R 3.4.4 | https://pathview.unc-c.edu/ | |
| Perseus v.1.6.14 | Tyanova and Cox, 2018 | |
| Primer3Plus v. available 11-2ß18 or later | https://www.primer3plus.com/ | |
| R software v. 3.4.4 | R Core Team (2023) | |

| Reagent/resource | Reference or source | Identifier or catalog number |
|---|---|---|
| STRING v.11.0 or newer | Szklarczyk et al, 2023, https://string-db.org/ | |
| Other | | |
| Focused-Ultrasonicator Covaris | Covaris | S220x |
| milliTUBE 1 ml AFA Fiber | Covaris | Cat.#520135 |
| Bioruptor | Diagenode | Plus |

## Cell lines, cytokine treatment, and starvation

HeLa and KB cells were maintained in Dulbecco's modified Eagle's medium (DMEM; PAN Biotech; #P04-03550), complemented with 10% filtrated bovine serum (FBS Good Forte; PAN Biotech; #P40-47500) or tetracycline-free FBS (PAN Biotech; #P30-3602), 2 mM L-glutamine, 100 U/ml penicillin and 100 µg/ml streptomycin at 37 °C with a humidified atmosphere of 5% $CO_2$. Cells were tested for mycoplasma with PCR Mycoplasma Test Kit (Applichem; #A3744), and their identity was confirmed by commercial STR testing at the DSMZ-German Collection of Microorganisms and Cell Cultures; https://www.dsmz.de/dsmz) as previously described (Weiterer et al, 2020). Stable pools of p65-depleted cells (HeLa Δp65), generated by transfection of pX459-based CRISPR/Cas9 constructs (Weiterer et al, 2020), were selected and maintained in puromycin (1 µg/ml). Lentivirally transduced HeLa cells were selected by puromycin treatment for the depletion of p65 (p65 KO) or TFE3 (TFE3 KO) or an empty vector control without sgRNA insert (EV). Prior to all experiments, puromycin was omitted for 24 h. Human recombinant IL-1α was prepared in our laboratory as described (Rzeczkowski et al, 2011) and used at 10 ng/ml final concentration in all experiments by adding it to the cell culture medium for the indicated time points. Starvation of cells was induced by washing the cells four times with Hanks' Balanced Salt solution (HBSS; PAN Biotech; #P04-32505) for the indicated time points. Starved cells were compared to non-treated control cells or cells washed four times with HBSS and then supplemented with their own culture medium to exclude effects caused by the washing procedure. HEK293FT cells (Thermo Fisher Scientific; #R70007) were cultured in DMEM high glucose with 10% FBS, 0.1 mM non-essential-amino acids (Gibco; #11140-035), 6 mM L-glutamine (PAN Biotech; #P04-80100), 1 mM sodium pyruvate (Gibco; #11360-039) and maintained according to the manufacturer's recommendation with 500 µg/ml geneticin (Gibco; #10131019).

## Reagents and antibodies

The following reagents were used: Leupeptin hemisulfate (Carl Roth, #CN33.2; solved in ddH$_2$O), microcystin (Enzo Life Sciences, #ALX-350-012-M001; solved in EtOH), pepstatin A (Applichem, #A2205; solved in EtOH), PMSF (Sigma-Aldrich, #P-7626; solved in EtOH), protease inhibitor cocktail tablet (Roche; #11873580001; solved in ddH$_2$O), DTT (Serva; #20710.04; solved in ddH$_2$O), E-64 (Sigma-Aldrich; #324890), doxycycline (Sigma-Aldrich; #D9891;

solved in ddH$_2$O), puromycin (Merck Millipore; #540411; solved in ddH$_2$O), biotin (Sigma-Aldrich; #B4501; solved in DMEM as 20× stock solution and sterile filtrated). Primary antibodies against the following proteins or peptides were used: Anti-β-actin (Santa Cruz; #sc-47778), anti-c-JUN (Invitrogen; #MA5-15889), anti-Phospho-c-JUN Ser63 (Cell Signaling; #9261), anti-HA (Roche; #11583816001), anti-HA (Abcam; #ab9110), anti-IκBα (Cell Signaling; #9242), anti-NF-κB p65 (Santa Cruz; #sc-372; #sc-8008; Bethyl Lab.; #A303-945A; Cell Signaling; #8242), anti-Phospho-NF-κB p65 (Cell Signaling; Ser276 #3037, Ser468 #3039, Ser529 #78764, Ser536 #3033) anti-NF-κB p50 (Santa Cruz; #sc-8414, #sc-114), anti-TFE3 (Sigma-Aldrich; #HPA023881; Protein-tech; #14480-1-AP), anti-TFEB (Cell Signaling; #4240), anti-GLIS2 (Invitrogen; #PA5-40314), anti-RNA-Pol II (Millipore #17-620), anti-Phospho RNA-Pol II Ser5 (Abcam #ab5131) anti-tubulin (Santa Cruz #sc-8035), anti-ZBTB5 (Sigma; #HPA021521), normal rabbit IgG (Cell Signaling #2729). Secondary antibodies: Dylight 488-coupled anti-mouse IgG (ImmunoReagent, #DkxMu-003D488NHSX), HRP-coupled anti-mouse IgG (DakoCytomation, #P0447), HRP-coupled anti-rabbit IgG (DakoCytomation, #P0448), HRP-Streptavidin (PerkinElmer; #NEL750001EA).

## Cloning of pTet-on-Puro-HA-miniTurbo plasmids

For generating pTet-on-Puro-HA-miniTurbo (EV, empty vector), the linker-HA-miniTurbo sequence was synthesized by General Biosystems and provided in donor vector pUC57-Bsal-Free. The donor vector was used to replace the Myc-BirA cassette with the linker-HA-miniTurbo insert in the target plasmid pTet-on-Puro-Myc-BirA using a restriction digestion reaction with FastDigest MluI and BshTI (AgeI) and subsequent ligation by T4 DNA ligase. The PCR for cloning p65 gene variants into the pTet-on-Puro-HA-miniTurbo (EV) vector was based on the donor vectors pEF-Puro-hu p65 WT-HA, p65 E/I-HA, and p65 FL/DD-HA (Riedlinger et al, 2019). Amplicons with restriction site overhangs (BshTI and Bsp1407I) were generated with Phusion™ high-fidelity DNA polymerase (Thermo Fisher Scientific; #F-530XL) using a 3-step PCR program (98 °C for 30 s, 35 × (98 °C for 10 s, 69 °C for 30 s, 72 °C for 25 s) followed by 72 °C for 10 min, 4 °C hold). The resulting PCR amplicons as well as the target vector pTet-on-Puro-HA-miniTurbo (EV) were digested by FastDigest BshTI (AgeI) and Bsp1407I (BsrGI) and ligated. All PCR or vector digestion products were purified using the NucleoSpin® Gel and PCR Clean-Up Kit (Macherey-Nagel; #740609.50). The final plasmids were transformed into competent E.coli XL-1-Blue, extracted by using the NucleoSpin® Plasmid or NucleoBond® Xtra Midi Kit (Macherey-Nagel; #740588.250 and #740410.50) and controlled by Sanger Sequencing (Eurofins Genomics or Seqlab Microsynth).

## Cloning and transduction of lentiviral vectors

Single guide (sg) RNAs were prepared for cloning into the lentiCrisprV2 vector (Addgene; #52961) as follows. Oligonucleotides were heated for 5 min at 95 °C, slowly cooled down to 25 °C and phosphorylated by T4-Polynukleotid-Kinase (PNK) in PNK buffer A (Thermo Fisher Scientific; #EK0031) plus ATP (Thermo Fisher Scientific; #R0441). Vectors were digested and ligated simultaneously with the annealed oligos by a FastDigest reaction using Esp3l (Thermo Fisher Scientific; #FD0454) and T4 DNA ligase (Invitrogen; #15224041). The ligated plasmid DNA was transformed into NEB® Stable Competent E. coli bacteria (NEB #C3040H). Plasmids were subsequently isolated and controlled for the correct insertion of the sgRNA by Sanger sequencing with the hU6-fwd primer (GAGGGCCTATTTCCCATGATT). For lenti-virus production, the cloned sgRNA transfer vectors were transfected into HEK293FT cells in Opti-MEM™ (Gibco; #31985070) together with packaging plasmids pMD2.G and psPAX2 (Addgene; #12259 and #12260) using Lipofectamine LTX Plus Reagent (Thermo Fisher Scientific; #15338100). For one T175 flask, 4.5 ml Opti-MEM, 15.3 µg pMD2.G, 23.4 µg psPAX2 and 30.6 µg transfer plasmid were mixed with 279 µl Plus Reagent and incubated for 5 min at room temperature. In all, 4.5 ml Opti-MEM supplemented with 270 µl Lipofectamine were added, incubated for 5 min and evenly distributed on the cells. Cells were then returned to the incubator for 4 h until the medium was replaced. Forty-eight hours post transfection, the virus-containing supernatant was collected in a 50-ml tube, cleared of cell debris by centrifugation (5 min, 500×$g$), subsequently filtered (0.45-µm sterile syringe filters), and either stored at −80 °C or used directly for target cell transduction. To generate stable knockout cells, parental HeLa cells were seeded in six-well plates (1.8 × 10$^5$ cells) and transduced with lentiviral particles on the following day. 8 µg/ml polybrene was added to each 6-well and incubated for 10 min at 37 °C. In all, 0.5–1 ml of the desired lentiviral stock was supplemented to the well and mixed by gently shaking the culture vessel. Cells were returned to the incubator and transduced overnight (24 h). Cells were expanded to 100-mm dishes and transduced cells were selected by puromycin (0.75 µg/ml). After 10–14 days, confluent cells were harvested for further analysis.

## Transfection of cells with branched polyethyleneimine (PEI)

We optimized the transfection conditions of the miniTurbo constructs as follows. For transient transfection of cells with expression vectors, branched polyethyleneimine (PEI; Sigma-Aldrich; #408727) was used. For T145 cell culture dishes, 1 ml of pre-warmed Opti-MEM™ (serum-reduced medium; Gibco; #31985070) was mixed with 50 µg plasmid DNA and 120 µl ice-cold branched PEI (1 mg/ml, pH 7.0), vortexed, and incubated for 10 min at room temperature. DMEM supplemented with 10% FBS or Tet-free FBS (w/o Pen./Strep.) was added to the mixture (filled-up to 20 ml), vortexed and carefully spread over the cell layer (~70% confluency) after the culture medium was aspirated. Cells were further incubated overnight (24 h). For other cell culture dish sizes, the volumes were adjusted accordingly.

## miniTurboID proximity labeling and purification

For each experimental condition, 5 × 10$^5$ parental HeLa and HeLa Δp65 cells were seeded in a T145 cell culture dish and grown for 4 days. On day three, cells were transfected with pTet-On-Puro expression vectors encoding HA-miniTurbo (EV), p65(wt)-HA-miniTurbo, p65(E/I)-HA-miniTurbo or p65(FL/DD)-HA-miniTurbo using branched PEI. Following transfection, cDNA expression was induced with 1 µg/ml doxycycline for 17 h. On the next day (24 h post transfection), the medium was supplemented with 50 µM exogenous biotin (Sigma-Aldrich; #B4501) 10 min prior to IL-1α treatment

(10 ng/ml for 1 h) enabling biotinylation of p65 interacting proteins during inflammatory cytokine treatment and the final 70 min prior to harvesting, respectively. Thereafter, cells were washed with PBS and harvested on ice by scraping and centrifugation (900×g/4 °C/5 min). Cell pellets were resuspended in 475 µl Tris/HCl (50 mM; pH 7.5) and 50 µl Triton X-100 (10% w/v). Cells were lysed by the addition of 250 µl lysis buffer (50 mM Tris/HCl, 500 mM NaCl, 2% SDS (w/v), including freshly added 1 mM DTT and 1× Roche inhibitor cocktail) followed by incubation on ice for 10 min. Cells were sonicated (settings: 3–4 ×30 s on/30 s off, 4 °C, power high; Bioruptor, Diagenode) and lysates were cleared by centrifugation at 16,000×g at 4 °C for 15 min. For validation of cDNA expression and induced biotinylation, 1% of the lysates (input, 7 µl) were analyzed by SDS-Page and immunoblotting. For the pulldown of biotinylated proteins, 700 µl of the lysates were added to 60 µl of streptavidin-agarose beads (Thermo Scientific; #20349), equilibrated in lysis buffer and rotated end over end overnight (16-18 h) at 4 °C. Beads were collected by centrifugation at 1000×g for 2 min and were washed once with 0.5 ml wash buffer I (2% w/v SDS), twice with 0.5 ml wash buffer II (50 mM HEPES, 0.5 M NaCl, 1 mM EDTA, 0.1% w/v sodium deoxycholate, 1% v/v Triton X-100; pH 7.5), twice with 0.5 ml wash buffer III (10 mM Tris/HCl, 1 mM EDTA, 250 mM LiCl, 0.5% w/v sodium deoxycholate, 0.5% w/v NP-40; pH 7.4), twice with 0.5 ml wash buffer IV (50 mM Tris/HCl, 50 mM NaCl, 0.1% v/v NP-40; pH 7.4) and once with 0.5 ml wash buffer V (50 mM Tris/HCl; pH 7.5). The beads were resuspended in 1 ml buffer V, of which 80% were used for mass spectrometry analysis. For the validation of affinity purifications, the remaining 20% of the beads (supplemented with 40 µl of 2× ROTI®Load and boiled at 95 °C for 10 min) were subjected together with 1% input samples to SDS-PAGE and proteins and modifications were detected by immunoblotting using HRP-streptavidin conjugate (PerkinElmer; #NEL750001EA) or anti-p65 (Santa Cruz Biotechnology; #sc-8008), anti-HA (Roche; #11583816001), anti-β-actin (Santa Cruz Biotechnology; #sc-4778) antibodies.

## Mass spectrometry analysis of miniTurboID and bioinformatics

Samples bound to streptavidin-agarose beads were washed three times with 100 µl 0.1 M ammonium bicarbonate solution. Proteins were digested "on-bead" by the addition of sequencing grade modified trypsin (Serva) and incubated at 37 °C for 45 min. Subsequently, the supernatant was transferred to fresh tubes and incubated at 37 °C overnight. Peptides were desalted and concentrated using Chromabond C18WP spin columns (Macherey-Nagel; #730522). Finally, Peptides were dissolved in 25 µl of water with 5% acetonitrile and 0.1% formic acid. The mass spectrometric analysis of the samples was performed using a timsTOF Pro mass spectrometer (Bruker Daltonics). A nanoElute HPLC system (Bruker Daltonics), equipped with an Aurora C18 RP column (25 cm × 75 µm) filled with 1.7-µm beads (IonOptics) was connected online to the mass spectrometer. A portion of approximately 200 ng of peptides corresponding to 2 µl was injected directly on the separation column. Sample Loading was performed at a constant pressure of 800 bar. Separation of the tryptic peptides was achieved at 50 °C column temperature with the following gradient of water/0.1% formic acid (solvent A) and acetonitrile/0.1% formic acid (solvent B) at a flow rate of 400 nl/min: Linear increase from 2% B to 17% B within 60 min, followed

by a linear gradient to 25% B within 30 min and linear increase to 37% B for an additional 10 min. Finally, B was increased to 95% within 10 min and held for additional 10 min. The built-in "DDA PASEF-standard_1.1sec_cycletime" method developed by Bruker Daltonics was used for mass spectrometric measurement. Data analysis was performed using MaxQuant (version 1.6.17.0) with the Andromeda search engine. All amino acid sequences of the Uniprot database (Uniprot Human reviewed proteins, database version 2021_03) were used for annotating and assigning protein identifiers (Tyanova et al, 2016a). For the detection of phosphorylated peptides, the same raw data were processed again with MaxQuant (version 2.3.0.0) using the same settings as before plus an additional search for Phospho (STY)Sites as provided by MaxQuant. The phosphosite results were extracted from the MaxQuant output files "modificationSpecificPeptides.txt", "Phospho (STY)Sites.txt" and "proteinGroups.txt". Perseus software (version 1.6.14) was used for further analyses of protein intensity values (Tyanova et al, 2016b). For the calculation of ratio values between conditions, biological and technical replicates from each condition were assigned to one analysis group using tools for categorical annotation rows. All values were $Log_2$-transformed and missing values were imputed using a $log_2$ intensity value of 9, which was below the lowest intensity value measured across all samples. No further normalization of the pulldown experiments was performed to preserve the anticipated differences between samples. Enriched proteins between pairwise comparisons were identified by Student's t tests using Perseus functions and were visualized by Volcano plots. Interesting groups of enriched p65/ RELA interactors were defined by $log_2$ fold change (LFC) and statistical significance of changes based on $-Log_{10}$ P values ≥ 1.3 as indicated in the legends. Subsequent filtering steps and heatmap visualizations were performed in Excel 2016 according to the criteria described in the figure legends. Venn diagrams were created with tools provided at http://bioinformatics.psb.ugent.be/webtools/Venn/. Overrepresentation analyses of gene sets were done using the majority protein IDs or gene IDs of differentially enriched proteins or mRNAs uploaded to Metascape software and processed with the predefined express settings (Zhou et al, 2019). Protein networks were inferred from filtered gene ID lists using information of the most current version of the STRING database (https://string-db.org/), and networks were visualized and annotated with enriched pathway terms using Cytoscape, version 3.8.0 or higher and the STRING plugin (Shannon et al, 2003; Szklarczyk et al, 2019).

## Transfection of cells with siRNAs

Cells were seeded in 60-mm cell culture dishes and grown overnight. The medium was reduced to 3 ml and a transfection mixture was prepared as follows: 187.5 µl Opti-MEM™, 15 µl Hi-PerFect Transfection Reagent (Qiagen; #301705) and 60 µl of siRNA mixture (1 µM, final concentration 20 nM). The reaction mixture was vortexed and incubated for 10 min at room temperature, subsequently dripped on the culture dish, and gently mixed. After 6 h of incubation, the transfection mixture was aspirated and replaced by 4 ml fresh complete DMEM medium. Cells were then incubated for 48 h until they were further processed. The following FlexiTube GeneSolution siRNAs (Qiagen) were used: RELA/p65 (#GS5970), TFE3 (#GS7030), TFEB (#GS7942), GLIS2 (#GS84662), ZBTB5 (#GS9925), S100A8

(#GS6279), S100A9 (#GS6280). As a non-targeted control, siRNA against *Firefly luciferase* was synthesized (Eurofins Genomics).

## mRNA expression analysis by RT-qPCR

In total, 1 µg of total RNA was prepared by column purification using the NucleoSpin® RNA Kit (Macherey-Nagel; #740955.250) and transcribed into cDNA using 0.5 µl RevertAid Reverse Transcriptase (Fisher Scientific #EP0441), 4 µl 5× reaction buffer, 0.5 µl Random Hexamer Primer, 0.5 mM dNTP mix (10 mM) in a total volume of 20 µl at 25 °C for 10 min, 42 °C for 1 h and 70 °C for 10 min. 1 µl of the reaction mixture was used to amplify cDNA using Taqman® Gene Expression Assays (0.25 µl, (Applied Biosystems) for *ACTB* (#Hs99999903_m1), *GUSB* (#Hs99999908_m1), *GAPDH* (#Hs02758991_g1), *IL8* (#Hs00174103_m1), *NFKBIA* (#Hs00153283_m1), *CXCL2* (#Hs00236966_m1), *RELA* (#Hs01042019_g1) and TaqMan® Fast Universal PCR Master Mix (Applied Biosystems; #4352042). All PCRs were performed as duplicate reactions on an ABI7500 Fast real-time PCR instrument in a total volume of 10 µl. PCR cycles were as follows: 95 °C (20 s), 40× (95 °C (3 s), 60 °C (30 s). The cycle threshold value (ct) for each individual PCR product was calculated by the instrument's software and Ct values obtained for inflammatory/target mRNAs were normalized by subtracting the Ct values obtained for *GUSB*, *ACTB* or *GAPDH*. The resulting ΔCt values were then used to calculate relative fold changes of mRNA expression according to the following equation: $2^{-((\Delta ct\ stim.)-(\Delta ct\ unst.))}$ or $2^{-((\Delta ct\ siRNA\ target.)-(\Delta ct\ siRNA\ Luciferase))}$.

## Targeted siRNA screen

For the siRNA screens, cDNA was synthesized in cell lysates and amplified using the TaqMan® PreAmp Cells-to-Ct Kit™ (Applied Biosystems; #4387299) and TaqMan® Gene Expression Assays (Applied Biosystems) following an adapted miniaturized protocol. The kit enables to perform gene expression analysis directly from small numbers of cultured cells without RNA purification by an intermediate pre-amplification step between reverse transcription and qPCR. In all, $3 \times 10^3$ HeLa cells were seeded in 48-well plates and cultured overnight (24 h). Transfection occurred as described above with downscaled reagent volumes as follows: 12.5 µl Opti-MEM™, 1 µl Hi-PerFect Transfection Reagent and 4 µl of siRNA mixture (1 µM, finally 20 nM). 38 HCI and RELA were targeted by pools of 3–4 gene-specific siRNAs and compared to siLuciferase, Hi-PerFect only (HP), or untreated control samples. Forty-eight hours after transfection, half of the cells per plate were treated for 1 h with IL-1α (10 ng/ml). Cells were harvested by trypsinization and transferred to RNase-free reaction tubes on ice. Cells were washed twice with ice-cold PBS and lysed in 12.5 µl lysis solution (DNase I was diluted at 1:100). After vortexing, the lysates were incubated for 5 min at room temperature. The reaction was stopped by adding 1.25 µl stop solution. After repeated mixing, samples were incubated for 2 min at room temperature. The reverse transcription was directly conducted on the lysates by mixing 4.5 µl lysate (or nuclease-free water as a control) with a 5.5 µl RT mixture that was prepared as follows: 5 µl of 2× RT-buffer and 0.5 µl of 20× RT enzyme mix. The reaction tubes were incubated in a thermal cycler at 37 °C for 60 min, followed by 95 °C for 5 min to inactivate the RT enzyme. Next, the cDNA was pre-amplified using

gene-specific primers contained in TaqMan® Gene Expression Assays. The assays of interest were diluted 1:100 in TE buffer. For this purpose, two pools of assays were prepared: pool 1 for target genes 1–19 and pool 2 for target genes 20–38. Both pools were additionally supplemented with assays for three prototypical NF-κB target genes *IL8* (#Hs00174103_m1), *NFKBIA* (#Hs00153283_m1) and *CXCL2* (#Hs00236966_m1), two housekeeping genes *GUSB* (#Hs99999908_m1) and *GAPDH* (#Hs02758991_g1) and the positive control *RELA* (#Hs01042019_g1). The pre-amplification PCR mixtures of pool 1 or pool 2 were prepared as follows: 2.5 µl pool 1 or pool 2 were used in a 10 µl reaction volume with 5 µl 2×TaqMan PreAmp MasterMix and 2.5 µl cDNA. Samples with siRNA targets 1–19 were supplemented with the mixture of pool 1 and samples with siRNA targets 20–38 were supplemented with the mixture of pool 2. Controls were supplemented with each of the two pre-amplification mixtures. The pre-amplification reaction was performed in a thermal cycler at 95 °C for 10 min, followed by 15 cycles at 95 °C for 15 s/60 °C for 4 min. Prior to real-time PCR, the pre-amplification products were diluted 1:5 with TE buffer. The expression of the indicated target genes was determined by real-time PCR using the TaqMan® Fast universal PCR master mix and 7500 Fast Real-Time PCR System from Applied Biosystems. Based on Ct values, mRNA levels were quantified and normalized against *GUSB*. The effects of knockdowns were calculated separately for basal and IL-1α-inducible conditions against the luciferase siRNA.

## Microarray transcriptomics

HeLa cells were transiently transfected for 48 h with 20 nM siRNA mixtures against *RELA*, *ZBTB5*, *S100A8*, *S100A9* (series 1) or *RELA*, *GLIS2*, *TFE3*, *TFEB* (series 2) and a siRNA against *luciferase* (siLuc) as control as described above. Half of the cells were treated with IL-1α (10 ng/ml) for 1 h at the end of the incubation, and Agilent microarray analyses were performed from isolated total RNA using the NucleoSpin® RNA Kit (Macherey-Nagel; #740955.250). Per reaction, 200 ng RNA was amplified and Cy3-labeled using the LIRAK kit (Agilent; #5190-2305) following the kit instructions. The Cy3-labeled cRNA was hybridized overnight to 8× 60 K 60-mer oligonucleotide spotted microarray slides (Agilent Technologies; # G4851C design ID 072363). Hybridization and subsequent washing and drying of the slides were performed following the Agilent hybridization protocol. The dried slides were scanned at 2 µm/pixel resolution using the InnoScan is900 (Innopsys, Carbonne, France). Image analysis was performed with Mapix 9.0.0 software, and calculated values for all spots were saved as GenePix results files. Stored data were evaluated using the R software and the limma package from BioConductor. Mean spot signals were background-corrected with an offset of 1 using the NormExp procedure on the negative control spots. The logarithms of the background-corrected values were quantile-normalized. The normalized values were then averaged for replicate spots per array. From different probes addressing the same NCBI gene ID, the probe showing the maximum average signal intensity over the samples was used in subsequent analyses. Genes were ranked for differential expression using a moderated t-statistic. Pathway analyses were done using gene set tests on the ranks of the *t* values. Pathway annotations were obtained from KEGG (Kanehisa et al, 2016). The genes assigned to these annotations, including the signal intensity values (*E* values) and the differential expression between samples (Log$_2$

fold change, LFC) with the associated significance ($-\text{Log}_{10} P$ value) were listed in an Excel file and used for further filtering steps as mentioned in the figure legends.

## Preparation of (sub)cellular extracts, co-immunoprecipitation, and immunoblotting

For whole-cell extracts, cells were lysed in Triton cell lysis buffer (10 mM Tris, pH 7.05, 30 mM NaPPi, 50 mM NaCl, 1% Triton X-100, 2 mM Na$_3$VO$_4$, 50 mM NaF, 20 mM ß-glycerophosphate and freshly added 0.5 mM PMSF, 2.5 µg/ml leupeptin, 1.0 µg/ml pepstatin, 1 µM microcystin) and incubated for 15 min on ice. Lysates were cleared by centrifugation at 10,000×$g$/4 °C/15 min.

For preparation of nuclear and cytosolic extracts, cells were suspended and pelleted (800×$g$ at 4 °C for 5 min) in buffer A (10 mM HEPES, pH 7.9, 10 mM KCl, 1.5 mM MgCl$_2$, 0.3 mM Na$_3$VO$_4$, 20 mM β-glycerophosphate, freshly added 200 µM leupeptin, 10 µM E-64, 300 µM PMSF, 0.5 µg/ml pepstatin, 5 mM DTT and 1 µM microcystin). The pellet was resuspended in buffer A containing 0.1% NP-40 and incubated for 10 min on ice. After centrifugation at 10,000×$g$ for 5 min at 4 °C, supernatants were taken as cytosolic extracts. Nuclear pellets were resuspended in buffer B (20 mM Hepes, pH 7.9, 420 mM NaCl, 1.5 mM MgCl$_2$, 0.2 mM EDTA, 25% glycerol, 0.3 mM Na$_3$VO$_4$, 20 mM β-glycerophosphate, freshly added 200 µM leupeptin, 10 µM E-64, 300 µM PMSF, 0.5 µg/ml pepstatin, 5 mM DTT, and 1 µM microcystin). After 30 min on ice, nuclear extracts were cleared at 10,000×$g$ for 5 min at 4 °C, and supernatants were collected.

For the analysis of phosphorylations, whole-cell extracts were prepared in freshly made urea buffer as follows. Cells were lysed in 20 mM HEPES (pH 8.0), 9.0 M urea, 1 mM activated sodium orthovanadate, 2.5 mM sodium pyrophosphate, and 1 mM ß-glycerophosphate for 10 min on ice and sonified for five cycles with high power for 30 s on and 30 s off at 4 °C in the Bioruptor® Plus sonication device (Diagenode). Lysates were cleared by centrifugation (16,000×$g$ at 4 °C for 15 min).

For the preparation of cytosolic, soluble nuclear, and chromatin extracts, cells were washed, resuspended, and pelleted (500×$g$/4 °C/5 min) in PBS. Pellets were suspended in lysis buffer I (20 mM HEPES (pH 8.0), 10 mM KCl, 1 mM MgCl$_2$, 0,1% (v/v) Triton X-100, 20% (v/v) glycerol, freshly added 50 mM NaF, 1 µM microcystin, 1 mM Na$_3$VO$_4$, 1× Roche protease inhibitor cocktail) on ice for 10 min. After centrifugation for 1 min at 2300×$g$ at 4 °C, supernatants were taken as cytosolic extracts (C). The pellet was resuspended in lysis buffer II (20 mM HEPES, 2 mM EDTA, 400 mM NaCl, 0,1% (v/v) Triton X-100, 20% (v/v) glycerol, freshly added 50 mM NaF, 1 µM microcystin, 1 mM Na$_3$VO$_4$, 1× Roche protease inhibitor cocktail), incubated on ice for 20 min, and briefly mixed twice during this time. The nuclear soluble fraction (N1) was separated into the supernatant by centrifugation at 20,400×$g$ and 4 °C for 5 min. The remaining pellet was resuspended in lysis buffer III (20 mM Tris (pH 7.5), 2 mM EDTA, 150 mM NaCl, 1% (w/v) SDS, 1% (w/v) NP-40, freshly added 50 mM NaF, 1 µM microcystin, 1 mM Na$_3$VO$_4$, 1× Roche protease inhibitor cocktail) and sonicated (6 cycles with high power for 30 s on and 30 s off at 4 °C in a Bioruptor® Plus sonication device (Diagenode). This was followed by incubation for 30 min on ice and centrifugation at 20,400×$g$ and 4 °C for 5 min. The supernatant contained the chromatin-bound nuclear fraction (N2).

Protein concentrations of all cell extracts were determined by the Bradford method (Carl Roth; ROTI®Quant; #K929.3), and ~20–50 µg protein per sample was supplemented with reducing gel-loading buffer 4× RotiLoad (Carl Roth; #K929.3). Immunoblotting was performed essentially as previously described (Hoffmann et al, 2005). Proteins were separated on SDS-PAGE and electrophoretically transferred to PVDF membranes (Roti-PVDF (0.45 µm); Carl Roth; #T830.1). After blocking with 5% dried milk in Tris/HCl-buffered saline/0.05% Tween (TBS-T) for 1 h, membranes were incubated for 12–24 h with primary antibodies (diluted 1:500–1:10,000 in 5% milk or BSA in TBS-T), washed in TBS-T and incubated for 1–2 h with the peroxidase-coupled secondary antibodies (HRP-coupled anti-rabbit IgG (Dako, #P0448), HRP-coupled anti-mouse IgG (Dako; #P0447). After washing in TBS-T, proteins were detected by using enhanced chemiluminescence (ECL) systems from Merck Millipore (Immobilon Western Chemiluminescent HRP Substrate; #WBKLS0500) or GE Healthcare (Amersham ECL Western Blotting Detection Reagent; #RPN2106). Images were acquired and quantified using the ChemiDoc TouchImaging System (Bio-Rad) and the software ImageLab versions 5.2.1 or 6.1.0 (Bio-Rad). For visualization of biotinylated proteins, membranes were blocked with 5% BSA/TBS-T for 24 h at 4 °C and afterward membranes were incubated for 1 h with HRP-Streptavidin (PerkinElmer; #NEL750001EA; diluted 1:5000 in 5% BSA/TBS-T).

Co-immunoprecipitation was performed using 500 µg of whole-cell extracts, 15 µl TrueBlot® Anti-Mouse Ig IP Agarose Beads (Rockland, #00-8811-25) and 1 µg of anti-NF-κB p65 (ms, sc-8008) or mouse IgG (ms, sc-2025). For immunoprecipitation, the cell extracts were incubated with antibody-coupled beads (rotating 2 h at 4 °C) in total volume of 500 µl of complete lysis buffer overnight at 4 °C on a rotating device. After centrifugation at 2500×$g$ at 4 °C for 1 min, the supernatant was discarded and the beads were washed three times with lysis buffer. The precipitated proteins were eluted by boiling the beads in 2× Roti-Load for 10 min before analyzing them by western blotting using TrueBlot® ULTRA Anti-Mouse Ig HRP (Rockland; #18-8817-30).

## Immunofluorescence coupled to in situ proximity ligation assay (Immuno-PLA)

Immunofluorescence (IF) was coupled to in situ proximity ligation assay (Immuno-PLA). For PLA, the Duolink® PLA reagents were used (Sigma-Aldrich; #DUO92007, #DUO92006, #DUO92002). In total, 9000 cells per channel, of parental or p65-deficient cells, were seeded in ibiTreat µ-Slides VI 0.4 (Ibidi; #80606). On the next day, cells were washed twice with 150 µl PBS for 5 min and fixed with 100 µl of 4% paraformaldehyde (in PBS, Santa Cruz, #281692) for 10 min at room temperature. Afterward, 100 µl 0.1 M Tris/HCl, pH 7.4, were added and incubated for 10 min at room temperature. Cell permeabilization was performed by adding 100 µl of a 0.005% saponin/0.1% Triton X-100/PBS solution for 10 min at room temperature. Permeabilized cells were washed twice with 150 µl PBS for 5 min and incubated overnight (24 h) with 40% glycerol/PBS at room temperature. On the next day, nuclei were permeabilized by three cycles of freeze-and-thaw. Specifically, Ibidi-slides were kept in liquid nitrogen for 1 min and thawed until glycerol cleared up. After nuclear permeabilization, cells were washed twice with PBS for 5 min and embedded in 100 µl blocking solution which was incubated in a humidity chamber for 30 min at 37 °C. Protein–protein interaction analyses for p50 were performed

with cells seeded with the same density as described above, but samples were only washed twice with HBSS and fixed with 4% paraformaldehyde (in PBS, Santa Cruz, #281692) for 5 min. Afterward, cells were washed and permeabilized two-times with 100 μl of a 0.005% saponin/HBSS solution for 5 min at room temperature, followed by a blocking step as described before.

The blocking solution was discarded and cells were incubated with 50 μl of the appropriate primary antibody mixture in a humidity chamber for 1 h at 37 °C. The primary antibody mixture contained PLA and IF antibodies (anti-NF-κB p65 (Santa Cruz Biotechnology; #sc-8008, diluted 1:100, ms and Bethyl Lab.; #A303-945A, diluted 1:1000, gt), anti-TFE3 (Sigma-Aldrich; #HPA023881, diluted 1:50, rb), anti-TFEB (Cell Signaling; #4240, diluted 1:25, rb), anti-GLIS2 (Thermo Fisher Scientific; #PA5-40314, diluted 1:25, rb), anti-ZBTB5 (Sigma, #HPA021521, diluted 1:150, rb) and anti-p50 (Santa Cruz Biotechnology; #sc-8414, diluted 1:100, ms), which were diluted in antibody diluent. Cells were washed three times with 150 μl buffer A for 5 min and then incubated with 50 μl of a secondary antibody mixture in a humidity chamber for 1 h at 37 °C. The secondary antibody mixture contained PLA probes which were diluted 1:5 in antibody diluent and, if indicated, the Dylight 488-coupled secondary anti ms IF antibody. From the time of incubation, all further steps were carried out in the dark. Cells were washed three times with 150 μl buffer A for 5 min and incubated with 50 μl ligase solution in a humidity chamber for 30 min at 37 °C. The ligase was diluted 1:60 in ligase solution by diluting the stock solution 1:5 in nuclease-free water). Cells were washed three times with 150 μl buffer A for 2 min and then incubated with 50 μl polymerase solution in a humidity chamber for 100 min at 37 °C. The polymerase was diluted 1:80 in amplification solution by diluting the stock solution 1:5 in nuclease-free water). Cells were first washed three times with 150 μl buffer B for 5 min followed by two washes with HBSS for 1 min. Nuclear DNA was stained with 1 μM Hoechst 33342 for 5 min at room temperature and washed twice with 150 μl HBSS for 5 min. Cells were finally embedded in 50 μl 30% glycerol/HBSS and stored in the dark until microscopic documentation. Fluorescence imaging of Immuno-PLA samples was carried out using the inverse fluorescence microscope THUNDER imager DMi8 (Leica Microsystems CMS GmbH; HC PL APO 20×/0.8 DRY objective, camera Leica-DFC9000GT-VSC13705) which was equipped with filter cubes suited for Hoechst 33342 (excitation 391/32 and emission 435/30) and DyLight488 (excitation 506/21 and emission 539/24) and with the Leica LASX software (version 3.7.4.23463). The obtained fluorescence signals of the PLA analyses were detected with the inverse microscope DMi8 (Leica) using filter cubes suited for DyLight488 (excitation 480/40 and emission 527/30), Cy5 (excitation 620/60 and emission 700/75) and Leica LASX software (version 1.5.1.13187).The Quantification of protein–protein interaction (PPI) complexes was performed using the Blobfinder software from the Centre for Image Analysis (Uppsala University, Sweden) and Olink Bioscience (Allalou and Wahlby, 2009). The software configuration was adjusted to HeLa cells with a nucleus size of 100 pixels$^2$ and cytoplasm size of 100 pixels. For the blob size, the $3 \times 3$ default setting was applied whereas the blob threshold of 15 was determined by a test image. Alternatively, images were segmented using Cellpose (version 2.2.3) in combination with the analyzing software CellProfiler (version 4.2.5, Broad Institute). To segment the whole cell, phase contrast channel

images were used to train our own Cellpose model starting from the built-in model "cyto2". Similarly, we used the Cellpose model "nuclei", customized it to our own nuclei and segmented them in the channel stained with Hoechst 33342. The output of cellular and nuclear masks were further processed in our own CellProfiler pipeline, wherein these masks were identified as objects in which the PLA spots were counted per cell.

## Chromatin immunoprecipitation (ChIP)

In total, $1.25\text{–}2.5 \times 10^7$ KB cells were seeded in T175 cell culture flask per condition. $1.8\text{–}3.5 \times 10^6$ HeLa lentiCRISPRV2 cell lines were seeded in 2× T145-mm cell culture dishes per condition. At the next day cells were starved for 24 h in HBSS, left untreated or stimulated with IL-1α (10 ng/ml) for 1 h. For combined treatments, cells were stimulated with IL-1α (10 ng/ml) after 23 h of starvation for 1 h or left unstimulated. Proteins bound to DNA were cross-linked in vivo with 1% formaldehyde added directly to the medium. After 10 min incubation at room temperature, 0.1 M glycine was added for 5 min to stop cross-linking. Then, cells were collected by scraping and centrifugation at 1610×$g$ (5 min, 4 °C), washed in cold PBS containing 1 mM PMSF and centrifuged again. KB cells were lysed for 10 min on ice in 3 ml ChIP lysis buffer (1% SDS, 10 mM EDTA, 50 mM Tris pH 8.1, 1 mM PMSF, 1,5× Roche protease inhibitor mix). The DNA was sheared by sonication (4 ×7 cycles, 30 s on/30 s off, power high; Bioruptor, Diagenode) at 4 °C and lysates were cleared by centrifugation at 16,300×$g$ at 4 °C for 15 min. HeLa lentiCRISPRV2 cell lines were lysed for 10 min on ice in 1.5 ml ChIP lysis buffer (1% SDS, 10 mM EDTA, 50 mM Tris pH 8.1, 1 mM PMSF, 1,5× Roche protease inhibitor mix) and centrifuged (845×$g$, 4 °C, 7 min). The pellets were resuspended in 900 μl shearing buffer (0.1% SDS, 1 mM EDTA, 10 mM Tris/HCl pH 7.6, 1 mM PMSF, 1,5× Roche protease inhibitor mix. The DNA was sheared in 1 ml milliTUBE AFA Fiber (Covaris #520135) by focused-ultrasonicator Covaris S220x (Peak Power 140.0, Duty Factor 5.0, Cycles/Burst 200, Duration 15 min, 4 °C) and lysates were cleared by centrifugation at 16,300×$g$ at 4 °C for 15 min. Supernatants from cleared lysates were collected and stored in aliquots at −80 °C. For determination of DNA concentration, 20 μl of sheared lysate was diluted with 100 μl TE buffer including 10 μg/ml RNAse A. After 30 min at 37 °C, 3.8 μl proteinase K (20 mg/ml) and 1% SDS was added and incubated for at least 2 h at 37 °C followed by overnight incubation at 65 °C for re-cross-linking. Samples were resuspended in two volumes of buffer NTB (Macherey-Nagel; #740595.150) and DNA was purified using the NucleoSpin® Gel and PCR Clean-Up Kit (Macherey-Nagel; #740609.50) according to the manufacturer's instructions. DNA was eluted with 50 μl 5 mM Tris pH 8.5 and concentration was determined by Nano Drop. For ChIP, the following antibody amounts were used: anti-NF-κB p65 (2–3 μg, Santa Cruz; #sc-372), anti-TFE3 (2 μg, Sigma-Aldrich; #HPA023881), and IgG (2–3 μg, Cell Signaling; #2729). Antibodies were added to precleared lysate volumes equivalent to 15 or 20 μg of chromatin. Then, 900 μl of ChIP dilution buffer (0.01% SDS, 1.1% Triton X-100, 1.2 mM EDTA, 167 mM NaCl, 16.7 mM Tris/HCl, pH 8.1) were added and the samples were rotated at 4 °C overnight. Thereafter, 30 μl of a protein A/G sepharose mixture (GE Healthcare; #17-0780-01 and #17-0618-01), pre-equilibrated in ChIP dilution buffer was added to the lysates and incubation continued for 2 h at 4 °C. Beads were

collected by centrifugation, washed once in ChIP low salt buffer (0.1% SDS, 1% Triton X-100, 2 mM EDTA, 20 mM Tris, pH 8.1, 150 mM NaCl), once in ChIP high salt buffer (0.1% SDS, 1% Triton X-100, 2 mM EDTA, 20 mM Tris pH 8.1, 500 mM NaCl), once in ChIP LiCl buffer (0.25 M LiCl, 1% NP-40, 1% desoxycholate, 1 mM EDTA, 10 mM Tris, pH 8.1) and twice in ChIP TE buffer (10 mM Tris, pH 8.1, 1 mM EDTA) for 5 min at 4 °C end over end rotating. Beads were finally resuspended in 100 μl TE buffer including 10 μg/ml RNAse A. In parallel, 1/10 volume of the initial lysate (10% input samples) were treated accordingly. After 30 min at 37 °C, 3.8 μl proteinase K (20 mg/ml) and 1% SDS were added and both, input and immunoprecipitates were incubated for at least 2 h at 37 °C followed by overnight incubation at 65 °C for re-cross-linking. Samples were resuspended in two volumes of buffer NTB and DNA was purified using the NucleoSpin® Gel and PCR Clean-Up Kit. DNA was eluted with 50 μl 5 mM Tris pH 8.5 and stored at −20 °C until further use. PCR products derived from ChIP-DNA were quantified by real-time PCR using the Fast ABI 7500 instrument (Applied Biosystems). The reaction mixture contained 2 μl to 4 μl of ChIP or input DNA (diluted 1:10 to represent 1% of input DNA), 0.25 μM of specific primers and 10 μl of Fast SYBR™ Green PCR Master Mix (Applied Biosystems; #4385612) in a total volume of 20 μl. PCR cycles were as follows: 95 °C (20 s), 40× (95 °C (3 s), 60 °C (30 s). Melting curve analysis revealed a single PCR product. Calculation of enrichment by immunoprecipitation relative to the signals obtained for 1% input DNA was performed based on the equation % input $= 2^{-(Ct\ IP\ -\ Ct\ 1\%\ input)}$. A list of oligonucleotides is provided in the Reagents & Tools Table.

### Motif analyses of ChIPseq datasets

NF-κB p65 ChIPseq peaks were compiled from four previously described data sets (Data ref:Jurida et al, 2015; Jurida et al, 2015) and were searched for enrichment of position-weight TF matrices using MEME-ChIP (https://meme-suite.org/) (Ma et al, 2014). Overrepresentation of TF motifs within ±500 bp flanking the experimentally determined p65/RELA ChIPseq peaks were calculated against the background of the whole human genome sequence (HG19) and is indicated by p value. The 1 kb windows were further searched for motifs of the TFs RELA, REL, TFE3, TFEB, GLIS2, and various ZBTB factors and predicted binding regions were annotated to genomic features to localize the next adjacent gene. The resulting matrices were filtered to assign motifs to the genes affected by siRNA knockdowns of TFs as indicated in the legends.

### Quantification and statistical analysis

Protein bands detected by Western blotting were quantified using Bio-Rad Image Lab, version 5.2.1 (build 11) or version 6.1.0 (build 7). Statistics (t tests, Mann–Whitney–Rank-Sum test, one-way ANOVA, correlations) were calculated using GraphPadPrism 9.5.1, Perseus 1.6.14.0 or Microsoft Excel 2016.

## Data availability

The proteomic datasets of this study have been submitted to the ProteomeXchange Consortium via the PRIDE partner repository (Perez-Riverol et al, 2022) with the dataset identifier PXD045888. The microarray data sets of this study have been deposited in NCBI's Gene Expression Omnibus (Edgar et al, 2002) and are accessible through GEO Series accession number GSE244637.

The source data of this paper are collected in the following database record: biostudies:S-SCDT-10_1038-S44319-024-00339-8.

## Peer review information

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

## Acknowledgements

This work was supported by the following grants from the Deutsche Forschungsgemeinschaft (DFG, German Research Foundation): SFB1213/3 (B03 (to MK); project 268555672); TRR81/3 (A07 (to MLS), B02 (to MK); project 109546710); KR1143/9-2 (KFO309, P3 (to MK); project 284237345); SFB1021/3 (C02 (to MK), Z03 (to MK and UL); project 197785619); and GRK2573/2 (RP5 (to MK); project 416910386). Work in the laboratories of MK is also supported by the LOEWE program of the state of Hesse (Coropan, P4 (to MK), the IMPRS program of the Max Planck Society and the Excellence Cluster CardioPulmonary Institute (EXC 2026: CardioPulmonary Institute (CPI), project 390649896) and the DZL/UGMLC program.

## Author contributions

**Lisa Leib**: Investigation. **Jana Juli**: Investigation. **Liane Jurida**: Investigation. **Christin Mayr-Buro**: Investigation. **Jasmin Priester**: Investigation. **Hendrik Weiser**: Investigation. **Stefanie Wirth**: Investigation. **Simon Hanel**: Investigation. **Daniel Heylmann**: Methodology. **Axel Weber**: Formal analysis; Investigation. **M Lienhard Schmitz**: Methodology; Writing—review and editing. **Argyris Papantonis**: Writing—review and editing. **Marek Bartkuhn**: Investigation; Writing—review and editing. **Jochen Wilhelm**: Formal analysis; Investigation. **Uwe Linne**: Investigation. **Johanna Meier-Soelch**: Supervision; Validation; Investigation; Writing—original draft; Writing—review and editing. **Michael Kracht**: Conceptualization; Data curation; Formal analysis; Supervision; Funding acquisition; Investigation; Visualization; Writing—original draft; Writing—review and editing. In addition to the CRediT author contributions listed above, the contributions in detail are: LL designed, performed and analyzed TurboID, RNAi and the follow-up validation experiments and prepared graphs and tables; JJ designed, performed and analyzed TurboID, RNAi and the follow-up validation experiments and prepared graphs and tables; LJ performed, analyzed and visualized ChIP-qPCR experiments; CM-B analyzed PLAs and cell death; JP analyzed phospho-proteins and cell death; HW performed protein analyses; SW performed protein analyses; SH analyzed PLAs; DH helped with cell selection experiments; AW processed LC-MS/MS raw data; MLS helped with p65-HA-mTb cloning; AP edited the first version of the manuscript; MB re-analyzed ChIPseq data from LJ; JW performed microarray experiments and processed raw data; UL performed LC-MS/MS analyses; JM-S assembled the method section, performed and analyzed the experiments for the revision; MK conceived the study, performed bioinformatics analyses, prepared figures and tables and wrote the initial draft. Source data underlying figure panels in this paper may have individual authorship assigned. Where available, figure panel/source data authorship is listed in the following database record: biostudies:S-SCDT-10_1038-S44319-024-00339-8.

## Funding

## Disclosure and competing interests statement

The authors declare no competing interests.

# Expanded View Figures

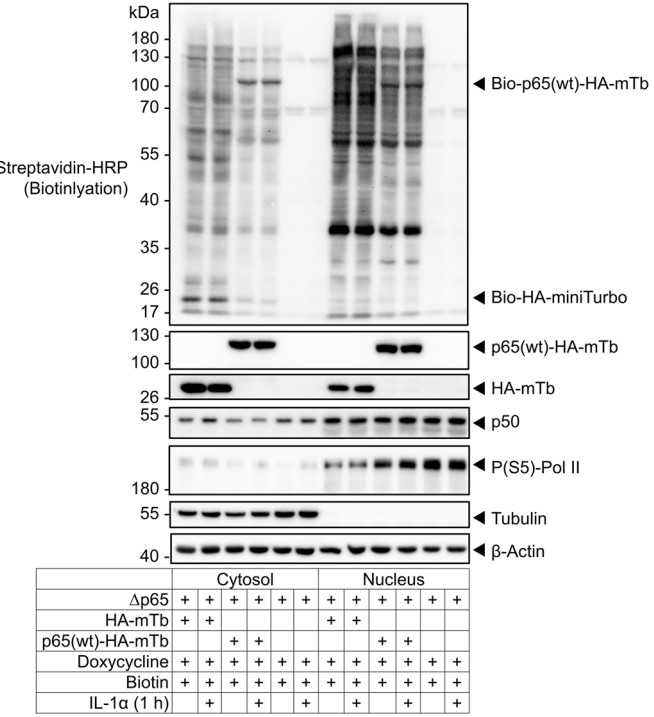

| | Cytosol | | | | | | Nucleus | | | | | |
|---|---|---|---|---|---|---|---|---|---|---|---|---|
| Δp65 | + | + | + | + | + | + | + | + | + | + | + | + |
| HA-mTb | + | + | | | | | + | + | | | | |
| p65(wt)-HA-mTb | | | + | + | | | | | + | + | | |
| Doxycycline | + | + | + | + | + | + | + | + | + | + | + | + |
| Biotin | + | + | + | + | + | + | + | + | + | + | + | + |
| IL-1α (1 h) | | + | | + | | + | | + | | + | | + |

**Figure EV1. Efficient biotinylation of cytosolic and nuclear proteins by HA-mTb and p65-HA-mTb.**

Pools of HeLa cells with CRISPR/Cas9-based suppression of endogenous p65/RELA (Δp65) were transiently transfected (using branched polyethyleneimine, PEI) with plasmids encoding HA-miniTurbo (empty vector, EV) or p65(wt) -HA-mTb or were left untransfected. Expression of HA-mTb or p65-HA-mTb(wt) was induced with doxycycline (1 μg/ml) for 17 h. Intracellular biotinylation was induced by the addition of 50 μM biotin for further 60 min during which time half of the samples were additionally treated with IL-1α (10 ng/ml). Cytosolic and nuclear fractions were prepared from cell lysates and proteins were analyzed by Western blotting for the subcellular expression of HA-miniTurbo, p65-HA-miniTurbo or the endogenous p50 NF-κB subunit using anti-HA, anti-p65 and anti-p50 antibodies, respectively. Antibodies against tubulin and P-Pol II were used to control separation of cell fractions. Equal loading of fractions was confirmed by probing the blots with anti β-actin antibodies. Shown is one representative out of two experiments.

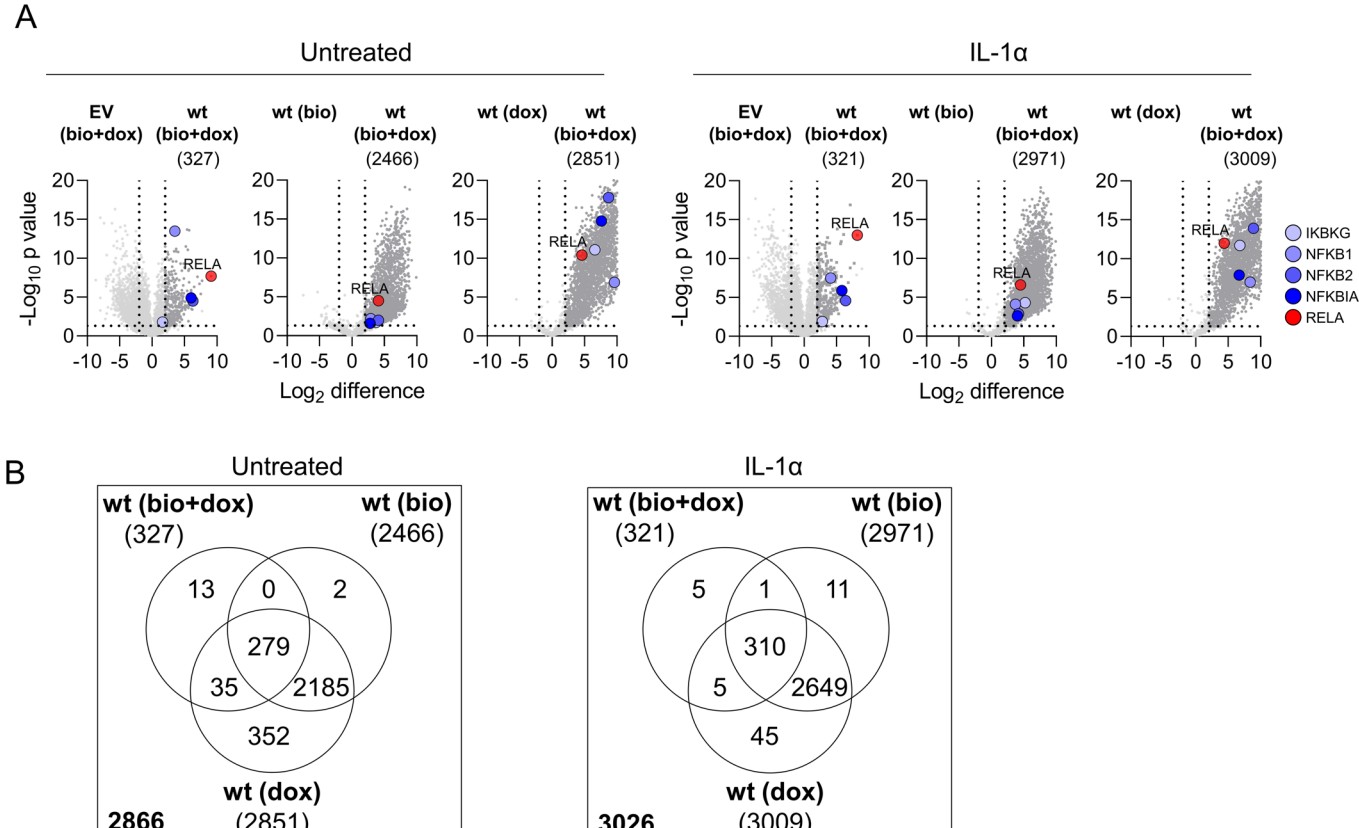

**Figure EV2.  Identification of p65/RELA high-confidence interactors.**

(A) Biotinylated proteins from the experiments shown in Fig. 1C and from a second biological replicate were identified by mass spectrometry in the presence or absence of IL-1α treatment of cells. Volcano plots show the ratio distributions of $Log_2$ transformed mean protein intensity values obtained with wild-type p65 in the presence of doxycycline and biotin (wt) compared to the empty vector control (EV) or compared with conditions in which only biotin (wt(bio)) or doxycycline (wt(dox)) were added to the cell cultures, to determine false positive values in the absence of expression of fusion protein but facilitated biotinylation, or in the absence of biotinylation but induced expression of the fusion protein, respectively. *X* axes show mean ratio value and Y axes show *P* values from *t* test results. Strong enrichment of the bait p65/RELA proteins together with the core canonical NF-κB components is shown in red and blue colors, respectively (two biologically independent experiments and three technical replicates per sample). (B) Specific proteins binding to p65/RELA wild type were defined by significant enrichment (LFC ≥ 2, $-log_{10}$ *P* ≥ 1.3, Student's *t* test) compared to HA-miniTurbo only and to cells exposed to doxycycline or biotin only as shown in (A). Venn diagrams show the total numbers of specific p65/RELA interactors and their overlaps before and after IL-1α-treatment. The intersecting 279 (without IL-1α) and 310 (with IL-1α) interactors were pooled, resulting in the set of 366 specific p65/RELA interactors that was used for further downstream analyses. Numbers in the left lower corner of the boxes indicate the total number of detected interactors.

# A

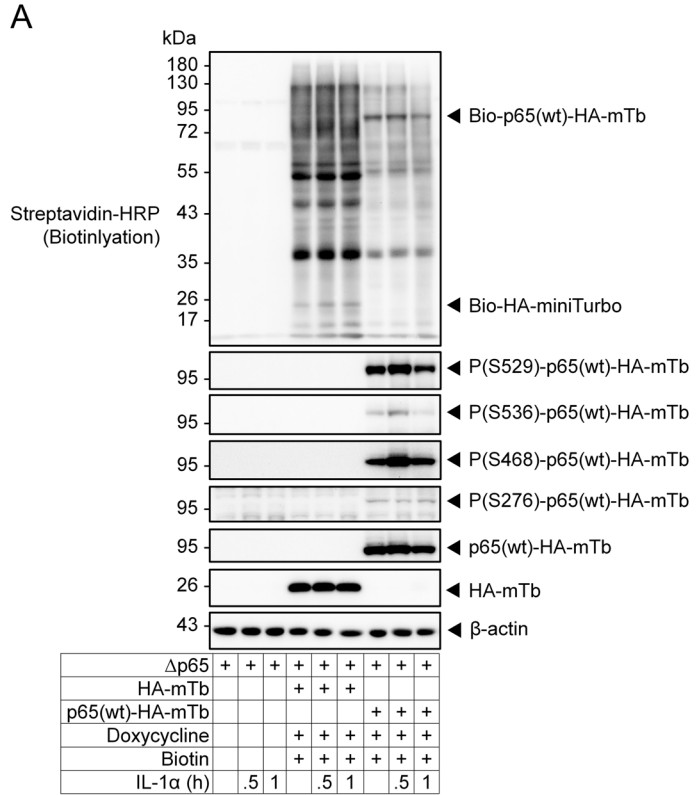

# B

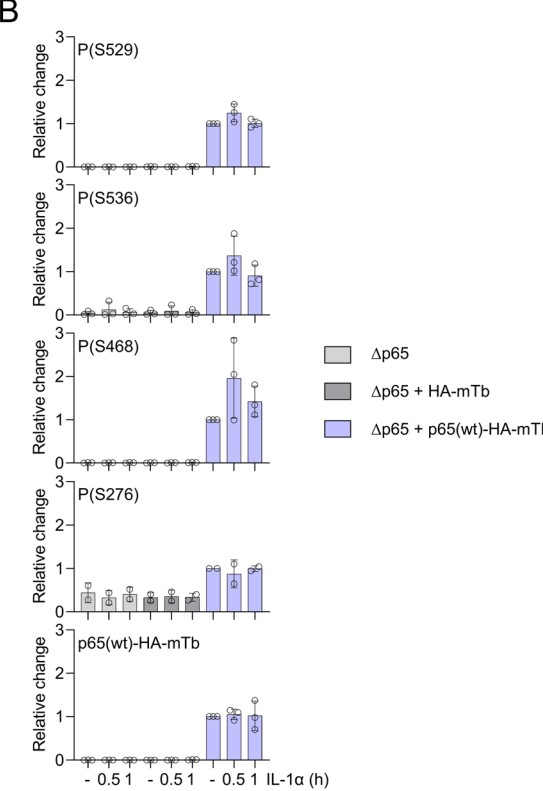

# C

| Gene name | Phospho-site (STY) probabilities | Localization probability | Amino acid | Positions within human RELA |
|---|---|---|---|---|
| RELA | VNRN**S**(0.8)GS(0.2)CLGGDEIFLLCDK | 0.800213 | S | 203 |
| RELA | VNRNS(0.367)G**S**(0.633)CLGGDEIFLLCDK | 0.63334 | S | 205 |
| RELA | RP**S**(1)DRELSEPMEFQYLPDTDDR | 0.999673 | S | 276 |
| RELA | RPS(0.292)DREL**S**(0.708)EPMEFQYLPDTDDR | 0.707675 | S | 281 |
| RELA | TYETFK**S**(1)IMKK | 0.999997 | S | 311 |

# D

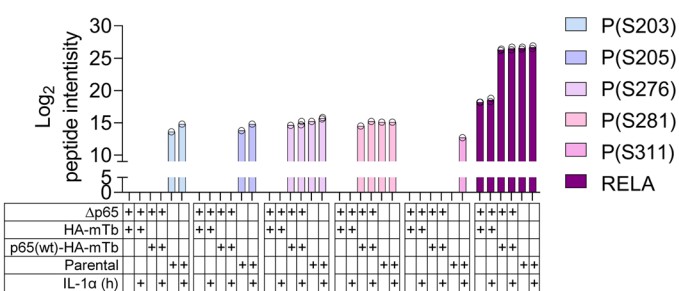

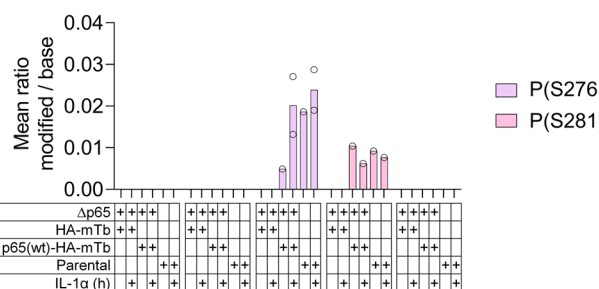

◀ **Figure EV3. Phosphorylation of the p65-HA-mTb fusion protein.**

(A) Pools of HeLa cells with CRISPR/Cas9-based suppression of endogenous p65/RELA (Δp65) were transiently transfected (using branched polyethyleneimine, PEI) with plasmids encoding HA-miniTurbo (empty vector, EV) or p65(wt)-HA-mTb or were left untransfected. Expression of HA-mTb or p65(wt)-HA-mTb was induced with doxycycline (1 μg/ml) for 17 h. Intracellular biotinylation was induced by the addition of 50 μM biotin for further 60 min during which time half of the samples were additionally treated with IL-1α (10 ng/ml) for 30 or 60 min. Whole-cell extracts were prepared in urea buffer and (phospho-)proteins were analyzed by Western blotting for the expression of HA-miniTurbo or p65-HA-miniTurbo or the modification of p65-HA-mTb using the indicated antibodies. Equal loading was confirmed by probing the blots with anti β-actin antibodies. (B) Phospho-protein bands of p65-HA-mTb were normalized to the expression of p65-HA-mTb and changes were quantified relative to the corresponding untreated conditions of cells reconstituted with p65-HA-mTb. Bar graphs show data points and mean values ± s.d. from two or three independent experiments. The small increase of S468, S529 and S536 phosphorylation at 0.5 h of IL-1α-stimulation is not significant (according to one-way ANOVA). (C) The mass spectra of the p65-HA-mTb interactome analyses described in Fig. 1 were re-investigated for phosphorylated peptides of p65/RELA. The table shows peptide sequences and the positions of amino acids with mass changes indicating phosphorylation. (D) The left graph shows (phospho-)peptide intensities for p65-HA-mTb across all conditions and the right graphs shows proportion of modified peptides.

## A

**Phospho-proteins in the p65(wt)-HA-mTb interactome**

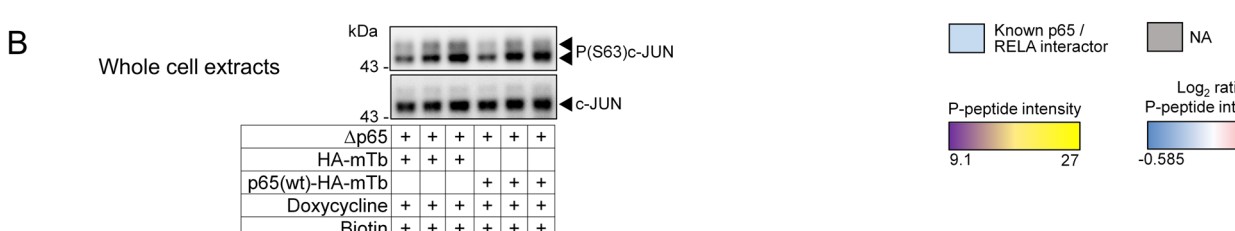

| mTurboID enrichment of protein (Group) | Gene Name | Phospho-site (STY) probabilities | Amino acid | Positions within proteins | HA-mTb Exp.1 | HA-mTb Exp.2 | HA-mTb +IL-1 Exp.1 | HA-mTb +IL-1 Exp.2 | p65-HA-mTb Exp.1 | p65-HA-mTb Exp.2 | p65-HA-Tb +IL-1 Exp.1 | p65-HA-Tb +IL-1 Exp.2 | Mean Log₂ Ratio ± IL-1α |
|---|---|---|---|---|---|---|---|---|---|---|---|---|---|
| wt,wt+IL-1(223) | JUN | NSDLLTS(1)PDVGLLK | S | 63 | | | | 14.79 | 14.24 | 15.52 | 16.12 | 16.24 | 1.30 |
| wt+IL-1(87) | TCF20 | ILQLMPQLS(0.985)PT(0.013)PS(0.002)MMPSPNSHAAGFK | S | 419 | | | | | 15.18 | 15.45 | 15.66 | 16.05 | 0.55 |
| wt,wt+IL-1(223) | MED8 | NQVIIPLVLS(1)PDRDEDLMR | S | 82 | | | | | 13.18 | 14.16 | 13.79 | 14.29 | 0.37 |
| wt,wt+IL-1(223) | FOSL1 | SSSSSGDPSSDPLGS(0.999)PT(0.001)LLAL | S | 265 | | | | | 13.40 | 14.48 | 14.05 | 14.55 | 0.36 |
| wt(56) | ELMSAN1 | RRAS(1)QEANLLTAQK | S | 461 | | | | | 15.20 | 15.23 | 15.48 | 15.63 | 0.34 |
| wt+IL-1(87) | TRIM24 | YLMLPAPMLGSAET(0.003)PPPVPAPGS(0.991)PVS(0.003)GS(0.003)S(0.001)PFATQVGVIR | S | 110 | | | | | 14.84 | 14.64 | 14.47 | 15.69 | 0.34 |
| wt+IL-1(87) | TRIM24 | YLMLPAPMLGS(0.025)AET(0.957)PPPVPAPGS(0.018)PVSGSSPFATQVGVIR | T | 101 | | | | | 15.96 | 16.31 | 15.87 | 17.02 | 0.31 |
| wt,wt+IL-1(223) | KMT2D | ASEPLLS(1)PPPFGESR | S | 2274 | | | | | 15.92 | 16.21 | 16.63 | 16.06 | 0.28 |
| wt+IL-1(87) | TLE3 | DAPTS(1)PASVASSSSTPSSK | S | 286 | | | | | 16.34 | 16.66 | 16.47 | 16.90 | 0.18 |
| wt(56) | DOT1L | VAGPADAPMDS(1)GAEEEKAGAATVK | S | 374 | | | | | 15.20 | 15.07 | 15.34 | 15.29 | 0.18 |
| wt+IL-1(87) | NCOR2 | MGS(0.005)KS(0.993)PGNT(0.002)SQPPAFFSK | S | 2258 | | | | | 16.03 | 15.91 | 16.05 | 15.92 | 0.02 |
| wt,wt+IL-1(223) | BNC2 | TEPACVS(1)PIQNSAPVSDLTK | S | 403 | | | | | 15.28 | 15.50 | 15.20 | 15.41 | -0.09 |
| wt,wt+IL-1(223) | SMARCA4 | KAENAEGQTPAIGPDGEPLDETS(0.001)QMS(0.998)DLPVK | S | 613 | | | | | 15.84 | 15.84 | 15.66 | 15.51 | -0.25 |
| wt+IL-1(87) | FOSL2 | SSSSGDQSSDSLNS(0.999)PT(0.001)LLAL | S | 320 | | | | | 15.58 | 15.98 | 15.78 | 14.87 | -0.45 |

Known p65 / RELA interactor

NA

P-peptide intensity — 9.1 to 27

Log₂ ratio P-peptide intensity — -0.585 to 0.585

## B

Whole cell extracts

P(S63)c-JUN

c-JUN

| | | | | | | |
|---|---|---|---|---|---|---|
| Δp65 | + | + | + | + | + | + |
| HA-mTb | + | + | + | | | |
| p65(wt)-HA-mTb | | | | + | + | + |
| Doxycycline | + | + | + | + | + | + |
| Biotin | + | + | + | + | + | + |
| IL-1α (h) | | .5 | 1 | | .5 | 1 |

## C

c-JUN interactors in the p65(wt)-HA-mTb interactome

untreated — EV | wt (52); IL-1α — EV | wt (53)

-Log₁₀ p value vs Log₂ difference

wt; wt, wt+IL-1α; wt+IL-1α

## D

Interaction (STRING)

wt; wt, wt+IL-1α; wt+IL-1α

Biotinylation; Phosphorylation

## E

**Figure EV4. Phosphorylated p65/RELA interactors and their protein interaction networks.**

(A) The mass spectra of the 366 p65-HA-mTb interactors were re-investigated for phosphorylated amino acids resulting in 185 phospho-peptides representing 56 unique proteins. The table shows 14 phospho-peptides from 13 unique proteins with intensity values in all p65-HA-mTb samples and their regulation by IL-1α along with the HA-mTb negative controls. Phosphorylation sites are colored in red. (B) Immunoblots showing phosphorylation of c-JUN at Ser 63 in whole cells extracts of untreated or IL-1α-treated Δp65 cells transiently expressing p65-HA-mTb or HA-mTb as described in Fig. EV3. (C) Volcano plots visualizing c-JUN and known c-JUN interactors (based on STRING entries) that were significantly enriched with p65(wt)-HA-mTb (LFC ≥ 2, −log10 $P$ ≥ 1.3, Student's $t$ test) compared with HA-mTb (empty vector control, EV) before and after IL-1α treatment. (D) STRING-based protein interaction networks of c-JUN interactors according to their IL-1α-dependent enrichment in the biotinylated p65(wt)-HA-mTb interactome. Phosphorylated proteins (P) shown in (A) are indicated. (E) Similar networks were constructed for FOSL2, KMT2D and NCOR2 interactors. Known p65/RELA interactors (based on STRING) are colored in light blue.

A

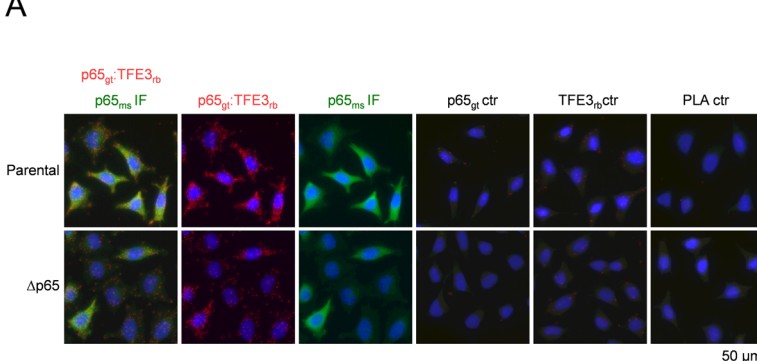

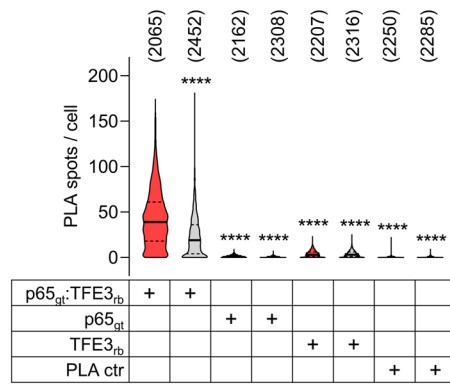

B

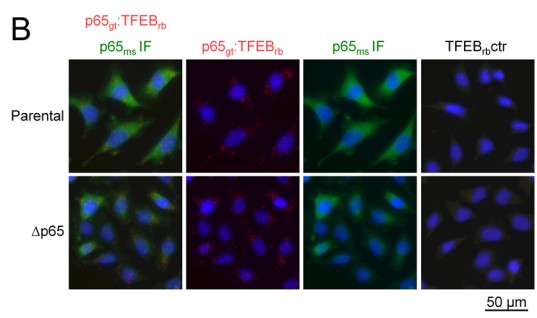

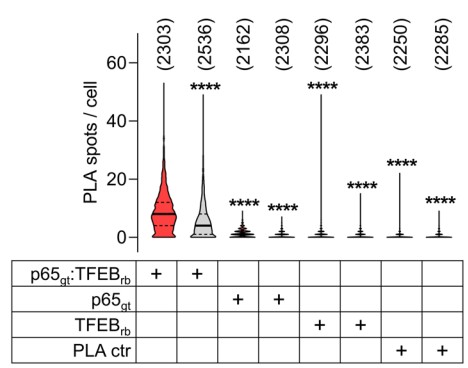

C

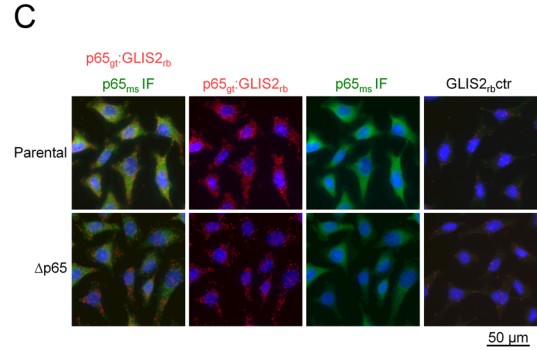

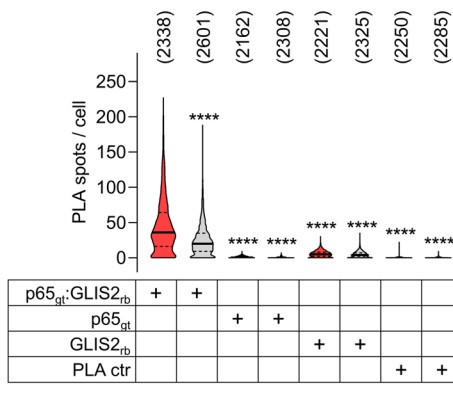

D

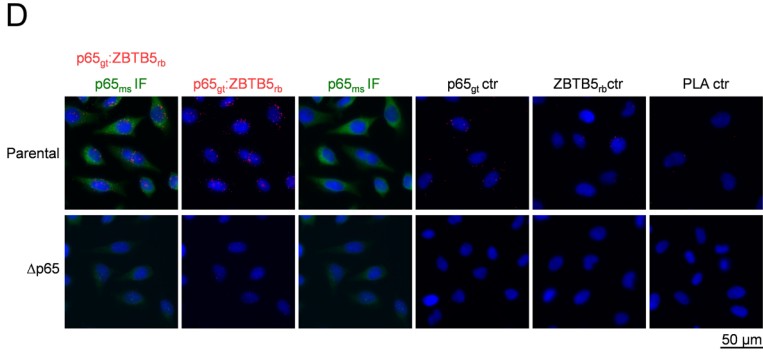

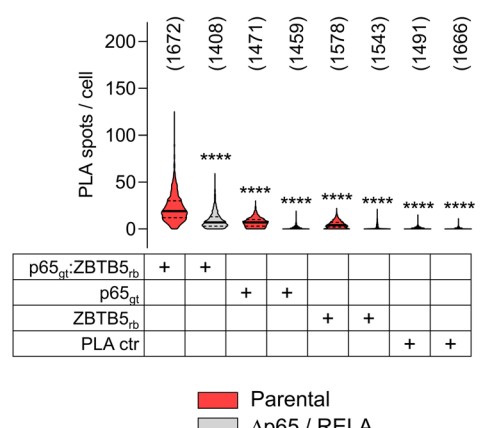

◀ **Figure EV5. Proximity ligation assays confirm endogenous protein–protein interactions of p65/RELA with interactors.**

Proximity ligation assays coupled to immunofluorescence (IF) were performed with HeLa cells or Δp65 HeLa cells lacking endogenous p65/RELA to demonstrate interactions of p65/RELA with TFE3 (**A**), TFEB (**B**), GLIS2 (**C**) and ZBTB5 (**D**) using pairs of antibodies along with negative control conditions as indicated. PLA spots are colored in red, while p65 IF is colored in green. Nuclear DNA is counterstained with Hoechst 33342 (blue signals). The images show representative fluorescence raw data. The violin plots on the right show quantification of PLA spots per cell from the numbers of cells indicated in brackets as obtained from three (TFE3, TFEB, GLIS2) or two (ZBTB5) independent experiments. Samples lacking one or both primary antibodies (PLA ctr) served as negative controls. Experiments shown in (**A**–**C**) were performed in parallel with one set of PLA ctr and p65$_{gt}$ antibody only samples that were included in each of the graphs shown in (**A**–**C**) for comparison. Solid lines indicate medians and dashed lines indicate 1st and 3rd quartiles. Asterisks indicate results from Kruskal-Wallis tests compared to the parental control (****$P \le 0.0001$) obtained by one-way ANOVA. Scale bars indicate 50 μm. gt goat, ms mouse, rb rabbit.

