## [Peer Review File · EMBO Reports]

The proximity-based protein interactome and regulatory logics of the transcription factor p65 NF-kappaB / RELA

Lisa Leib, Jana Juli, Liane Jurida, Christin Mayr-Buro, Jasmin Priester, Hendrik Weiser, Stefanie Wirth, Simon Hanel, Daniel Heylmann, Axel Weber, M. Lienhard Schmitz, Argyris Papantonis, Marek Bartkuhn, Jochen Wilhelm, Uwe Linne, Johanna Meier-Soelch, and Michael Kracht

Corresponding author(s): Michael Kracht (Michael.Kracht@pharma.med.uni-giessen.de) , Johanna Meier-Soelch (Johanna.Meier-Soelch@pharma.med.uni-giessen.de)

Review Timeline:

Submission Date:	31st Jan 24
Editorial Decision:	5th Apr 24
Revision Received:	16th Sep 24
Editorial Decision:	25th Oct 24
Revision Received:	6th Nov 24
Accepted:	14th Nov 24

Transaction Report:

Dear Prof Kracht

Thank you once more for the submission of your research manuscript to our journal, for providing feedback on the referee reports and for the further discussion today. Referee #1 and #3 are experts in NF- κ B signaling and consider your study interesting with great value as a resource.

As discussed, I would like to invite you to revise your study along the lines suggested in your revision plan and along the lines discussed today. Please address the concerns from referee #2 in an extended discussion of the potential limitations of the approach chosen.

Acceptance of the manuscript will depend on a positive outcome of a second round of review, including potential additional, independent feedback on the proteomics aspect and your response to these. It is EMBO Reports policy to allow a single round of revision only and acceptance or rejection of the manuscript will therefore depend on the completeness of your responses included in the next, final version of the manuscript.

We realize that it is difficult to revise to a specific deadline. In the interest of protecting the conceptual advance provided by the work, we recommend a revision within 3 months (July 5th). Please discuss the revision progress ahead of this time with the editor if you require more time to complete the revisions.

*****IMPORTANT NOTE:

We perform an initial quality control of all revised manuscripts before re-review. Your manuscript will FAIL this control and the handling will be delayed IN CASE the following APPLIES:

- 1) A data availability section providing access to data deposited in public databases is missing. If you have not deposited any data, please add a sentence to the data availability section that explains that.
- 2) Your manuscript contains statistics and error bars based on $n=2$. Please use scatter blots in these cases. No statistics should be calculated if $n=2$.

When submitting your revised manuscript, please carefully review the instructions that follow below. Failure to include requested items will delay the evaluation of your revision. *****

2) individual production quality figure files as .eps, .tif, .jpg (one file per figure). Please download our Figure Preparation Guidelines (figure preparation pdf) from our Author Guidelines pages <https://www.embopress.org/page/journal/14693178/authorguide> for more info on how to prepare your figures.

4) a complete author checklist, which you can download from our author guidelines (<<https://www.embopress.org/page/journal/14693178/authorguide>>). Please insert information in the checklist that is also reflected in the manuscript. The completed author checklist will also be part of the RPF.

5) Please note that all corresponding authors are required to supply an ORCID ID for their name upon submission of a revised manuscript (<<https://orcid.org/>>). Please find instructions on how to link your ORCID ID to your account in our manuscript tracking system in our Author guidelines (<<https://www.embopress.org/page/journal/14693178/authorguide#authorshipguidelines>>)

6) We replaced Supplementary Information with Expanded View (EV) Figures and Tables that are collapsible/expandable online. A maximum of 5 EV Figures can be typeset. EV Figures should be cited as 'Figure EV1, Figure EV2' etc... in the text and their respective legends should be included in the main text after the legends of regular figures.

7) Please note that a Data Availability section at the end of Materials and Methods is now mandatory. In case you have no data that requires deposition in a public database, please state so instead of refereeing to the database. See also <<https://www.embopress.org/page/journal/14693178/authorguide#dataavailability>>. Please note that the Data Availability Section is restricted to new primary data that are part of this study.

Additional information on source data and instruction on how to label the files are available <<https://www.embopress.org/page/journal/14693178/authorguide#sourcedata>>.

10) Figure legends and data quantification:
The following points must be specified in each figure legend:

- the name of the statistical test used to generate error bars and P values,
 - the number (n) of independent experiments (please specify technical or biological replicates) underlying each data point,
 - the nature of the bars and error bars (s.d., s.e.m.)
- If the data are obtained from n {less than or equal to} 5, show the individual data points in addition to the SD or SEM.
- If the data are obtained from n {less than or equal to} 2, use scatter blots showing the individual data points.

See also the guidelines for figure legend preparation:
<https://www.embopress.org/page/journal/14693178/authorguide#figureformat>

11) Our journal encourages inclusion of *data citations in the reference list* to directly cite datasets that were re-used and obtained from public databases. Data citations in the article text are distinct from normal bibliographical citations and should directly link to the database records from which the data can be accessed. In the main text, data citations are formatted as follows: "Data ref: Smith et al, 2001" or "Data ref: NCBI Sequence Read Archive PRJNA342805, 2017". In the Reference list, data citations must be labeled with "[DATASET]". A data reference must provide the database name, accession number/identifiers and a resolvable link to the landing page from which the data can be accessed at the end of the reference. Further instructions are available at <<https://www.embopress.org/page/journal/14693178/authorguide#referencesformat>>.

12) All Materials and Methods need to be described in the main text. We would ask you to use 'Structured Methods', our new Methods format. According to this format, the Methods section should include a Reagents and Tools Table (listing key reagents, experimental models, software and relevant equipment and including their sources and relevant identifiers) followed by a Methods and Protocols section in which we encourage the authors to describe their methods using a step-by-step protocol format with bullet points, to facilitate the adoption of the methodologies across labs. More information on how to adhere to this format as well as downloadable templates (.doc or .xls) for the Reagents and Tools Table can be found in our author guidelines: <<https://www.embopress.org/page/journal/14693178/authorguide#manuscriptpreparation>>.

An example of a Method paper with Structured Methods can be found here:
<<https://www.embopress.org/doi/10.15252/msb.20178071>>.

13) As part of the EMBO publication's Transparent Editorial Process, EMBO Reports publishes online a Review Process File to accompany accepted manuscripts. This File will be published in conjunction with your paper and will include the referee reports, your point-by-point response and all pertinent correspondence relating to the manuscript.

Yours sincerely,

Referee #1:

Despite the NF- κ B pathway being one of the most highly studied, there are still areas where there is a significant lack of important information and understanding. This manuscript addresses one of these by identifying proteins associated with the p65/RelA NF- κ B subunit. Although this issue has been addressed in previous studies, it was clear from these that, likely for technical reasons, only a small number of p65/RelA associated proteins were identified and in particular, there was a lack of insight into the range of other transcription factors it can interact with. Here, using the Turbo-ID approach, Leib et al have characterised p65/RelA interacting protein before and after IL-1 stimulation in HeLa cells. Moreover, they have functionally validated a significant amount of their data and performed more in depth analysis on a group of novel p65/RelA interacting proteins.

Overall, I was impressed by the detail and thoroughness of this study. There is a wealth of new and interesting information reported here. However, there are some areas where I think some additional clarity is required.

Specific concerns

(1) The system used here involves exogenously expressed p65/RelA in a cellular background where the endogenous protein has been depleted. From Supp Fig 1 it can be seen at the dosage of doxycycline used to induce p65/RelA for Turbo-ID experiments, it is significantly over-expressed relative to endogenous levels. Moreover, from Supp Fig 1C, it can be seen that while nuclear/chromatin localization of endogenous p65/RelA is induced by IL-1, the exogenous tagged p65/RelA is constitutively in these compartments at a high level that does not change upon IL-1 stimulation. There are a number of implications of this:

(a) This may skew the data towards more effects/interactors being seen in unstimulated cells than would be the case for endogenous p65/RelA. This is to a large extent an unavoidable issue associated with the experimental system that does not detract from the overall data presented and just needs to be discussed appropriately in the manuscript.

(b) One consequence of this is that the majority of p65/RelA dimers in the cells being used for Turbo-ID are homodimers, with heterodimers with other subunits such as p50 being under-represented. The proteins identified here might therefore be better described as the p65/RelA homodimer interactome. This should be clarified experimentally, with the nature of the p65/RelA dimer being studied being defined by an approach such as EMSA or co-IP. Should it be the case that it is a p65/RelA homodimer being studied, then it would be informative if it could be experimentally determined whether some (or all) of the interactors studied in Figure 4 can also interact with a p50/RelA heterodimer, as this is the NF- κ B complex most frequently observed following an inflammatory stimulus such as IL-1.

(c) Despite the issues discussed above, there is still a significant change in p65/RelA associated proteins following IL-1

stimulation. This suggests that inducible post-translational modification of p65/RelA might play an important role in promoting these. It would be useful therefore if, by western blot with commercially available antibodies, it could be shown if the p65/RelA fusion protein used for the Turbo-ID experiments is still inducibly phosphorylated following IL-1 treatment.

(d) Although a doxycycline dose of 1ug/ml is used to induce p65/RelA for the Turbo-ID experiments, Supp Fig 1C states that a dose of 10ng/ml was used. Is this a typo or were different doses used for different experiments?

(2) With the STRING analysis in Fig. 2A, it would be useful to know how many of the proteins not reported to interact with p65/RelA have been reported to interact with proteins that have been shown to do so. That is, how many of the interactions picked up here result from binding a protein complex and are indirect? Some more details on the settings used for the STRING analysis would be useful as when I tried to repeat this I got subtly different results.

(3) In Figure 7G, does knockdown of one factor abolish recruitment of the other? One thing missing from this analysis is whether there is cooperative recruitment of these TFs with p65/RelA to target genes.

(4) The sections from lines 328-330 and 336-337 I found confusing. The authors should consider rewriting it to improve clarity, perhaps describing more what is being seen in the figures. For example, the violin plots in Supp Fig 7C. With this figure, I assume the one on the left is series 1 and on the right, series 2?

(5) The Supplementary tables would benefit from having a page that describes the content of the table. There should also be an additional table that contains the details of the STRING analysis shown in Fig. 6C.

Referee #2:

The manuscript 'The proximity-based protein interaction landscape of the transcription factor p65 NF- κ B / RELA and its gene-regulatory logics' by Leib et al., presents an investigation into the protein interactome of p65 / RELA, a crucial subunit of the NF- κ B transcription factor, which interactions have not been extensively studied using interaction proteomics. By employing p65-miniTurbo fusion proteins and biotin tagging, the study identifies over 350 interactors of RELA, highlighting the significance of dimerization for these interactions. It provides comprehensive insights into gene regulatory networks controlled by RELA and its interactors, contributing to a deeper understanding of its role in gene expression patterns. This research could offer a resource for the scientific community, presenting a novel framework for exploring RELA's function and its cooperativity in determining gene expression patterns -provided that some meaningful validation for the claimed 350 interactions would be provided.

Major comments:

-The analysis currently lacks the standard controls typically employed in proximity labeling (PL) studies. At a minimum, the use of mini-Turbo tagged GFP/LacZ, both with and without an NLS-sequence, is essential to identify and exclude unspecific interactors. Incorporating an NLS-tagged control is especially crucial when investigating transcription factor (TF) interactomes via PL, a practice commonly adhered to in numerous studies. As it stands, the identified list of 350 interactors includes several entities frequently recognized as contaminants.

-It is clear from the volcano plots that the empty vector (EV) does not serve as an effective control because the comparison yields a distribution that is not comparable to the bait samples.

-Many of the purported high-confidence interactors are detected in very low amounts (MS1, MS2) and with only a few peptides. Interactors with a relative abundance of less than 5% are most likely not meaningful, and some rationale or validation for including these should be provided.

-There should be protein-protein interaction (PPI) validation for minimally one-fifth of the 38 novel, high-confidence interactors mentioned in later analyses. Methods such as co-expression co-immunoprecipitation (co-IP), bimolecular fluorescence complementation (BiFC), or reciprocal interactome analysis using the prey as bait should be employed. These methods have been successfully used in several TF interactome studies for validation.

-The RELA ChIP-Seq experiments could benefit from repetition after knocking down TFE3 and GLIS2 using siRNAs.

Minor comments:

-Figure 1A) AF model is not very useful and could be omitted.

Referee #3:

In the paper entitled « The proximity-based protein interaction landscape of the transcription factor p65 NF-kappaB / RELA and its gene-regulatory logics », the authors carried out an impressive and robust interactomic study to extensively identify new p65-binding proteins. This new approach led to the identification of already known interacting proteins such as NF-kappaB proteins p50, p52 etc., which demonstrates the robustness of their experimental design. Importantly, 366 new p65-associated candidates were identified by the authors, which reveals the complexity of all signaling pathways and biological processes controlled by p65/RelA. As expected, many of them are acting as transcription factors or chromatin remodelers and histone-modifying enzymes. To validate the biological relevance of these multiple new interactions, the authors conducted a siRNA-based screening to functionally establish a link between new p65-associated proteins and NF-kappaB-dependent gene expression. Overall, this is a very interesting study that merits publication. I have few comments that should help to further improve the manuscript.

Major points

1. In Figure 3C in which the authors assess the relevance of new p65-associated proteins in the expression of NF-kappaB target genes, they show that the depletion of some candidates impairs some but not all tested NF-kappaB target genes, suggesting some specificity of these interactions for the regulation of some but not all target genes. The authors should comment this finding in the discussion. What would be the underlying mechanism ?
2. For selected p65-associated proteins, the authors confirmed these interactions with PLA (Supplementary Figure 4). However, that would also be nice to confirm some interactions at the endogenous level through co-immunoprecipitation studies, especially for candidates whose antibodies are available (TF3B for example as nice western blots for this candidate are illustrated in Figure 4B). I would for example select some candidates whose interaction is expected to be regulated upon IL-1 stimulation. Very good co-immunoprecipitations would further validate their findings.
3. In Figure 4D, the authors demonstrate that GLIS2 knockdown reduced p65 protein levels. The authors should discuss the possibility that GLIS2 prevents p65 degradation, possibly by competing with any E3 ligases for binding to this NF-kappaB family member.
4. The authors are interested in IL-1alpha signaling but one may wonder whether most of the described interactions with p65 could also be found in cells treated with TNFalpha, a widely studied signaling pathway. That would be nice to assess this issue for some p65-associated proteins.
5. As p65 promotes cell survival in most cell types and given the fact that NF-kappaB limits TNF-dependent cell death through the expression of target genes such as c-FLIP, etc...do some p65-associated proteins also prevent TNF-dependent cell death ? The authors could deplete each candidate and assess the consequence on cell apoptosis upon TNF stimulation. These experiments would also increase, if relevant, the biological relevance of their findings.
6. The authors should also discuss or even experimentally address the possibility that some interactions are regulated by p65 phosphorylation, especially the ones found on IL1alpha-stimulated cells. Does p65 phosphorylation enhance some interactions ? Using WT p65 or a mutant lacking the Serine 536 residue known to be phosphorylated upon TNF stimulation would be a good strategy to look for interaction by IP with some identified candidates.

Minor points

1. There is a typing mistake in the title mentioned on the website of EMBO reports...

Editorial comments:

As you will see, the referees consider your manuscript potentially interesting but they also raise a number of important concerns. E.g., the use of overexpressed proteins with might skew the interaction partners to that interacting with p65/RelA homodimers and create significant background (referee #1). Referee #2, an expert in proximity labeling and proteomics, raises significant concerns regarding the control experiments provided. Since the protein of interest is a nuclear protein, a nuclear control is required to appropriately assess unspecific binding in this cellular compartment. As discussed, I would like to invite you to revise your study along the lines suggested in your revision plan and along the lines discussed today. Please address the concerns from referee #2 in an extended discussion of the potential limitations of the approach chosen.

Reply: We thank the editors for their interest in our manuscript, the opportunity for direct scientific discussion of the reviewer reports and the invitation to revise our study.

We have carefully considered the comments of all three reviewers and have addressed them both experimentally and by adapting the text. Accordingly, the revised manuscript now includes several new experiments and analyses that fully address the comments of reviewers #1 and #3.

Concerning the comments of reviewer #2 regarding the need for additional bait protein controls, we are providing additional *in silico* analyses utilizing data from a recent GFP-BirA* interactome. However, we would like to point out that p65 / RELA is not a solely nuclear protein (see Fig. 5B for an example) and thus our study was not designed to focus only on the nuclear compartment. Moreover, miniTurbo expression is found throughout the cell, including the nucleus, as already published in the original paper describing this enzyme and now corroborated in our model system by new experiments (Branon *et al*, 2018) (New Fig. EV1). It thus provides a suitable control for our type of study. Here, by pursuing a targeted, highly specific approach we report the entire, point mutation sensitive interactome of p65 / RELA, which is enriched for nuclear proteins, but also includes all key components of the cytosolic NF- κ B pathway (NEMO, NFKBIA, etc.). Please find our point-to-point responses below.

Reviewer comments:

Referee #1

Despite the NF- κ B pathway being one of the most highly studied, there are still areas where there is a significant lack of important information and understanding. This manuscript addresses one of these by identifying proteins associated with the p65 / RelA NF- κ B subunit. Although this issue has been addressed in previous studies, it was clear from these that, likely for technical reasons, only a small number of p65 / RelA associated proteins were identified and in particular, there was a lack of insight into the range of other transcription factors it can interact with. Here, using the Turbo-ID approach, Leib *et al* have characterised p65 / RelA interacting protein before and after IL-1 stimulation in Hela cells. Moreover, they have functionally validated a significant amount of their data and performed more in depth analysis on a group of novel p65 / RelA interacting proteins.

Overall, I was impressed by the detail and thoroughness of this study. There is a wealth of new and interesting information reported here. However, there are some areas where I think some additional clarity is required.

Reply: We thank the reviewer for the very positive evaluation of our study and for the thoughtful suggestions to improve our manuscript.

(1) The system used here involves exogenously expressed p65 / RelA in a cellular background where the endogenous protein has been depleted. From Supp Fig 1 it can be seen at the dosage of doxycycline used to induce p65 / RelA for Turbo-ID experiments, it is significantly over-expressed relative to endogenous levels. Moreover, from Supp Fig 1C, it can be seen that while nuclear/chromatin localization of endogenous p65 / RelA is induced by IL-1, the exogenous tagged p65 / RelA is constitutively in these compartments at a high level that does not change upon IL-1 stimulation. There are a number of implications of this:

(a) This may skew the data towards more effects/interactors being seen in unstimulated cells than would be the case for endogenous p65 / RelA. This is to a large extent an unavoidable issue associated with the experimental system that does not detract from the overall data presented and just needs to be discussed appropriately in the manuscript.

Reply: We agree and have emphasized this point in the Limitations section of the Discussion as follows:

“A further limitation is the necessity to ectopically express a p65 / RELA fusion protein that, although performed in a p65 / RELA-deficient background using a conditional system, could influence the stoichiometry of some of the interactions we discovered with a potential preference for constitutive binding events or for p65 / RELA homodimers.”

(b) One consequence of this is that the majority of p65 / RelA dimers in the cells being used for Turbo-ID are homodimers, with heterodimers with other subunits such as p50 being under-represented. The proteins identified here might therefore be better described as the p65 / RelA homodimer interactome. This should be clarified experimentally, with the nature of the p65 / RelA dimer being studied being defined by an approach such as EMSA or co-IP. Should it be the case that it is a p65 / RelA homodimer being studied, then it would be informative if it could be experimentally determined whether some (or all) of the interactors studied in Figure 4 can also interact with a p50 / RelA heterodimer, as this is the NF- κ B complex most frequently observed following an inflammatory stimulus such as IL-1.

Reply: This is an important and very interesting point. As shown in Fig. 2A, the p50 protein (=NFKB1) is reproducibly and significantly enriched in our mass spec analyses. However, the RELA bait protein (\log_2 protein intensity of 26, Dataset EV1. Source_data_of_LC-MS_MS_experiments.) is more abundant than NFKB1 (\log_2 protein intensity of 19, Dataset EV1. Source_data_of_LC-MS_MS_experiments.), which would be consistent with a more dominant p65 homodimer interactome as suggested. Because the LC-MS/MS data are based on quantifying a range of tryptic peptides, in our view, the proteomic data do not allow concluding on how many interactions are mediated by RELA homo- or heterodimers. Following the reviewers suggestions, we tried the antibodies available to us, to test the co-association of RELA interactors from previous Fig. 4 (now Fig. 5) with p50. Endogenous TFE3, but not TFEB, interacted specifically with p50 NF- κ B in PLA assays in a p65-dependent manner, providing exemplary evidence for interactions with p65:p50 heterodimers (New Appendix Fig. S3A). Moreover, we could validate co-immunoprecipitation of small amounts of TFE3 only with endogenous p65:p50 complexes (New Appendix Fig. S3B). Beyond this, the (few) antibodies for TFEB, GLIS2 and ZBTB5 that worked in our hands for immunoblots or PLAs, failed in other interaction assays, thus highlighting the difficulty of validating proximity-based interaction assays with conventional reagents and methods. We have adapted the discussion to point out that the interactome may have a preference towards p65 homodimers as stated above in our reply to comment (a).

(c) Despite the issues discussed above, there is still a significant change in p65 / RelA associated proteins following IL-1 stimulation. This suggests that inducible post-translational modification of p65 / RelA might play an important role in promoting these. It would be useful therefore if, by western blot with commercially available antibodies, it could be shown if the p65 / RelA fusion protein used for the Turbo-ID experiments is still inducibly phosphorylated following IL-1 treatment.

Reply: For this study, we decided to employ a standardized transient expression system for p65-HA-mTb to minimize adaptation effects caused by long-term expression of mTb. Although we used an inducible expression system in a p65-depleted background, it was difficult to accurately recapitulate the endogenous cytosol-nuclear regulation of p65. One reason for this is that stable p65-deficient cell lines generated by Crispr-Cas9 show reduced I κ B α protein levels as reported in our previous characterization of the Δ p65 HeLa cells, since p65 regulates this gene via a transcriptional feedback loop and I κ B α becomes unstable in the absence of p65 (Weiterer *et al*, 2020). This explains why

population-based assays such as immunoblot experiments show, on average, a constitutive accumulation of p65-HA-mTb in the cell nucleus. Following the reviewer's suggestion, we have now analyzed four phosphorylation sites in p65 using commercial P-p65 antibodies in new experiments (New Fig. EV3A). The quantification of these data from three independent replicates shows a small but statistically not significant regulation of the important C-terminal P-sites S468, S529 and S536 at 30 minutes of IL-1 stimulation (New Fig. EV3B).

Furthermore, by re-analyzing the existing LC-MS/MS data, we found five additional phosphorylation sites in p65 / RELA (New Fig. EV3C). However, these are only detectable in a small fraction of the peptides (New Fig. EV3D), consistent with a low abundant substoichiometric phosphorylation of p65 and a more difficult detection of P-peptides in conventional LC-MS/MS, without prior enrichment of P-peptides. In summary, these new analyses demonstrate that the p65-HA-mTb construct is still susceptible to phosphorylation over the entire amino acid sequence range. In a smaller proportion of cells, this phosphorylation may still be IL-1-regulatable and may therefore be responsible for the described IL-1-dependent effects in the interactome.

Another possibility for how IL-1-dependent interactions could still be identified in the system used is that other IL-1-responsive signaling pathways and PTMs modulate the recruitment of factors to p65-HA-mTb. We tested this at the example of c-JUN, whose phosphorylation is stimulated by IL-1 in cells transiently expressing p65-HA-mTb. Interestingly, the phospho-form of c-Jun was also specifically enriched in the p65-HA-mTb interactome after IL-1 treatment. As c-Jun is supposed to interact with approximately 50 factors that are also part of the p65-HA-mTb interactome, the data provide one example how parts of the p65 / RELA pathway remain cytokine-responsive, even though the p65-HA-mTb fusion protein is constitutive nuclear. These data are shown in a new Fig. EV4.

(d) Although a doxycycline dose of 1 μ g/ml is used to induce p65/RelA for the Turbo-ID experiments, Supp Fig 1C states that a dose of 10ng/ml was used. Is this a typo or were different doses used for different experiments?

Reply: This was an earlier experiment in which we used lower doxycycline doses. Later on, we found that higher doses (1 μ g / ml) were required for efficient biotinylation of cellular proteins in the pulldown assays. Therefore, in most experiments, including the new ones performed for the revision, we used 1 μ g / ml doxycycline to keep the conditions constant and to achieve sufficient biotinylation. We have indicated the doxycycline concentrations used in each legend.

(2) With the STRING analysis in Fig. 2A, it would be useful to know how many of the proteins not reported to interact with p65/RelA have been reported to interact with proteins that have been shown to do so. That is, how many of the interactions picked up here result from binding a protein complex and are indirect? Some more details on the settings used for the STRING analysis would be useful as when I tried to repeat this I got subtly different results.

Reply: For previous Fig. 2A, we analyzed all 366 p65 / RELA interactors in STRING using the default settings (all categories) with a confidence score of 0.4 or higher. 330 out of 366 interactors (nodes) were mapped in the STRING data base and had 2479 one-way (A:B) interactions (edges). These data were searched for RELA in the second node list and yielded the 46 factors shown in previous Fig. 2A (now Fig. 2E), displaying only the RELA interactions as the second node partner along with the relative enrichment (node colors) from the miniTurboID experiments.

Altogether, the 46 RELA interactors had 318 interactions with 881 redundant additional interactors, of which 199 were unique and overlapped with the STRING input list of 330 interactors. This analysis supports the reviewer's suggestion of indirect recruitment of factors into the p65 / RELA interactome. We have combined this re-analysis with search for phospho-proteins in the p65 / RELA interactome and the results are shown in a revised Fig. 2 and new Fig. EV4.

All STRING data, obtained in September 2023 when finalizing the bioinformatics analyses, were exported and are shown in the Source Data for Fig. 2. In the light of this comment, during the revision, we repeated the STRING analysis on August, 05.08.2024, and the output was identical for the 318 interactions of the 46 factors having known interactions with p65 / RELA. However, for the

entirety of all 366 interactors, STRING mapped 348 nodes with 2254 total interactions. The latter difference reflects the dynamic nature of STRING that is constantly updated and revised. Thus, we envisage that should our study be published and the data included in STRING, the known p65 / RELA network will significantly change.

(3) In Figure 7G, does knockdown of one factor abolish recruitment of the other? One thing missing from this analysis is whether there is cooperative recruitment of these TFs with p65 / RelA to target genes.

Reply: We have performed additional CHIP-qPCR assays to address this important question. However, as the only anti TFE3 antibody that in our hands works for CHIP, has been discontinued in 2024 during the revision, we were forced to limit this analysis to the recruitment of p65 / RELA in TFE3-deficient cells. The data for three genomic regions are shown in a new Appendix Fig. S7 and reveal a small reduction of inducible p65 / RELA binding to the promoters of TNFAIP3 and CXCL2, but not an intergenic control region. We also refer to our detailed response to reviewer #2 concerning additional evidence from CHIP assays and the difficulties obtaining suitable commercial antibodies.

(4) The sections from lines 328-330 and 336-337 I found confusing. The authors should consider rewriting it to improve clarity, perhaps describing more what is being seen in the figures. For example, the violin plots in Supp Fig 7C. With this figure, I assume the one on the left is series 1 and on the right, series 2?

Reply: We have restructured and rephrased the text to improve clarity. The left graphs were indeed series 1 and the right ones series 2 and this information has been added.

(5) The Supplementary tables would benefit from having a page that describes the content of the table. There should also be an additional table that contains the details of the STRING analysis shown in Fig. 6C.

Reply: We have added the information to describe the content of the tables. The details of the STRING and Cytoscape analyses shown as network in previous Fig. 6C (now Fig. 7C) were included in the Source Data for Fig. 6B (now Fig. 7B).

Referee #2

The manuscript 'The proximity-based protein interaction landscape of the transcription factor p65 NF- κ B / RELA and its gene-regulatory logics' by Leib et al., presents an investigation into the protein interactome of p65 / RELA, a crucial subunit of the NF- κ B transcription factor, which interactions have not been extensively studied using interaction proteomics. By employing p65-miniTurbo fusion proteins and biotin tagging, the study identifies over 350 interactors of RELA, highlighting the significance of dimerization for these interactions. It provides comprehensive insights into gene regulatory networks controlled by RELA and its interactors, contributing to a deeper understanding of its role in gene expression patterns. This research could offer a resource for the scientific community, presenting a novel framework for exploring RELA's function and its cooperativity in determining gene expression patterns -provided that some meaningful validation for the claimed 350 interactions would be provided.

Reply: We thank the reviewer for this overall favorable opinion. As indicated below, we have tried to address the concerns expressed from a rather broader perspective regarding proximity labelling as best as possible in the context of our study.

Major comments:

-The analysis currently lacks the standard controls typically employed in proximity labeling (PL) studies. At a minimum, the use of mini-Turbo tagged GFP/LacZ, both with and without an NLS-sequence, is essential to identify and exclude unspecific interactors. Incorporating an NLS-tagged control is especially crucial when investigating transcription factor (TF) interactomes via PL, a practice

commonly adhered to in numerous studies. As it stands, the identified list of 350 interactors includes several entities frequently recognized as contaminants.

Reply: We found these more general criticisms difficult to address as, in our view, we pursued a highly targeted, focused approach in which the controls were based on an inducible expression system as well as established insights from the domain structure and regulation of the p65 / RELA protein. We would like to respond to the specific points as follows.

The analysis currently lacks the standard controls typically employed in proximity labeling (PL) studies.

Reply: In our view, MiniTurbo ID is a relatively new approach. With the key date 06.08.2024, we found only 27 studies in Pubmed using miniTurbo as a keyword in titles or abstracts. Therefore, it is not clear to us, if there really is already a commonly agreed standard procedure for performing this type of proximity labeling.

At a minimum, the use of mini-Turbo tagged GFP/LacZ, both with and without an NLS-sequence, is essential to identify and exclude unspecific interactors. Incorporating an NLS-tagged control is especially crucial when investigating transcription factor (TF) interactomes via PL, a practice commonly adhered to in numerous studies.

Reply: We found it difficult to appreciate the need to include GFP or LacZ baits with or without nuclear localization sequences (NLS) in our study. It is well known that the p65 / RELA protein is not restricted to the nucleus, but is also abundant in the cytoplasm (see Figure 5B), where it interacts with key components of the NF- κ B signaling pathway I κ BKG (encoding NEMO) and NFKBIA (encoding I κ B α) as shown in Fig. 1E. In fact, sequestering p65 / RELA in the cytoplasm by I κ B α is a key mechanism to keep NF- κ B inactive and to restrict its access to the nuclear compartment in the absence of triggers (Meier-Soelch *et al*, 2021). Similar shuttling-based control mechanisms exist for other signal-inducible TFs such as NFATs (Muller & Rao, 2010) or TFE3 / TFEB (see Fig. 5B and Appendix Fig. S4), which are part of the p65 / RELA interactome. This may be different for developmental or cell-type specific transcription factors that localize exclusively in the nucleus. Here, it was our principal aim to survey all p65 / RELA interactors throughout the cell.

For this purpose, we used three types of negative controls to minimize false positive hits, i.e. (i) omission of doxycycline, (ii) omission of biotin and (iii) miniTurbo enzyme only. MiniTurbo has been described to localize throughout the cell, including the nucleus (Branon *et al.*, 2018). We have confirmed this in our cell system and under the conditions used for LC-MS / MS. These new data show that mTb is readily detectable in the nucleus and efficiently biotinylates both, cytosolic and nuclear proteins (new Fig. EV1).

Most importantly, we found that the interactomes we assigned to p65 / RELA are highly sensitive to p65/ RELA point mutations as shown in Fig. 1 and Fig. 3. This provides strong independent evidence for their specificity.

As it stands, the identified list of 350 interactors includes several entities frequently recognized as contaminants.

Reply: We do not know which contaminants the reviewer is referring to. Prior to downstream bioinformatics analysis, we removed common contaminants found in proteomic studies as assigned by MaxQuant (as well as reverse peptides and peptides identified by site only) during the filtering steps that resulted in the final list of 366 p65 / RELA interactors. In the light of this comment, we compared the 366 specific p65 / RELA interactors that we defined in our study with a recently published list of 3977 proteins that were identified in 113 BioID pulldowns from HEK293 cells using a GFP-BirA* fusion protein as the bait (new Appendix Fig. S9A) (Gawryski *et al*, 2024). 189 factors (51.6%) of our p65 / RELA interactors were absent from this list and more detailed analysis indicated that the overlapping ones were detected mostly at low frequency and low spectral counts (new Appendix Fig. S9B). Overall, we felt that these data sets were difficult to compare and to interpret, because they arose from a different cell type and BioID approach. For example, the TF TFE3, which

we confirmed as p65 / RELA interactor in our study by multiple means, is amongst the GFP contaminant list from Gawriyski et al (new Appendix Fig. S9). However, more detailed inspection of the data shows that TFE3 was found in only 16.8 % of GFP pulldowns at low spectral counts (new Appendix Fig. S9). We conclude that the long list of probable unspecific interactors of the GFP BioID data base might be most useful in large scale proteomic experiments that survey multiple TFs or other baits at a time, because it allows the definition of background sets of proteins by statistical modeling approaches. In our case, we decided to combine a targeted approach, user-defined filtering criteria and multiple follow-up analyses to justify the analysis strategy of our p65 / RELA interactome. To address this point in the text, we have added the following statement to the Limitations section of the discussion:

“Despite including three types of negative controls (omission of doxycycline or biotin, miniTurbo enzyme only) and the usage of p65 / RELA point mutants to confirm the specificity of the p65 / RELA interactome, we cannot fully exclude false positive hits. 189 factors (51.6%) of our 366 p65 / RELA interactors were absent from a large list of 3977 probable unspecific interactors obtained with a GFP-BirA* fusion protein in a different cell type, whereas the overlapping ones were detected mostly at low frequency and low spectral counts (Appendix Fig. S9) (Gawriyski *et al.*, 2024). This also applies to TFE3, whose interaction with p65 / RELA was validated in our study by several means (Appendix Fig. S9A). However, the statistical definition of background proteins using GFP baits is difficult to compare with our targeted approach that includes multiple levels of functional validation.”

-It is clear from the volcano plots that the empty vector (EV) does not serve as an effective control because the comparison yields a distribution that is not comparable to the bait samples.

Reply: It is unclear which Volcano plots the reviewer is referring to. The Volcano plots shown in Fig. 2E and Fig. EV2, left graph (previous Fig. S2A, left graph), which provide an overview of all measured proteins, clearly show the typical, butterfly-like distribution of the differences of all measured protein intensities and the significance of their enrichments, comparing the negative control (minTurbo) with the specific p65 / RELA bait. The other two Volcano plots shown in Fig. EV2A (middle and right graphs) provide additional views on the enrichments of proteins found with the p65 / RELA wild type bait in the presence of both, doxycycline and biotin, compared to the same bait in the presence of biotin only or doxycycline only. These Volcano plots display a one-sided distribution, resulting from the desired strong specific enrichment in the presence of both, doxycycline and biotin. All other Volcano plots in the manuscript highlight subgroups of interesting sets of proteins only.

-Many of the purported high-confidence interactors are detected in very low amounts (MS1, MS2) and with only a few peptides. Interactors with a relative abundance of less than 5% are most likely not meaningful, and some rationale or validation for including these should be provided.

Reply: The MS / MS counts, peptide intensities and number of peptides per identified protein of all 366 p65 / RELA interactors as defined in our study did not vary grossly compared with the entire set of identified proteins (Fig. 1 for review). We consider low abundance proteins as important to include in our data analyses, as in our experience many regulatory cellular factors have relatively low expression levels in LC-MS / MS data sets. The 38 high confidence p65 / RELA interactors have lower MS / MS counts, but they have sufficient peptide intensity values and, with the exception of N4BP3, were detected by at least two peptides per protein as shown in Fig. 1 for review and Fig. 3A.

Fig. 1 for review. Distribution of MS / MS counts, peptide intensities and number of detected peptides in the p65 /RELA interactome and the entire set of proteins.

-There should be protein-protein interaction (PPI) validation for minimally one-fifth of the 38 novel, high-confidence interactors mentioned in later analyses. Methods such as co-expression co-immunoprecipitation (co-IP), bimolecular fluorescence complementation (BiFC), or reciprocal interactome analysis using the prey as bait should be employed. These methods have been successfully used in several TF interactome studies for validation.

Reply: The possibility of false-positive hits is intrinsic to all interactome studies. Instead of broadly validating larger numbers of factors by additional PPI assays using ectopic expression systems, which bear the risk of perturbing physiological regulation, we decided to include several levels of functional analyses in our validation strategy. These efforts posed a limitation on the total numbers of preys to be tested. Nevertheless, we have thoroughly validated six factors, i.e. 15.8% out of 38 high-confidence interactors, functionally (S100A8/9, ZBTB5, GLIS2, TFE3, TFEB) and four, i.e. 10.5%, at the endogenous protein level by PLA (ZBTB5, TFE3, TFEB, GLIS, Fig. EV5). Additionally, our approach confirmed 46 (13%) known p65 / RELA interactors and further 199 (54%) indirect interactors in the larger interactome of 366 p65 / RELA interactors (Fig. 2). We do not understand the scientific basis for why validating at least 20% of all HCI should increase the consistency of our study or lead to new conclusions.

As pointed out in our responses to reviewer #1, we were able to confirm the TFE3 interaction with p65 / RELA by endogenous Co-IP (New Appendix Fig. S3B), but this approach is restricted to stable interactions and limited by the availability of suitable antibodies. The BiFC method is not established in our lab and would require ectopic expression of fusion proteins, which bears the risk of affecting the delicate regulation of the endogenous NF- κ B pathway as outlined above.

-The RELA ChIP-Seq experiments could benefit from repetition after knocking down TFE3 and GLIS2 using siRNAs.

Reply: This remark is in line with the comments of reviewer #1. Our lab has extensive experience in ChIP-qPCR / ChIP-seq analysis (Jurida et al, 2015; Poppe et al, 2017; Weiterer et al., 2020). Of all antibodies we used for the six factors chosen for further validation (TFE3, TFEB, S100A8/9, ZBTB5, GLIS2), only the anti TFE3 antibody worked in ChIP-qPCR and was used for the experiments shown in Fig. 8G (previously Fig. 7G). Unfortunately, in 2024, the company discontinued selling this particular antibody and the two new ones we purchased and tested extensively during the revision, failed in ChIP-qPCR. As illustrated by the immunoblots shown below, they poorly recognize TFE3 and instead cross-react extensively (Fig. 2 for review). To address this comment, as well as the comment of reviewer #1, we generated stable, lentivirally transduced HeLa cell lines and performed additional ChIP-qPCR experiments looking at p65 / RELA recruitment only. These data show a small, specific reduction of p65 / RELA recruitment in TFE3-deficient cells and have been included in a new Appendix Fig. S7. Beyond this, an extensive ChIP-seq-based characterization of p65 / RELA interactors will require further systematic studies including the implementation of alternative experimental systems that overcome the limited availability of ChIP-grade commercial antibodies. These studies are beyond the scope of our study.

Fig. 2 for review. Immunoblot validation of TFE3 antibodies. Extracts from cells deficient for p65 / RELA or TFE3 as described in the legend of Appendix Fig. S7 were analyzed by Western blotting as indicated to compare three different anti TFE3 antibodies. The rabbit antibody from Sigma worked well in our hands but was discontinued in 2024.

Minor comments:

-Figure 1A) AF model is not very useful and could be omitted.

Reply: We agree that the AF model shown Fig. 1A is not essential, but would like to keep it for readers not familiar with the p65 RELA domains and structures.

Referee #3

In the paper entitled « The proximity-based protein interaction landscape of the transcription factor p65 NF-kappaB / RELA and its gene-regulatory logics », the authors carried out an impressive and robust interactomic study to extensively identify new p65-binding proteins. This new approach led to the identification of already known interacting proteins such as NF-kappaB proteins p50, p52 etc., which demonstrates the robustness of their experimental design. Importantly, 366 new p65-associated candidates were identified by the authors, which reveals the complexity of all signaling pathways and biological processes controlled by p65/RelA. As expected, many of them are acting as transcription factors or chromatin remodelers and histone-modifying enzymes. To validate the biological relevance of these multiple new interactions, the authors conducted a siRNA-based screening to functional establish a link between new p65-associated proteins and NF-kappaB-dependent gene expression. Overall, this is a very interesting study that merits publication. I have few comments that should help to further improve the manuscript.

Reply: We thank the reviewer for this very positive evaluation of our study and for multiple important suggestions.

Major points

1. In Figure 3C in which the authors assess the relevance of new p65-associated proteins in the expression of NF-kappaB target genes, they show that the depletion of some candidates impairs some but not all tested NF-kappaB target genes, suggesting some specificity of these interactions for the regulation of some but not all target genes. The authors should comment this finding in the discussion. What would be the underlying mechanism ?

Reply: We assume that different cofactor assemblies repress or activate individual NF-κB target genes, similar to observations from an earlier shRNA screen from our laboratory (Meier-Soelch *et al*, 2018). We have added this interpretation to the discussion as follows:

“The TFs exhibited disparate quantitative contributions to basal and IL-1α-inducible gene expression, encompassing both gene activation or repression. These phenotypes are likely indicative of TF cooperativity in the fine-tuning of NF-κB responses in conjunction with gene-specific

coactivator/corepressor assemblies, in accordance with observations from an earlier targeted shRNA screen of nuclear cofactors conducted in murine fibroblast cells (Meier-Soelch et al., 2018)."

2. For selected p65-associated proteins, the authors confirmed these interactions with PLA (Supplementary Figure 4). However, that would also be nice to confirm some interactions at the endogenous level through co-immunoprecipitation studies, especially for candidates whose antibodies are available (TF3B for example as nice western blots for this candidate are illustrated in Figure 4B). I would for example select some candidates whose interaction is expected to be regulated upon IL-1 stimulation. Very good co-immunoprecipitations would further validate their findings.

Reply: To address this comment, we performed several co-immunoprecipitation experiments. As outlined in our response to reviewer #1, we were able to confirm the interaction of p65, p50 and TFE by endogenous Co-IP. Unfortunately, all other Co-IPs (for GLIS2 and TFEB) did not work. Co-immunoprecipitation will only allow detection of protein interactions that are sufficiently abundant and remain stable in cell lysates and is heavily dependent on suitable antibody pairs. The failure of these Co-IPs, therefore, does not compromise the relevance of our findings.

3. In Figure 4D, the authors demonstrate that GLIS2 knockdown reduced p65 protein levels. The authors should discuss the possibility that GLIS2 prevents p65 degradation, possibly by competing with any E3 ligases for binding to this NF-kappaB family member.

Reply: This is an important point, which we included in the discussion as follows:

"The reduced p65 / RELA levels in GLIS2 knockdown cells possibly suggest a role of GLIS2 in preventing E3 ligase-mediated ubiquitination and proteasomal degradation of p65 / RELA that has been observed in several systems (Geng et al, 2009)."

4. The authors are interested in IL-1alpha signaling but one may wonder whether most of the described interactions with p65 could also be found in cells treated with TNFalpha, a widely studied signaling pathway. That would be nice to assess this issue for some p65-associated proteins.

Reply: In general, we assume that, besides the canonical NF- κ B components, large parts of the proximity-based interactomes are cell type and stimulus-specific. We have addressed this point using affinity proteomics data from Bouwmeester et al., who identified 92 p65 / RELA interactors from TNF α -stimulated HEK293 cells by tandem affinity purification (Bouwmeester *et al*, 2004). 80 of these interactors mapped to Uniprot IDs and only seven (REL, KPNA3 NFKB1, MRPL12, NFKBIA, NFKB2, RELA) overlapped with our 366 p65 /RELA interactors (Dataset EV5). Hence, thoroughly addressing this interesting question will require repeating the miniTurbo approach with TNFalpha in future studies.

5. As p65 promotes cell survival in most cell types and given the fact that NF-kappaB limits TNF-dependent cell death through the expression of target genes such as c-FLIP, etc...do some p65-associated proteins also prevent TNF-dependent cell death ? The authors could deplete each candidate and assess the consequence on cell apoptosis upon TNF stimulation. These experiments would also increase, if relevant, the biological relevance of their findings.

Reply: To address this point, we knocked down TFE3 and TFEB along with p65 (as a control) using siRNAs and assessed cell death using MTS assays in the presence or absence of TNFalpha and cycloheximide according to references (Miura et al, 1995; Wajant et al, 2003). The data show no additional cytotoxicity or protection following reduction of TFE3 or TFEB, thus these two factors appear to have roles other than regulation of cell death (see Fig. 3 for review). Our study focuses on the constitutive and IL-1-induced interactome and already contains an extensive discussion of the results, including the new data obtained during the revision. Therefore, we have refrained from presenting these data in the revised manuscript. We concur with the opinion that a comparison of p65-HA-mTb interactomes following IL-1 or TNF α stimulation would prove highly insightful, but the clarification of this question requires extensive future studies.

Fig. 3 for review. Transient suppression of TFE3 or TFEB does not affect TNF α -induced cell death. (A) HeLa cells were transfected with 20 nM of siRNAs directed against p65 / RELA, TFE3, TFEB or luciferase (as a control) for 48 h. Thereafter, transfected or parental cells were treated with cycloheximide (CHX, 10 μ g/ml), TNF α (20ng/ml) or solvent (DMSO) for 24 h. Cell death was determined by MTS assay. The graphs show relative cell viability (compared to the mean of samples treated with luciferase siRNA and DMSO). Box plots show data points from four biologically independent experiments including technical triplicates with means and minimum/maximum values. Asterisks indicate significant changes according to one-way ANOVA (* $P \leq 0.05$, ** $P \leq 0.01$, *** $P \leq 0.001$, **** $P \leq 0.0001$). (B) Suppression of p65 / RELA, TFE3 or TFEB was confirmed by immunoblotting. Shown is one representative out of three experiments.

6. The authors should also discuss or even experimentally address the possibility that some interactions are regulated by p65 phosphorylation, especially the ones found on IL1alpha-stimulated cells. Does p65 phosphorylation enhance some interactions? Using WT p65 or a mutant lacking the Serine 536 residue known to be phosphorylated upon TNF stimulation would be a good strategy to look for interaction by IP with some identified candidates.

Reply: This is a very interesting and important point. In general, as discussed in a number of reviews, it has been difficult to nail down the mechanisms and biological functions of Ser536 (or Ser468) phosphorylations (Christian *et al*, 2016; Riedlinger *et al*, 2017; Wietek & O'Neill, 2007).

As pointed out in our response to reviewer #1, we have analyzed the phosphorylation of the p65-HA-mTb bait protein in new experiments and also re-analyzed our mass spectrometry data for phosphorylation sites. These new data provide additional evidence for phosphorylation-dependent remodeling of the p65-HA-mTb interactome as shown in two new Fig. EV3 and Fig. EV4.

Clarifying the functional effects of phosphorylation on novel interactions partners will require overexpression, optimized reconstitution systems or genome-editing using p65 phosphorylation-deficient mutants combined with a suitable, sensitive interaction assay.

Given the difficulties and limitations in confirming proximity-based observations by conventional interactions assays, as discussed above, at this point we prefer to discuss this issue in the Discussion section. Clarifying the role of p65 phosphorylations more general, at the level of proximity-based interactomes, is beyond the scope of our present work and will be subject of another study.

Minor points

1. There is a typing mistake in the title mentioned on the website of EMBO reports...

Reply: We have corrected this.

References

- Bouwmeester T, Bauch A, Ruffner H, Angrand PO, Bergamini G, Croughton K, Cruciat C, Eberhard D, Gagneur J, Ghidelli S *et al* (2004) A physical and functional map of the human TNF-alpha/NF-kappa B signal transduction pathway. *Nat Cell Biol* 6: 97-105
- Branon TC, Bosch JA, Sanchez AD, Udeshi ND, Svinkina T, Carr SA, Feldman JL, Perrimon N, Ting AY (2018) Efficient proximity labeling in living cells and organisms with TurboID. *Nat Biotechnol* 36: 880-887
- Christian F, Smith EL, Carmody RJ (2016) The Regulation of NF-kappaB Subunits by Phosphorylation. *Cells* 5
- Gawryiski L, Tan Z, Liu X, Chowdhury I, Malaymar Pinar D, Zhang Q, Weltner J, Jouhilahti EM, Wei GH, Kere J *et al* (2024) Interaction network of human early embryonic transcription factors. *Embo Rep* 25: 1589-1622
- Geng H, Wittwer T, Dittrich-Breiholz O, Kracht M, Schmitz ML (2009) Phosphorylation of NF-kappaB p65 at Ser468 controls its COMMD1-dependent ubiquitination and target gene-specific proteasomal elimination. *EMBO Rep* 10: 381-386
- Jurida L, Soelch J, Bartkuhn M, Handschick K, Muller H, Newel D, Weber A, Dittrich-Breiholz O, Schneider H, Bhujra S *et al* (2015) The Activation of IL-1-Induced Enhancers Depends on TAK1 Kinase Activity and NF-kappaB p65. *Cell Rep* 10: 726-739
- Meier-Soelch J, Jurida L, Weber A, Newel D, Kim J, Braun T, Schmitz ML, Kracht M (2018) RNAi-Based Identification of Gene-Specific Nuclear Cofactor Networks Regulating Interleukin-1 Target Genes. *Frontiers in Immunology* 9
- Meier-Soelch J, Mayr-Buro C, Juli J, Leib L, Linne U, Dreute J, Papantonis A, Schmitz ML, Kracht M (2021) Monitoring the Levels of Cellular NF-κB Activation States. *Cancers* 13
- Miura M, Friedlander RM, Yuan JY (1995) Tumor Necrosis Factor-Induced Apoptosis Is Mediated by a Crma-Sensitive Cell-Death Pathway. *P Natl Acad Sci USA* 92: 8318-8322
- Muller MR, Rao A (2010) NFAT, immunity and cancer: a transcription factor comes of age. *Nat Rev Immunol* 10: 645-656
- Poppe M, Wittig S, Jurida L, Bartkuhn M, Wilhelm J, Muller H, Beuerlein K, Karl N, Bhujra S, Ziebuhr J *et al* (2017) The NF-kappaB-dependent and -independent transcriptome and chromatin landscapes of human coronavirus 229E-infected cells. *PLoS Pathog* 13: e1006286
- Riedlinger T, Dommerholt MB, Wijshake T, Kruit JK, Huijkman N, Dekker D, Koster M, Kloosterhuis N, Koonen DPY, de Bruin A *et al* (2017) NF-kappaB p65 serine 467 phosphorylation sensitizes mice to weight gain and TNFalpha-or diet-induced inflammation. *Biochim Biophys Acta Mol Cell Res* 1864: 1785-1798
- Wajant H, Pfizenmaier K, Scheurich P (2003) Tumor necrosis factor signaling. *Cell Death Differ* 10: 45-65
- Weiterer SS, Meier-Soelch J, Georgomanolis T, Mizi A, Beyerlein A, Weiser H, Brant L, Mayr-Buro C, Jurida L, Beuerlein K *et al* (2020) Distinct IL-1alpha-responsive enhancers promote acute and coordinated changes in chromatin topology in a hierarchical manner. *EMBO J* 39: e101533
- Wietek C, O'Neill LA (2007) Diversity and regulation in the NF-kappaB system. *Trends Biochem Sci* 32: 311-319

Dear Dr. Kracht

Thank you for the submission of your revised manuscript to EMBO reports. We have now received the full set of referee reports that is copied below.

As you will see, all referees are positive about the revised study and recommend publication.

Browsing through the manuscript myself, I noticed a few editorial things that we need before we can proceed with the official acceptance of your study.

- 1) Please reduce the number of keywords to 5.
- 2) Please rename the Competing interests statement to "Disclosure Statement and Competing Interests" and place it after Acknowledgments.
- 3) Regarding the Author Contributions, we now use CRediT to specify the contributions of each author in the journal submission system. Therefore, please remove the Author Contributions from the manuscript file and make sure that the author contributions in our online manuscript tracking system are correct and up-to-date. The information you specified in the system will be automatically retrieved and typeset into the article. You can enter additional information in the free text box provided, if you wish.
- 4) Author checklist: Please choose 'N/A' as response for D82. You already state that no blinding was done in the Checklist and I don't think such a statement is required in the methods section in this case as blinding seems not a requirement here. Please also select responses for the following cells to complete the checklist: D87, D88 (e.g. N/A)
- 5) The information on funding in the manuscript Acknowledgment section and in the online submission tracking system must be congruent. Please note that the information you give in the online system is the most relevant one that will be visible in PubMed. We note that in some cases the grant was added in the online system but not the related project number. Moreover, it might be necessary to add the following funding information in the online manuscript tracking system: LOEWE program of the state of Hesse, the IMPRS program of the Max Planck Society and the Excellence Cluster, DZL/UGMLC/ILH program. Please check this.
- 6) Figure callouts: "Supplementary Information" on p23, line 1092 needs to be updated as we have no file types called "Supplementary". It is either "Appendix" or "Expanded View".
- 7) EV Datasets: each Dataset should be uploaded as separate Excel files: Dataset EV1, etc.; each Dataset should have a legend provided as a separate tab/sheet in each Excel file.
- 8) Appendix: please add page numbers (also to the Table of Content) and rearrange so that each figure legend follows each figure.
- 9) Appendix Figure S3A: even though you analysed a high number of cells, the data originate from two independent experiments and therefore the statistical analysis seems not appropriate, as it might result in pseudoreplication. Please clarify.
- 10) Appendix Figure S4, S5, S7: Please list the exact p-values in the legend (or the figure panel).
- 11) Please re-group the source data as follows: all folders/panels of one figure need to be zipped up so that we have one folder uploaded per one figure.
- 12) Doing a spot check I noticed that the source data for Fig. 1B and 1D are both in the folder labeled "1D".
- 13) Data availability section:
 - Please provide the specific URL for the "PXD045888" dataset in the data availability statement.
 - Please be reminded to remove the reviewer access information.
 - Please remove the following text, because the Data availability statement should only refer to datasets that were generated in this study and deposited at public repositories:
"For KB cells, RNA-seq, ChIP-seq and ATACseq data are available via our previous NCBI GEO submissions with the accession numbers GSE64224, GSE52470 and GSE134436 (<https://www.ncbi.nlm.nih.gov/geo>) (Jurida et al., 2015; Weiterer et al., 2020).
The remaining data generated in this study are provided in the Supplementary Information / Source Data sections. Source data are provided with this paper."

- The reference to your previous datasets from Jurida et al., 2015 and Weiterer et al., 2020 should be included as a data reference where applicable (either in the results or in the methods section, or both). If you use data references, you should first refer to the publication and then in addition add a reference to the dataset itself. E.g. (Jurida et al, 2015; Data ref: Jurida et al, 2015). Your reference list will then list the paper and the data reference, the latter must provide the database name, accession number/identifiers and a resolvable link to the landing page from which the data can be accessed at the end of the reference. It is labeled with "[DATASET]"

Further instructions are available at <<https://www.embopress.org/page/journal/14693178/authorguide#referencesformat>>

14) Our production/data editors have asked you to clarify several points in the figure legends (see below). Please incorporate these changes in the manuscript and return the revised file with tracked changes with your final manuscript submission.

A) Statistical test information. Only p-values that are actually shown in the figure panel(s) should (and must) be defined in the legends, all others should be removed from (or added to) the legend. Moreover, we ask for the specification of exact p-values:

- Please indicate the statistical test used for data analysis in the legends of figures 1f, 1h, 3e, 3f, 3g, 4d, 5a, 6b, 7b, 8b, 8c, EV2b, EV4c.

- Please note that the exact p-values are not provided in the legends of figures 5d, 5e, 6c, EV5a, EV5b, EV5c, EV5d.

B) Replicates and error bars:

- Please note that the box plots need to be defined in terms of minima, maxima, centre, bounds of box and whiskers, and percentile in the legend of figure 8g.

- Please note that the error bars are not defined in the legend of figure EV3b.

C) Data presentation:

- "Please note that axis gap are not labeled appropriately in figure EV3d.

15) As a standard procedure we modify the title and abstract of the manuscript to make it more accessible to a general readership. Please find the suggested edits below my signature.

16) Please remove the "Highlights" from the manuscript text.

17) Please provide a short (1-2 sentences) summary of the findings and their significance and a schematic summary figure that provides a sketch of the major findings (not a data image).

We need the summary figure as a separate file in PNG or JPG format at a size of 550x300-600 pixels (width x height). Please note that the size is rather small and that text needs to be readable at the final size.

With kind regards,

=====

Referee #1:

The authors have satisfactorily addressed my original concerns regarding this manuscript and I have no additional issues.

I have also, as requested, looked closely at the authors response to Referee 2. I should say that my lab has not performed Bio-ID type approaches so I would not consider myself an expert in this area.

While Referee 2 did raise a number of major concerns about the data, in particular the controls used, I found the response of the authors to be reasonable and convincing.

For example, Referee 2 thinks it would be important to use a mini-Turbo tagged GFP/LacZ, both with and without an NLS-sequence. The authors respond by pointing out that RelA will be both cytoplasmic and nuclear in the cells being studied, and importantly, that mini-Turbo also localises to both compartments. The authors also confirm the specificity of many of the interactors identified through the use of RelA point mutations.

I am therefore of the opinion that the authors have addressed the concerns of Referee 2 and made appropriate changes to the

text of the paper to reflect these issues.

Referee #2:

The authors have made a strong effort to address the reviewers' concerns by providing new experimental data and validation that enhance the robustness of their findings. The use of proximity labeling with p65-miniTurbo fusion proteins, as well as the PLA and co-immunoprecipitation assays for interactors like TFE3, offer meaningful validation (Appendix Fig. S3). These additions help to solidify the interactome analysis, particularly with respect to key components in NF- κ B signaling.

That said, there remain some limitations, particularly the absence of a more robust nuclear-specific control, such as miniTurbo-tagged GFP/LacZ with an NLS sequence. While the authors justify their approach by noting that p65/RELA is not restricted to the nucleus, additional controls would have better addressed concerns about non-specific interactors, particularly in the nuclear compartment (Fig. EV1). The use of in silico analysis is helpful, but direct experimental validation in this regard would strengthen the conclusions.

The authors have successfully validated some novel interactors, yet the relatively low abundance of many proteins in the dataset warrants caution. The biological relevance of certain interactors, especially those detected in low amounts, remains somewhat uncertain due to the limited protein-protein interaction (PPI) validation. While proximity labeling is a powerful tool, the potential for false positives calls for broader validation of the interactors.

Despite these caveats, the manuscript makes a valuable contribution to our understanding of the NF- κ B interactome. The functional validation of several interactors, combined with thoughtful experimental design, provides sufficient grounds for publication. However, further validation of key findings and additional controls would further elevate the impact of this study

Referee #3:

I am satisfied with additional experiments and comments provided by the authors in this revised manuscript. Therefore, this version is ready for publication in my view.

Note that the typing mistake in the title still appears on the website of the journal ("The promximity...").

=====

Suggested title and abstract:

The proximity-based protein interactome and regulatory logics of the transcription factor p65 NF- B / RELA

The protein interactome of p65 / RELA, the most active subunit of the transcription factor (TF) NF- B, has not been previously determined in living cells. Using p65-miniTurbo fusion proteins and biotin tagging, we identify > 350 RELA interactors from untreated and IL-1 α -stimulated cells, including many TFs (47 % of all interactors) and > 50 epigenetic regulators belonging to different classes of chromatin remodeling complexes. A comparison with the interactomes of two point mutants of p65 reveals that the interactions primarily require intact dimerization rather than DNA binding properties. A targeted RNAi screen for 38 interactors and subsequent functional transcriptome and bioinformatics studies identifies gene regulatory (sub)networks, each controlled by RELA in combination with one of the TFs ZBTB5, GLIS2, TFE3 / TFEB or S100A8 / A9. The large, dynamic and versatile high resolution interactome of RELA and its gene-regulatory logics provides a rich resource and a new framework for explaining how RELA cooperativity determines gene expression patterns.

All editorial and formatting issues were resolved by the authors.

Michael Kracht
Justus Liebig University Giessen
Rudolf Buchheim Institute of Pharmacology
Schubertstrasse 81
Giessen 35392
Germany

Dear Dr. Kracht,

I am very pleased to accept your manuscript for publication in the next available issue of EMBO reports. Thank you for your contribution to our journal.

Yours sincerely,
